# KEIC: On Editing In-Context Knowledge of Large Language Models in Conversations

## Abstract

Large language models (LLMs), such as GPT-4, are adept at generating coherent and fluent responses within conversational contexts. However, there has been a paucity of comprehensive research exploring LLMs to dynamically update their knowledge in response to corrections of misinformation provided by users during dialogue sessions. In this paper, we present a novel framework termed Knowledge Editing In Conversation (KEIC), along with an accompanying dataset, devised to assess the efficacy of LLMs in aligning the user update in an in-context setting, given the previous chat history containing a false statement that conflicts with the subsequent user update. Through in-depth investigations, we observe that the contemporary LLMs exhibit a modicum of proficiency in this task. To enhance their in-context knowledge editing abilities, we propose a structured strategy to handle the information update for LLMs in a multi-turn conversation. We demonstrate that our approach is effective and suggest insights for research communities in this emerging and essential issue.

## 1 Introduction

Fluidity and inconsistency are characteristics of natural conversations. It is not rare to encounter scenarios where an individual's initial statement is based on false or obsolete information. As the conversation progresses, the speaker may rectify their statements upon recognizing an error or when presented with fresh information. Intriguingly, the other speaker adapts seamlessly to these changes and continues carrying on the conversation. From the cognitive psychology perspective, this adaptive process involves entailing the information update or alteration of stored knowledge that has already been in one's memory (Schrauf & Rubin, 2000; Wagner et al., 2000).

Over the past few years, the advancements in large language models (LLMs) have fostered an environment where people find it commonplace to engage in extended conversations with chatbots (OpenAI, 2022; 2023; Touvron et al., 2023; Zheng et al., 2023b; Team et al., 2023; 2024; Dubey et al., 2024, *inter alia*). These dialogues often encompass the sharing of daily experiences and emotional exchanges. A critical attribute for LLMs—especially in long-term interaction—is the capacity to have such **adaptability** similar to humans, meaning *the LLM should be adept at updating any misinformation or outdated knowledge shared by the human interlocutor earlier in conversation* (Huang et al., 2024). This adaptability feature, which we termed in-context knowledge editing (KE) or Knowledge Editing In Conversation (KEIC), is crucial for LLMs to serve as intelligent, long-term conversational companions.

Henceforth, a natural question arises: Do state-of-the-art LLMs have an innate capacity for KEIC? Before answering this, we summarize the advantages that LLMs shall be equipped with once they are proficient at KEIC, envision several real-world scenarios that favor models with KEIC capacity, and provide reasons why current approaches may not be suitable under these circumstances. Related work is in Appendix A.

These include: (1) Not all false statements require (and should not do so) parameter editing, as some of them are non-factual (see Figure 1). (2) To achieve KEIC, the LLM shall excel in temporal and contextualized information in an entire dialogue, as it can deal with real-time user intent, discern which information is obsolete, and resolve any discrepancies thereafter. (3) End users do not need to prepare examples for LLMs (Zheng et al., 2023a), nor to re-initiate the dialogue sessions, especially when conversations grow longer. In practice, the model can seamlessly update its knowledge

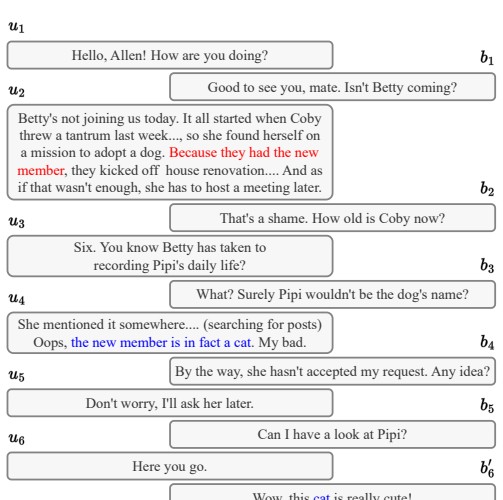

Figure 1: An example of $u$ and $b$ having a conversation. $u_2$ contains the false information; $u_4$ contains new information. Speaker $u$ directly corrects his false statement in $u_2$ (connected by "new member"). Note that $b_6'$ *inevitably* contradicts $b_3$, but it is reasonable. Though "this dog is really cute" does not make $b$ contradict himself, it sounds weird as though $b$ ignores what $u$ said. The KEIC task assesses whether an LLM can (1) identify the update, (2) locate the false statement within a long utterance, and (3) adapt to this change in a long-term conversation.

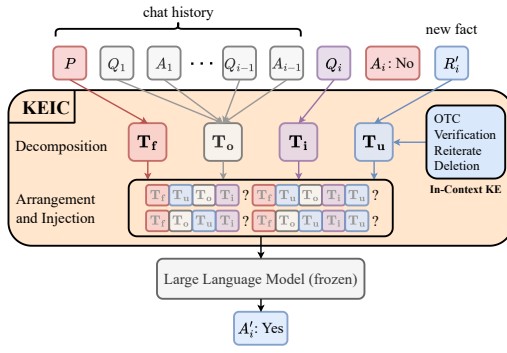

Figure 2: A high-level view of the KEIC framework: Given chat data (either human-human or human-AI) and a new fact, it decomposes the chat (in this paper, CoQA) into disjoint phases and performs operations to update an LLM's response. We expound the CoQA task in §2.1, what a new fact is in §2.2 (how they are generated in §4.1), four components in Decomposition in §2.3, how to map arbitrary dialogue into them in §2.4, and four in-context KE methods in §3. Each method has two settings in Arrangement and Injection (whether the new information is closer to the misinformation; see §4.4). We consider an LLM updates its knowledge if its answer to the same question is changed (*e.g.*, "No" → "Yes"), then we evaluate this "update" behavior on four LLMs (see §4.3). The terms fact, information, and knowledge are used interchangeably throughout this paper.

through a more efficient KEIC process by patching user mistakes.[1] Moreover, demonstrations often introduce undesired biases (Zhao et al., 2021; Lu et al., 2022) and overestimate the LLM's ability. (4) Traditional KE may be impractical for a few false facts since fine-tuning a few examples tends to overfit. In addition, most end users do not acquire the skills and resources to access and modify the LLMs (Yuksekgonul et al., 2024). (5) Current evaluations of KE are limited to testing the generality and specificity around the edited facts (Cohen et al., 2024), and it remains unclear whether modifying parameters has a significant impact on other task domains (Chen et al., 2023). In contrast, our proposed methodology circumvents such potential aftermath. (6) Analogous to the previous point of view, since the LLM parameters are frozen, it is transferable to other downstream tasks and can be shared by many users. Though maintaining additional models to perform KE also preserves the parameters (Mitchell et al., 2022b), it requires the memory "write" operation on another model, which is time-consuming. Furthermore, maintaining each individual's memory, classifier, and counterfactual model to keep them up-to-date is one of the most challenging aspects.

Based on the aforementioned perspectives, we explore whether LLMs can perform KEIC. Practically, if we can edit an LLM's in-context knowledge on the fly, there would be no need to modify its underlying parameters (Rafailov et al., 2023; Ethayarajh et al., 2024) or maintain additional models to rectify misinformation. As existing research often neglects this aspect to our knowledge, we formalize this task and propose a KEIC framework to measure the adaptability of LLMs (see Figure 2).

Our main contributions are three-fold:

- We introduce a new KEIC task for LLMs to be intelligent companions. We devise the KEIC framework to decompose a multi-turn dialogue and cope with the misinformation in the earlier conversation. The concept also applies to hallucination, the notorious problem of LLMs, and could further improve their reliability in a zero-shot and in-context setting.

---

[1] It is also strange if we have to provide examples so LLMs can update the pet's type (dog → cat) in Figure 1.

- We construct the challenging, human-annotated dataset to serve as a benchmark for the KEIC task. Our dataset of size 1,781 comprises topics from factual knowledge to non-factual narrative stories, suitable for overall adaptability assessment in long-term dialogue.

- We propose four model-agnostic KEIC methods, one of which is an algorithm for self-correction. Extensive results show that the Reiterate method (in Section 3) is overall effective and that GPT-3.5 exhibits a significant performance improvement with our approach.

## 2 TASK DEFINITION

The KEIC task aims to test if an LLM can dynamically update its knowledge when the user corrects the previous false fact. We first outline the CoQA task (Reddy et al., 2019) since we construct our KEIC dataset from it. Next, we define how to elicit knowledge stored in LLMs and formalize its form in a conversation. Finally, we present the KEIC framework and show it can fit any chat data.

### 2.1 CoQA FRAMEWORK

The CoQA task aims to test whether a dialogue system can answer the question $Q_i$ when a passage $P$ and previous chat history $\{Q_1, A_1, ..., Q_{i-1}, A_{i-1}\}$ are given. Each question-answer pair $(Q_i, A_i)$ is associated with a consecutive text span of **support sentence** $R_i \in P$ that serves as a rationale for answering $Q_i$. The conversation flow is denoted as $[P, Q_1, A_1, ..., Q_i, A_i]$. The term passage is used interchangeably with story.

### 2.2 THE FORM OF FACT

An intuitive way to probe knowledge acquired by LLMs is by asking questions (Levy et al., 2017; De Cao et al., 2021; Zhong et al., 2021; Meng et al., 2023). We assume fact or knowledge presented in the context $\mathcal{C}$ with the form: $(r, q, a)$, where $r \in \mathcal{C}$ is the text, $q$ is the question related to $r$, and $a$ is the answer to $q$. Given a fact $(r, q, a)$, it is straightforward yet informal to define the new fact $(r', q, a')$ as:

$$\exists r' \neq r \text{ s.t. } a' \neq a \tag{1}$$

However, as autoregressive LLMs are sensitive in nature, we cannot rule out the possibility that there exist two semantically invariant contexts that produce different answers to the same question. To ensure the effectiveness of the information update, we define a mapping $\mathcal{M} : X \to \tau$, where $X$ is a text string and $\tau_X = (\text{s}, \text{o}, \text{r})$ is the **subject-object relation** triplet of $X$. Note that the relation "r" is different from the notation of fact "$r$" (in italics). Then, we denote $\Delta_X$ (or, $\Delta(X)$ to avoid overusing subscript) as the set of tuples that are different from $\tau_X$:[2]

$$\Delta_X = \big\{(\text{s}', \text{o}, \text{r}), (\text{s}, \text{o}', \text{r}), (\text{s}, \text{o}, \text{r}') : \exists \tau_X \in \mathcal{M}(X) \wedge \text{s}' \neq \text{s} \wedge \text{o}' \neq \text{o} \wedge \text{r}' \neq \text{r}\big\} \tag{2}$$

Let $\mathcal{Y}$ be an LLM's output space and $a \in \mathcal{Y}$, we formally define new knowledge $(r', q, a')$ as **effective** if and only if

$$\exists \mathcal{M}(r) \text{ s.t. } \mathcal{M}(r') \in \Delta(r) \text{ and } a' \in \{x \in \mathcal{Y} : x \neq a\} \tag{3}$$

In this work, $\mathcal{C}$ is the text in the conversation. We bridge the gap of knowledge and the $(R_i, Q_i, A_i)$ tuple in CoQA since they share the same form. Because answers are free-form in CoQA, we focus on Yes/No (YN) questions to simplify the analysis, and thus $\mathcal{Y} = \{\text{Yes, No}\}$.[3] For instance, given a fact $(r, q, a) = $ (*Michael Jordan played fifteen seasons in the NBA*, *Did Jordan play basketball, Yes*) and its triplet $\mathcal{M}(r) = $ (Michael Jordan, basketball, played sport), one effective fact is $r' = $ "*Michael Jordan played fifteen seasons in the MLB*" because $\mathcal{M}(r') = $ (Michael Jordan, baseball, played sport) $\in \Delta(r)$ and $a' \in \{\text{No}\}$. For readability, when the term knowledge is mentioned, we typically refer to the text of knowledge instead of a tuple unless otherwise stated.

---

[2]Let $X$ be "Alice is Bob's mom," the set $\Delta_X$ can be {(Amy, Bob, isMom), (Alice, Bill, isMom), (Alice, Bob, isNotMom)}. Symbols with apostrophes denote effective.

[3]Note that the effective definition does *not* limit to YN questions. If we ask "Who is the current U.S. president?", then $\mathcal{Y} = \{\text{Donald Trump, Joe Biden}\}$.

## 2.3 KEIC Framework

In our KEIC dataset, we extend each instance from CoQA by labeling misinformation in the passage and adding a correction. We denote a $k$-turn conversation as $[T_1, ..., T_k]$, where $T_j$ is the $j$-th turn $\forall j \in [1, k]$, and *each turn $T_j = (u_j, b_j)$ is a pair of user and chatbot utterances*, respectively. The conversation flow is $[u_1, b_1, u_2, ..., b_k]$ after unrolling.

In the scenario of the KEIC task, there exist (1) a false fact, (2) a new fact, and (3) other contexts in a conversation. Consequently, we design the KEIC framework to segment the dialogue into smaller components, which also adheres to the multi-turn evaluation framework (Zheng et al., 2023b). We define four disjoint phases, and each turn of the conversation belongs to one of them. The concept is as simple as classifying each turn into four categories:

- **False phase** ($\mathbf{T_f}$) contains false information, and the user will point it out later. In Figure 1, the false fact is in $T_2$ (specifically, $u_2$), and the user corrects it in $T_4$ (specifically, $u_4$).

- **Update phase** ($\mathbf{T_u}$) involves in updating misinformation or in-context KE process. $\mathbf{T_u}$ is a *general* notation for KEIC (see Section 3). In Figure 1, the update phase has $T_4$.

- **Test phase** ($\mathbf{T_i}$) assesses if the update phase rectifies an LLM's knowledge successfully. In Figure 1, we can ask the LLM "Is Pipi (Betty's pet) a cat?" in $u_7$ to evaluate its answer.

- **Other phase** ($\mathbf{T_o}$) consists of the previous, on-going chat. One may think any turn here is more or less unrelated to KEIC. In Figure 1, $T_1$, $T_3$, $T_5$, and $T_6$ are in the other phase.

## 2.4 Mapping Arbitrary Dialogue into KEIC Framework

To standardize our KEIC methods and dataset construction in this task, we elaborate on the Decomposition in Figure 2, using CoQA data as an example. We mathematically define the above mapping process as $f : \{T_1, ..., T_k\} \rightarrow \{\mathbf{T_f}, \mathbf{T_u}, \mathbf{T_i}, \mathbf{T_o}\}$. For each turn $T_j$, the mapping $f$ works as follows:

- If either $u_j$ or $b_j$ (hallucination) contains false information, then $T_j \in \mathbf{T_f}$. In CoQA data, $T_1$ is always in the false phase because we render a piece of text in the passage $P$ obsolete for the user to correct afterward (and $P \in u_1$).

- If $u_j$ updates misinformation in the false phase ($u_j$ is effective) or involves in KEIC process, then $T_j \in \mathbf{T_u}$. CoQA does not have this phase. We devise four methods in Section 3.

- If $u_j$ consists of the question with which we want to test the LLM, then $T_j \in \mathbf{T_i}$. In CoQA, it is a question and is usually the last turn.

- Any $T_j$ that does not belong to the false, update, and test phases falls into the other phase. In CoQA, if the $i$-th question is selected among $\big\{(Q_1, A_1), ..., (Q_n, A_n)\big\}$ for the test phase, then its previous QA pairs $\bigcup_{m=1}^{i-1}(Q_m, A_m)$ fall into the other phase. If $i = 1$, then $\mathbf{T_o} = \emptyset$.

# 3 Knowledge Editing In Conversation (KEIC) Methods

We propose four methods in the update phase: One-turn correction, Verification, Reiterate, and Deletion. In this paper, $\mathbf{T_u} = \{\mathbf{T_c}, \mathbf{T_v}, \mathbf{T_r}, \mathbf{T_d}\}$, where each approach can be combined with the others.

**One-Turn Correction (OTC)**  One-turn correction is a **correction phase** ($\mathbf{T_c}$) that contains a single sentence (see $u_4$ in Figure 1). Once an LLM exhibits innate KEIC similar to humans, a simple OTC shall suffice. We apply the mining approach (Jiang et al., 2020) to extract the correction utterances from the DailyDialog (Li et al., 2017). Specifically, we select 15 sentences using 15 keywords that may be associated with corrections. For example, *"Wrong. It's not [old fact], but [new fact]."* and *"Actually, [new fact]."* are two types of our templates (whether the templates contain the old fact; see Appendix B for all). In this paper, we are explicitly referring to the simplest KEIC method when OTC is mentioned.

**Verification ($\mathbf{T_v}$)**  After the test phase, we launch the Verification to confirm if an LLM is sure of its response via re-questioning (see $u_9$ in Figure 3b). It mimics a real-world scenario when one shows disbelief or skepticism, which may stimulate it to reflect on the user update.

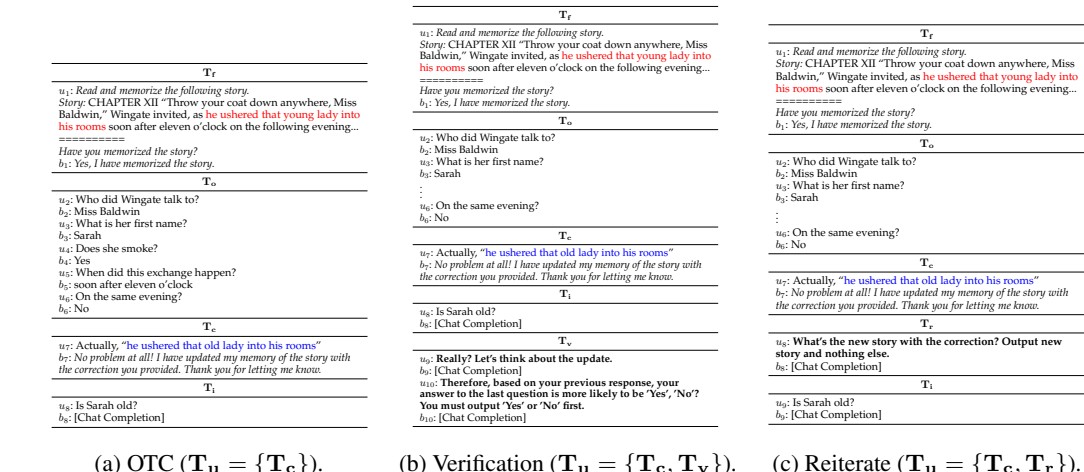

(a) OTC ($\mathbf{T_u} = \{\mathbf{T_c}\}$). (b) Verification ($\mathbf{T_u} = \{\mathbf{T_c}, \mathbf{T_v}\}$). (c) Reiterate ($\mathbf{T_u} = \{\mathbf{T_c}, \mathbf{T_r}\}$).

Figure 3: The exact prompt for the OTC, Verification, and Reiterate method (see Appendix C for the Deletion). The texts ($u_1$, $b_1$, and $b_7$) in italics are *pre-defined* (*i.e.*, fixed) and used in *all* experiments. Bold texts in Verification (Really? Let's think about the update.) and Reiterate (What's the new story with the correction? Output new story and nothing else.) are also pre-defined. The variation is the user utterance (see Appendix B). LLMs need to generate texts in "[Chat Completion]."

**Reiterate ($\mathbf{T_r}$)** As the LLM may overlook the importance of user correction, we introduce a Reiterate phase immediately after it (see the bold text in Figure 3c). It prompts the LLM to reiterate the corrected information from the psychological perspective (Bartlett, 1995). If an LLM generates a context containing the new fact in place of the old one, we define Reiterate as successful.[4]

**Deletion ($\mathbf{T_d}$)** If an LLM still performs poorly in Verification and Reiterate, we speculate that even if the false fact is corrected, we still need to modify other contexts in the chat history (because they may contain old facts). By leveraging the NLI task (Bowman et al., 2015), we propose a KEIC algorithm to iteratively delete any text in previous chat history that contradicts new knowledge, as summarized in Algorithm 1 and proved in Appendix D. The notion involves *fact propagation* and *self-correction* (Kamoi et al., 2024), where we edit the chat history turn by turn in a top-down fashion.[5]

*Claim* 1. Algorithm 1 modifies the old chat history $h = [\mathbf{T_f}, \mathbf{T_o}]$ and returns a new chat history $h^* = [\mathbf{T_f^*}, \mathbf{T_o^*}]$ such that $h^*$ entails $\mathbf{T_c}$.

---

**Algorithm 1** KEIC

**Input**: KEIC instance $\mathcal{I} = \{\mathbf{T_f}, \mathbf{T_o}, \mathbf{T_c}\}$
**Output**: history $h^* = [\mathbf{T_f^*}, \mathbf{T_o^*}]$
1: Let $[\mathbf{T_f}, \mathbf{T_o}]$ be $[T_1, T_2, ...]$ and $\mathbf{T_c}$ be $T_c$
2: $h \leftarrow [\mathbf{T_f}, \mathbf{T_o}]$
3: Queue.push($\mathbf{T_c}$)
4: **while** Queue is not empty **do**
5:  $q \leftarrow$ Queue.pop()
6:  **for** $j \leftarrow 1, 2, ..., |h|$ **do**
7:   **if** INCONSISTENT($h[j]$, $q$) **then**
8:    $z \leftarrow$ DELETE($h[j]$, $q$)
9:    Queue.push($z$)
10:    $h[j] \leftarrow z$
11:   **end if**
12:  **end for**
13: **end while**
14: **return** $h$

---

[4]It is $P'_{new} = P_{old} \setminus R_{old} \cup R'_{new}$ in KEIC data. For example, in Figure 3c, an LLM should generate a new story in the Reiterate phase ($b_8$) as follows: CHAPTER XII "..., Miss Baldwin," Wingate invited, as he ushered that old lady into his rooms soon after ....

[5]In each iteration $j$, an external INCONSISTENT module detects if the current history $h[j]$ and the introduced knowledge $q$ are contradictory. If so, another external DELETE module will remove inconsistent context in $h[j]$ and generate a new coherent text $z$, then $z$ is considered a newly introduced knowledge (*i.e.*, $z$ is new to the current chat history $h$) and pushed into the Queue for later use, and $h[j]$ is updated by $z$. Otherwise, we skip this turn of conversation since they are either in neutral or entailment relation (see Line 7 in Algorithm 1). This chain reaction process repeats until the Queue is empty, meaning no text in the updated history $h^* = [\mathbf{T_f^*}, \mathbf{T_o^*}]$ contradicts $\mathbf{T_c}$.

## 4 EXPERIMENTS

### 4.1 DATASET COLLECTION

We first discard the CoQA data that does not have any YN questions. After setting the random seed to 0, we randomly select one YN question for the test phase. Once the test question is selected, the corresponding support sentence and previous QA pairs are determined. Hence, the KEIC framework is aligned with CoQA (see Section 2.4). The remaining task is to modify the (old) support sentence.

To ensure the new support sentences are "effective, fluent, and ethically sound," we collect them through Amazon Mechanical Turk (MTurk), an online crowdsourcing platform. Our task is only visible to workers from English-speaking countries with HIT approval rate $\geq 95\%$ and |HITs| $\geq 1,000$ (Karpinska et al., 2021). Each data is distributed to three workers, and we perform a meticulous examination of their results (see Appendix E for details): They must fill in the blank only—without altering or pasting the context near the blank—so we can replace the old fact with the new one while maintaining contextualized (if not global) fluency in the story. We pay each worker $0.1 or $0.15 in each assignment. Finally, our KEIC dataset consists of 1,317 data in training set ($\mathcal{D}_{train}$) and 464 in validation ($\mathcal{D}_{val}$). Each data has three non-trivial and effective corrections to the original CoQA (more examples are in Appendix E). The average number of turns in the other phase is 8.27 and 8.48, respectively. We denote $\mathcal{D}_{KEIC} = \mathcal{D}_{train} \cup \mathcal{D}_{val}$ ($|\mathcal{D}_{KEIC}| = 1,781$).

### 4.2 MODELS

We test four LLMs of varying sizes: GPT (OpenAI, 2022; 2023), Gemma (Team et al., 2024), Vicuna (Zheng et al., 2023b), and Llama (Touvron et al., 2023; Dubey et al., 2024). We set the `temperature` to 0 to maximize reproducibility (it is 1e-8 in Llama-3). Half precision is used in the Vicuna and Llama LLMs to match the Gemma LLM. We do not set the system message in the GPT models to further test their zero-shot KEIC capability. As for others, we use their default ones.

| Model | Configuration | |
|---|---|---|
| GPT-4o | gpt-4o-2024-08-06 | |
| GPT-4o (mini) | gpt-4o-mini-2024-07-18 | |
| GPT-4 | gpt-4-1106-preview | (2023) |
| GPT-3.5 | gpt-3.5-turbo-0613 | (2023) |
| | gpt-3.5-turbo-0125 | (2024) |
| Gemma-2 (27B) | gemma-2-27b-it | |
| Gemma-2 (9B) | gemma-2-9b-it | |
| Gemma-2 (2B) | gemma-2-2b-it | |
| Vicuna (33B) | vicuna-33b-v1.3 | |
| Vicuna (13B) | vicuna-13b-v1.5-16k | |
| Vicuna (7B) | vicuna-7b-v1.5-16k | |
| Llama-3 (8B) | Meta-Llama-3-8B-Instruct | |
| Llama-2 (13B) | Llama-2-13b-chat-hf | |
| Llama-2 (7B) | Llama-2-7b-chat-hf | |

### 4.3 SETUP AND EVALUATION METRIC

All the experiments are run three times to stabilize the performance. We utilize GPT-3.5 (0613) to implement the INCONSISTENT and DELETE in Algorithm 1 (see Appendix F for details). In Verification and Deletion, we apply an answer extraction (AE) step (Kojima et al., 2022) to guide the model in mapping its last response into Yes/No (see $u_{10}$ in Figure 3b for implementation). As for evaluation, we report the accuracy metric by using the exact match (Rajpurkar et al., 2016) in the first token of an LLM's output and the gold answer. "Update" means the LLM catches the user update and correctly answers the YN question in the last turn, and "No Update" means the LLM sticks to the (original) false knowledge. We conduct experiments on $\mathcal{D}_{KEIC}$ unless otherwise stated.

### 4.4 BASELINE

We have two baselines: One contains the update phase, and the other does not. In the latter case, we directly replace the old fact in the story with a new one, and the goal is to test the importance of the update phase within a dialogue since its conversation flow is devoid of the update phase.[6] In the former case, we conduct two settings (*i.e.*, *when* users correct themselves) in the OTC:

- **C**orrect **A**fter **M**istake (CAM): CAM simulates the user immediately corrects after making a false statement. It allows the correction to be contextualized to the misinformation, making it easier for the chatbot to update the stored knowledge in a conversation. However, the LLM may forget the update as the conversation progresses.

---

[6]It can also be viewed as a special case of Reiterate prompting: We extract the new story and initiate a new chat to test the LLM *without* the update phase.

- **C**orrect **B**efore **A**sking (CBA): CBA simulates the user corrects the false statement before asking the test question. This scenario benefits the chatbot because the update turn is provided in a more contextualized manner to the current turn. However, the chatbot has to pinpoint the misinformation in a chat history, or it may lose track of the location of the misinformation as the number of other turns grows. An example is in Figure 3a.

Table 1: The conversation flow of all KEIC methods in each setting. For example, as the Reiterate phase is defined to be applied immediately after the correction phase, the conversation flow of Reiterate with respect to the CAM and CBA setting is $\mathbf{T_f T_c T_r T_o T_i}$ and $\mathbf{T_f T_o T_c T_r T_i}$. We report the estimated input tokens required for GPT-3.5 (0613) on $\mathcal{D}_{val}$ as a reference (see Appendix G for more). AE stands for Answer Extraction. It is employed when many responses do not start with YN. In a complete experiment, we run each instance 90 times (15 correction utterances, three MTurk responses, and two settings). In our KEIC dataset, the story dominates the number of input tokens consumed. Since the data length varies, we do not report the number of API calls per data in the Deletion.

| Methodology | Setting (Arrangement and Injection) | | # Input Tokens ($\mathcal{D}_{val}$) | | # APIs | |
| | CAM | CBA | Total (M) | per Data | per Data | AE |
|---|---|---|---|---|---|---|
| OTC (baseline) | $\mathbf{T_f T_c T_o T_i}$ | $\mathbf{T_f T_o T_c T_i}$ | 21.5 | 516 (base) | 1 | ✗ |
| Verification | $\mathbf{T_f T_c T_o T_i T_v}$ | $\mathbf{T_f T_o T_c T_i T_v}$ | 70.5 | 1,687 (3.3x) | 3 | ✓ |
| Reiterate | $\mathbf{T_f T_c T_r T_o T_i}$ | $\mathbf{T_f T_o T_c T_r T_i}$ | 55.2 | 1,323 (2.6x) | 2 | ✗ |
| Deletion | N.A. (budget constraint) | $\mathbf{T_f T_o T_c T_r T_d T_i}$ | 204.9 | 147,225 (285x) | depends | ✓ |

## 4.5 PROPOSED METHODS

As for the other three KEIC methods, we adopt the experimental settings of CAM and CBA, as summarized in Table 1. In this way, we explore how different update approaches impact KEIC performance and investigate the relationships and consequences of phase arrangements.

We also experiment with the oracle performance of Reiterate by using string replacement to automatically generate the new story. Hence, the LLM does not need to generate a new story before answering the test question (# API calls is 1). Regarding the Deletion approach, since it is far more expensive, we only select a subset of the correction phase. In Deletion, we evaluate the test question by (1) incorporating the modified history and by (2) appending it to the Deletion phase (the conversation flow is in Table 1). The goal of evaluating the former approach aligns with that of our baseline with no update phase: testing whether the update phase is important.

# 5 RESULTS AND DISCUSSION

Figure 4 shows the result of GPT-3.5 (0613) on $\mathcal{D}_{val}$. We plot the OTC, Verification, and Reiterate results of all LLMs on $\mathcal{D}_{KEIC}$ in Figure 5 (top-$K$ majority voting (Wang et al., 2023)). More experiments are in Appendix H.

In the following section, since OpenAI periodically released their newest LLMs, we focus on a comprehensive analysis of the GPT model, using it as an example to systematically gauge the state-of-the-art LLM's result (the ablation analysis is in Appendix H). Because we extensively conduct experiments on two versions of GPT-3.5 in this paper, we explicitly include the version to avoid confusion. We also report the top-$K$ upper bound performance of multiple Chain-of-Thought (CoT, Wei et al., 2022) in Table 2.

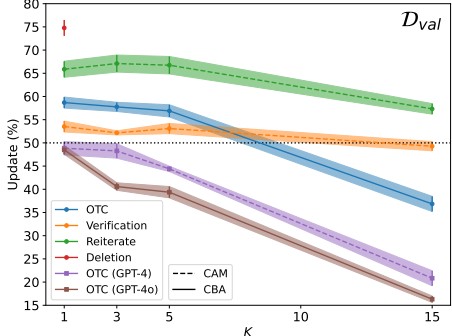

Figure 4: The best setting of each KEIC method in GPT-3.5 (0613) on $\mathcal{D}_{val}$. The x-axis is the top-$K$ correction templates in update ($|K| = 15$). GPT-4 performs poorly in OTC. In GPT-3.5 (0613), the baseline with no update phase has 56.5% of update (worse than the OTC by 2.2%). The average "random guess" update is 50%. Overall performance refers to the trend of top-1, 3, and 5 results.

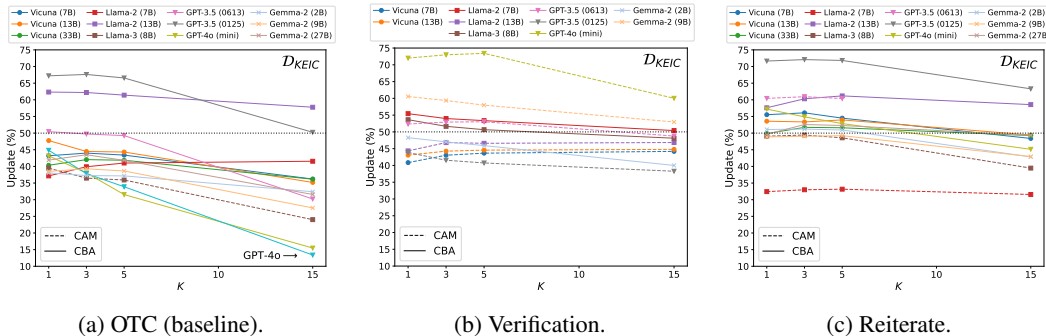

(a) OTC (baseline).

(b) Verification.

(c) Reiterate.

Figure 5: The best setting of all LLMs in each KEIC method on $\mathcal{D}_{KEIC}$. In Figure 5c, we plot the oracle of Reiterate in GPT-4o (mini), Vicuna (33B), and Gemma-2 (27B) due to the time constraint; however, we hypothesize that there should be no significant difference in Reiterate even if a new story is auto-generated in the Vicuna and Gemma LLMs (see Figure 11 in Appendix H for comparison). Due to the page limit, each LLM's KEIC results are in Figure 12 (in Appendix H).

Table 2: Percentage of Update/No Update/Upper Bound on $\mathcal{D}_{KEIC}$ using GPT-3.5 (0125). The standard deviations $s$ across three runs are shown in parentheses. We define the upper bound performance as follows: for example, to measure the top-5 upper bound in update, we first select the best five out of the 15 correction utterances. If *any* of these triggers an LLM to respond correctly based on the new fact, we consider that the LLM has KEIC capability in this KEIC instance. Verif stands for the Verification method. Maj stands for majority voting. $K$ means we select the Top-$K$ templates that perform best regarding the update. OTC is our baseline. The Verification method can be viewed as the CoT baseline (Kojima et al., 2022). Even if we apply an answer extraction turn, the output does not always start with a Yes/No (labeled as "N/A"), which also happens if there is a tie in majority voting. The sum of update and no update is not 100, as we exclude "N/A" in the table (due to the space, but it is almost always below 10%, except the top-1 of OTC in the CAM setting).

| Setting | $K$ | Update ($\uparrow$, Maj) | | | No Update ($\downarrow$, Maj) | | | Upper Bound ($\uparrow$) | | |
|---|---|---|---|---|---|---|---|---|---|---|
| | | OTC | Verif | Reiterate | OTC | Verif | Reiterate | OTC | Verif | Reiterate |
| CAM | 1 | $51.5_{(1.5)}$ | $\mathbf{43.9}_{(0.3)}$ | $64.6_{(1.0)}$ | $38.3_{(1.3)}$ | $\mathbf{55.5}_{(0.2)}$ | $27.7_{(1.1)}$ | $51.5_{(1.5)}$ | $43.9_{(0.3)}$ | $64.6_{(1.0)}$ |
| | 3 | $49.1_{(1.0)}$ | $41.6_{(0.5)}$ | $63.6_{(0.3)}$ | $44.1_{(1.1)}$ | $57.8_{(0.5)}$ | $30.7_{(0.6)}$ | $58.4_{(1.4)}$ | $61.7_{(0.8)}$ | $69.8_{(0.1)}$ |
| | 5 | $46.0_{(0.7)}$ | $40.7_{(0.4)}$ | $62.4_{(0.5)}$ | $48.2_{(0.8)}$ | $58.6_{(0.4)}$ | $32.6_{(0.5)}$ | $59.1_{(1.3)}$ | $68.2_{(0.4)}$ | $70.5_{(0.1)}$ |
| | 15 | $32.9_{(0.4)}$ | $38.3_{(0.5)}$ | $55.9_{(0.8)}$ | $62.5_{(0.3)}$ | $61.1_{(0.5)}$ | $40.4_{(1.0)}$ | $60.8_{(1.7)}$ | $80.7_{(0.4)}$ | $72.4_{(0.4)}$ |
| CBA | 1 | $67.2_{(0.3)}$ | $42.0_{(0.6)}$ | $71.7_{(0.9)}$ | $\mathbf{26.7}_{(0.1)}$ | $57.4_{(0.6)}$ | $\mathbf{22.9}_{(0.6)}$ | $67.2_{(0.3)}$ | $42.0_{(0.6)}$ | $71.7_{(0.9)}$ |
| | 3 | $\mathbf{67.6}_{(0.3)}$ | $41.0_{(0.6)}$ | $\mathbf{72.1}_{(0.9)}$ | $28.2_{(0.3)}$ | $58.4_{(0.6)}$ | $23.7_{(0.9)}$ | $74.4_{(0.2)}$ | $62.9_{(2.0)}$ | $76.9_{(0.7)}$ |
| | 5 | $66.6_{(0.1)}$ | $40.6_{(1.3)}$ | $71.8_{(1.0)}$ | $29.9_{(0.3)}$ | $58.8_{(1.3)}$ | $24.5_{(1.1)}$ | $76.5_{(0.1)}$ | $70.5_{(0.2)}$ | $78.9_{(1.1)}$ |
| | 15 | $50.3_{(0.8)}$ | $36.9_{(0.8)}$ | $63.3_{(1.1)}$ | $46.8_{(0.6)}$ | $62.5_{(0.8)}$ | $33.7_{(1.1)}$ | $\mathbf{77.9}_{(0.1)}$ | $\mathbf{83.3}_{(0.6)}$ | $\mathbf{80.5}_{(1.2)}$ |

**Transferability of correction phase across KEIC approaches** We first elaborate on our findings that different types of correction utterances significantly impact the performance (see Section 3). For instance, in GPT-3.5 (0613), we find that six templates, with only new knowledge to fill in, usually outperform the other nine in Verication, yet they significantly underperform in OTC and Reiterate. We speculate that the other nine templates contain the negation of old knowledge, so they may boost GPT-3.5's KEIC ability to update the answer in the OTC and Reiterate methods. In other words, these six templates perform poorly in OTC, suggesting GPT-3.5 does not pay attention to the correction phase if it only contains new knowledge. Consequently, after we re-question the model in Verification and tell it to reflect the update, GPT-3.5 may pay more attention to it and replies the updated answer. As for the other nine templates, we hypothesize that after re-questioning, the model is confused about which knowledge is correct, which means even if the GPT-3.5 response was indeed based on new information in the test phase, it may return to the old answer in the Verification phase, implying GPT-3.5 is not confident of its earlier answer. This observation also explains why there is a drastic drop in update between the performance of $K = 5$ and 15, as the other templates are poor at capturing the information update in different KEIC approaches (see Figure 5a). As for

GPT-3.5 (0125), the performance between different types of correction templates diminishes, for we found that templates with only new knowledge sometimes underperform the others in Verification. In this section, we refer to the overall performance when *top-1, 3, and 5* templates are selected.[7]

**GPT-3.5 exhibits a modicum of KEIC**   In Table 2, our OTC baseline demonstrates that when selecting the best or top-3 templates and making decisions through majority voting, GPT-3.5 (0125), on average, tends to edit the knowledge by more than 66% in the CBA setting and by around 50% in the CAM setting. Note that the CBA setting consistently outperforms CAM in OTC, indicating the model tends to give more importance to sentences that are in proximity to the current turn in the multi-turn scenario, which is similar to the recency effect in terms of memory recall from a psychological perspective. If we look at the best template, CBA surpasses CAM by 15.7%. Similarly, for $K = 3$ and 5, the CBA method continues to outperform CAM by around 18% to 20%. Unlike OTC, observe that the CAM setting slightly outperforms CBA in Verification;

however, its best result (43.9%) does not outperform the OTC baseline (67.6%) even if we apply an extra answer extraction turn. Although Verification is not as effective as it might be, its upper bound performance may be one of the most powerful, which is 83.3% in GPT-3.5. We also employ GPT-4 models to run the OTC baseline (see Figure 6); surprisingly, even with the aid of answer extraction (AE) in GPT-4 and GPT-4o, they are more "stubborn" and stick to the initial context provided by users or their underlying parametric memories. GPT-4 is generally recognized to be more intelligent and more discriminative to the input; nonetheless, we deduce it is also more susceptible to being misled by the fluctuating conditions and is vulnerable to inconsistent contexts in this scenario. We leave it as future work (McKenzie et al., 2023). In Figure 7, we plot all versions of GPT-3.5 in the OTC and display its improvement over time.

**Reiterate is better than OTC**   We find that prompting the LLM to reiterate new information has a significant improvement. Overall, GPT-3.5 (0125) has around 72% of update in the CBA setting. Furthermore, the best result of update in Reiterate outperforms the OTC by a large margin (13.1%) in CAM. Lastly, Reiterate has the smallest number of no update among these KEIC approaches. To delve into the data that GPT-3.5 does not update its knowledge, we employ GPT-3.5 (0613) to run our proposed KEIC algorithm. We choose the configurations in the best performance of update of Reiterate in the CBA setting, and then we extract data instances that GPT-3.5 (0613) consistently retains its old knowledge in $\mathcal{D}_{val}$. We construct the "hard" dataset as follows: Each data in the validation set contains three MTurk responses, and we run all of them three times using the top-3 correction utterances in the CBA setting. After that, we consider the data hard only if any run produces the same answer at least two times.

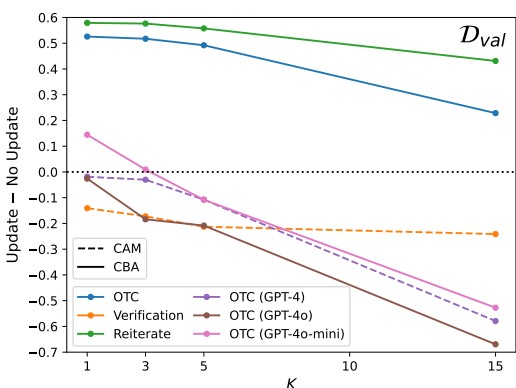

Figure 6: The difference between update and no update in GPT-3.5 (0125) on $\mathcal{D}_{val}$. Compared to GPT-3.5, GPT-4 LLMs significantly fail to capture the user update in the OTC baseline.

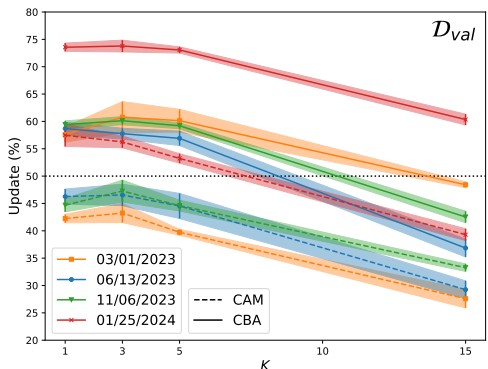

Figure 7: All versions of GPT-3.5 in the OTC on $\mathcal{D}_{val}$ (Chen et al., 2023). We conjecture that data similar to this work might have been added during training or that GPT-3.5 (0125) learned this task implicitly. Despite this, there is room for improvement in the CAM setting (a noticeable gap between CAM and CBA) and the other correction templates (a decline when $K = 15$).

---

[7]Llama-2 (7B and 13B) is rather robust for different types of templates. In Llama-3 (8B), however, it somehow behaves just like the others (see Figure 5a). It would be interesting to test their 70B LLMs.

**Deletion is one of the strongest KEIC methods** The empirical result of Algorithm 1 is tabulated in Table 3 and visualized in Figure 4. We deduce that it is not impossible to let GPT-3.5 (0613) "self-edit" its knowledge if previous information becomes outdated, and this method pushes the boundaries of in-context KE. GPT-3.5 could update its knowledge about 75% when the Deletion is employed, and it outperforms the Reiterate by 13.3% (see Table 7 in Appendix H). The update using only one template in Deletion also outnumbers the upper bound of 15 templates in the OTC, which is on par with that in Reiterate. Next, we explore whether Deletion can effectively delete old knowledge on the hard dataset. From the table, we observe that our algorithm can edit 51.9% of the "hard" data on average; nonetheless, this also indicates that GPT-3.5 still fails to edit nearly half of it. Although GPT-3.5 (0613) demonstrates its ability of self-correction, it comes at the expense of sacrificing around 15% "easy" data that Reiterate is capable of. On top of that, the cost is considerably high. We conclude the Deletion experiment by extracting the passage and all QA pairs when running the KEIC algorithm.

After we initiate a new chat, we find it has 66.2% of update and 33.3% of no update. Ideally, there should be no significant difference between these two; however, appending the test phase to the Deletion phase performs much better (8.6%) than initiating a new chat—higher than the difference between the OTC baselines (2.2%). We conjecture that repeated instructions boost GPT-3.5's KEIC.

Table 3: The result of Deletion (Algorithm 1) on $\mathcal{D}_{val}$. Standard deviations are in parentheses.

| Data | # data | Update ($\uparrow$) | No Update ($\downarrow$) |
|------|--------|---------|-----------|
| Validation | 464 | 74.8 (1.7) | 24.5 (1.8) |
| – Hard | 144 | 51.9 (2.2) | 47.7 (2.6) |
| – Easy | 320 | 85.1 (2.1) | 14.1 (2.3) |

## 6 CONCLUSION

As discrepancies arise in dialogue, either from users to correct themselves or from LLMs to start hallucinating, the capability of LLMs to accurately and efficiently update information on the fly is an essential yet underexplored issue. Inspired by this, we formalize it as the KEIC task and present the multi-turn framework to decompose the chat history. Then, we propose a structured methodology to systematically gauge the LLMs' KEIC ability in a zero-shot setting. Distinguished from existing datasets, we release a challenging dataset for LLMs to recognize the misinformation mid-paragraph in long-term conversations. Extensive studies have shown, in the main, that the correction phase containing the negation of the false fact performs better, the update phase is indispensable, its location also affects the result in each approach, Reiterate is an economical approach, KEIC algorithm can update nearly 75% of fact within a paragraph in extended conversations, and the KEIC task does not disappear with time and the scale of LLMs. Our framework and dataset form the foundation for constructing chatbots that are not only coherent but adaptive for long-term companionship. The limitation of this work, including the key takeaways of our framework, is in the Appendix. The code and dataset will be made publicly available; we also include them in the Supplementary Material.

**Ethics Statement** Any LLM shall not be treated as an authoritative source of facts, even though we test LLMs' adaptability and use their outputs as a knowledge base. It is important to note that our work could be potentially exploited by malicious users to produce harmful responses; hence, it should not be used in any harmful way. Our KEIC dataset is constructed based on the CoQA (and should follow its license), and the correction templates are excerpted from the DailyDialog dataset. On the other hand, the new support sentences are generated by MTurk workers and validated by us. We provide them with ethics statements (see Figure 10 in Appendix E) and manually filter out unsafe or unethical responses while preserving effectiveness. Nevertheless, as our primary goal is to modify existing knowledge, some results might still be offensive or inappropriate for some people. Our framework can be used for training. To avoid data contamination, however, the update sentences generated by workers should be used solely for inference unless a publicly available technical report or manuscript explicitly mentions they are used for training to ensure fairness in LLM evaluations.

**Reproducibility Statement** Appendix A is the related work, Appendix B lists 15 correction templates, Appendix C visualizes the Deletion approach, Appendix D contains the proof of our KEIC algorithm, Appendix E details how we validate MTurk responses and how hard our non-trivial information update is, Appendix F provides the exact prompt to implement two modules in our KEIC algorithm, Appendix G gives more time/cost estimations, and Appendix H has more experiments.

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

## LIMITATIONS

**Practicality and Key Takeaways**   In this paper, we present the ultimate goal for intelligent LLMs in the KEIC task: A single update sentence (*i.e.*, OTC) should effectively edit the LLM's in-context knowledge, mimicking human behavior. Considering real-time response requirements and the cost of token usage, incorporating an additional phase for LLMs to reiterate the updated fact through Reiterate is beneficial. Ideally, there should be no significant difference in *how* or *when* users correct themselves. Nevertheless, our findings reveal that clearly negating the false facts is far more effective than simply stating the updated information. Additionally, our results highlight a noticeable gap between CAM and CBA settings. Given that these contemporary LLMs have not fully excelled in the KEIC task, it would be advantageous to dispatch each component of our framework to specialized or more robust LLM-based system(s) for now. In this work, we leverage the invaluable, human-annotated CoQA dataset to assess whether LLMs can capture user updates within long utterances and extended conversations. Real-world data, however, lacks proper labels. While our KEIC algorithm can still be applied by repetitively scanning the entire chat to overwrite contradictions, it risks deleting other important information. Hence, before LLMs are trained with KEIC, it may be beneficial to maintain a classifier detecting whether a user is updating knowledge, along with one or more systems capable of handling the "Decomposition" and "Arrangement and Injection" processes in the background.

**KEIC Dataset**   Our dataset is limited to YN questions and does not cover various open-domain questions. However, as we take a step forward to construct our dataset in this task—which can be viewed as the zero-shot KE task in chat format—we speculated it would be much easier to edit the misinformation within a short utterance.[8] Thus, our goal is to find an existing dataset where a false fact lies within a long context. Hence, we select CoQA. After that, we resort to simple YN questions and try to keep our evaluation method noise-free so as not to increase the interference. Another direction for future work is to expand our work and test other open-domain questions in the CoQA.

**KEIC Framework**   Our framework is designed for multi-turn chat format, so it may require "filling" or "padding" in some datasets during the mapping process, in the sense that they are not so "natural." For example, the bot utterances in the false and update phase are not in the original CoQA data (*e.g.*, $b_1$ and $b_7$ in Figure 3a), nor they are all inherently learned or generated by LLMs. We pre-fined these texts in this paper as they can be used for evaluating the current KEIC capabilities of LLMs uniformly—though, admittedly, all human-generated prompts are not optimal in this sense—and save the API calls. To assess whether they play an important role in this task, we additionally conduct the ablation analysis by removing these texts in the OTC (see Table 5 in Appendix H). Another direction for future work is to propose new approaches to extend the update phase ($\mathbf{T_u}$) and explore various combinations of existing in-context KE methods.

**Experiments**   This paper is an in-depth study of the KEIC task, yet the experiments do not cover other open-domain LLMs. Consequently, constantly testing whether they are on par with GPT-3.5 is also a promising avenue of research. Regarding correction template generation, while we employ the mining approach, we have not conducted an exhaustive evaluation of possible text combinations within these templates (they are included in Appendix B.3). When evaluating our KEIC methodologies, we presume that specific processes are error-free without confirming whether all these processes fulfill our intended requirements. As a result, it is also worthwhile to conduct in-depth analyses of Reiterate (*e.g.*, how successful LLMs are in reiterating the story) and Deletion (*e.g.*, the two modules and extraction templates used in our KEIC algorithm). Similar to the oracle of Reiterate, it is also worth experimenting with the oracle of Verification. In the Deletion method, there are opportunities to investigate several approaches for condensing excessively long text that exceeds the conversation limit. Various operations of DELETE, including masking the old information, have not been implemented. Owing to the cost, we have not tested whether the Deletion method can substantially boost the performance of other "poor" templates with only one slot for new knowledge. Other limitations (such as modifying multiple facts simultaneously or evaluating open-ended questions) are beyond the scope of this research, and we leave them for future work.

---

[8]LLMs may fail at either locating the false utterance within a long story or overwriting it with the updated fact. Incidentally, our ablation analysis (without FP in Table 5) tests this scenario by removing the context after the support sentence. We find that the percentage of update increases when the passage is abridged.

## A    RELATED WORK

On top of adaptability, consistency has long been considered an ongoing and formidable challenge in the domain of chatbot development (Vinyals & Le, 2015; Li et al., 2016; Zhang et al., 2018), and a plethora of training methods has been put forward in an attempt to bolster the coherence of chatbot responses (Yi et al., 2019; Li et al., 2020; Bao et al., 2021; Ouyang et al., 2022; Rafailov et al., 2023; Ethayarajh et al., 2024, *inter alia*). To gauge the aptitude of a chatbot in maintaining consistency, existing benchmarks that focus on contradiction detection have been employed (Welleck et al., 2019; Nie et al., 2021; Zheng et al., 2022). These dialogue benchmarks, on the whole, categorize contradictory responses by chatbots as erroneous, and a common thread amongst most of them is the objective to deter chatbots from generating responses that conflict with their previous statements. Nevertheless, an often overlooked aspect of these benchmarks is the dynamism of natural conversations—they do not consider the information in earlier chat may have been rendered obsolete by the user. In such cases, to align with the user's updated knowledge, we highlight that *the chatbot sometimes even needs to contradict its previous in-context response to ensure the conversation remains accurate and coherent* (see Figure 1). We hypothesize that these conversational datasets, although aiming to improve an LLM's consistency and reduce self-contradiction is of paramount importance, may hamper its adaptability—an emerging issue of contemporary LLMs. In light of this, balancing between the two seemingly paradoxical yet highly correlated tasks during training would be one of the key challenges and opportunities for future work.

In previous work, knowledge editing (KE) typically involved proposing an efficient methodology to modify the parameters of an LLM (De Cao et al., 2021; Mitchell et al., 2022a; Meng et al., 2023). Efficient as they may be, these approaches are vulnerable to overfitting, where the edited LLMs do not generalize well on other inputs or tasks (Cohen et al., 2024). Concurrently, there has been a surge in exploiting additional system(s) and keeping the LLM unchanged (Mitchell et al., 2022b; Murty et al., 2022). To this end, their frameworks generally can be broken down into three components: a memory storage system that acts as a new knowledge base, a scope classifier that determines whether the input sequence is relevant to the external memory, and a counterfactual model trained on new knowledge. In parallel, there exist approaches that utilize external sources or specialized LLMs to aid or calibrate model predictions (Pan et al., 2019; Yao et al., 2023; Feng et al., 2024; Gou et al., 2024, *inter alia*). In sum, these methods require either parameter modification or additional systems; they often struggle with the rapid change of information or are incompatible with online conversations (Kamoi et al., 2024; Miao et al., 2024). Each fact in the previous KE datasets is usually a short sentence (De Cao et al., 2021; Meng et al., 2022; Lin et al., 2022), focusing on querying a specific real-world knowledge. On the other hand, the DIALFACT dataset aims to improve fact-checking performance in chat format (Gupta et al., 2022), yet the dataset is not suitable for assessing an LLM's long-term adaptability. Regarding the QA datasets for benchmarking an LLM's self-correction capability, there are HotpotQA (Yang et al., 2018), CommonsenseQA (Talmor et al., 2019) and STRATEGYQA (Geva et al., 2021), to name a few. However, *these datasets do not simulate human interactions in long-term dialogue either*. To address this gap, we design the KEIC framework and create our dataset based on the CoQA (Reddy et al., 2019) in this standard, which applies to both conversational (long and short) and non-conversational (*e.g.*, math and coding) datasets.[9] Our framework serves as a stepping stone for standardizing dataset construction in this task and could facilitate the evaluation of future LLMs across different domains, particularly in aligning user updates or addressing hallucination, the long-standing problem.

## B    ALL CORRECTION TEMPLATES USED AND KEYWORDS FOR MINING APPROACH

We first define 15 keywords (Appendix B.1) to extract the sentences from the DailyDialog dataset, then we modify it and generate 15 templates (Appendix B.2) in our experiments. In Appendix B.3,

---

[9]Take a simple math problem as an example for non-conversational data. A user initially asked an LLM to evaluate the math question "2 + 3 = ?". After it responds with "5" (in the false phase), the user can say "Wrong. It's not 2, but 4" in the update phase (the entity value "2" is replaced by an effective knowledge update "4"), and then ask the LLM what the final answer is in the test phase (in this example, an LLM could also directly correct its answer to 7 within the update phase). Concerning the Reiterate approach, we can ask the LLM what the new math question is in the subsequent turn, where an LLM should respond "4 + 3 = ?".

we release all the mined sentences pertaining to human correction utterances in DailyDialog, which researchers may find useful as it alleviates the need for prompt engineering.

### B.1 FIFTEEN KEYWORDS FOR MINING DAILYDIALOG

["sorry", "error", "actual", "correct", "mistake", "mistook", "oops", "apologize", "misunder", "underst", "clarif", "should be", "forget about", "forgot about", "wrong"]

### B.2 FIFTEEN CORRECTION TEMPLATES

In the following templates, `[O]` is replaced by old knowledge, and `[N]` is the slot for new knowledge. The correction phase ($T_c$) contains 15 diverse templates that fall into two categories (*i.e.*, *how* users correct themselves): Nine templates (index 1 to 9) require the incorporation of both old and new knowledge for completion, while the remaining six templates (index 10 to 15) feature a single slot for new knowledge.

1. I think "`[O]`" is wrong, so it should be "`[N]`"
2. Oh, I'm sorry. Should have been "`[N]`," not "`[O]`"
3. Something is wrong with the story. You can correct it by replacing "`[O]`" with "`[N]`"
4. There's a problem with the story. There's a mistake on "`[O]`." It should be "`[N]`"
5. I wouldn't say that. "`[O]`" seems to be correct but actually "`[N]`"
6. Wrong. It's not "`[O]`," but "`[N]`"
7. No, "`[O]`" sounds wrong. "`[N]`"
8. I'm sorry to bring this up, but I mistakenly gave you "`[O]`." In fact, "`[N]`"
9. Change "`[O]`" to "`[N]`" That was the only thing that I saw that was wrong in the story.
10. Actually, "`[N]`"
11. It's "`[N]`." Sorry. I forgot that the story has been updated.
12. Believe it or not, the truth is the opposite. "`[N]`"
13. I think there might be an error in the story. I think that "`[N]`"
14. I think I must have heard wrong. The truth is "`[N]`"
15. Oh, my mistake. "`[N]`" I'm sorry for the error.

### B.3 SENTENCES MINED FROM DAILYDIALOG

This section contains the prototype of our 15 correction templates used in the correction phase.

#### B.3.1 TRAINING SET

- Sam, I am so sorry. It was your birthday yesterday and I completely forgot about it.
- Maybe you can correct it by going to a driving range before you play again.
- There's problem with my bank statement. There's a mistake on it.
- I wouldn't say that. They seem to be on good terms but actually they always speak ill of each other.
- Wrong. It's not a place name, but a passionate act.
- No, it sounds wrong. He was born in the 16th century.
- I'm sorry, I didn't mean to forget our wedding anniversary.
- I thought she was going to call when she was done shopping. It was a misunderstanding. She was literally screaming on the phone over this.
- Excuse me, Professor. I think there might be an error in my test score. I think that the percentage is incorrect.
- I think you must have heard wrong. The truth is we are going to be taken over by Trusten.

- Oh, I'm sorry. It completely slipped my mind.

- Well, Yes. There are something wrong actually. Perhaps you can give me some advice.

- It looks like some kind of mistake.

- I think there's been a misunderstanding!

- Thank you for pointing that out. I mistakenly gave you your friend's breakfast.

- Oh, I am sorry sir. I forgot to explain that to you. This one is an allowance slip. We made a mistake in your bill and overcharged you 120 dollars.

- Oh, my mistake. The reservation is for a suite and it is a non-smoking room with a king bed. I'm sorry for the error.

- I'm afraid there has been a mistake.

- Oh. I made a mistake. I thought the guy on the right was Peckham.

- I apologize. This should not have to be this way.

### B.3.2 VALIDATION SET

- Believe it or not, it has the opposite effect. Employees are actually more productive on casual days.

- Excuse me. Something is wrong with my bank card. Can you help me?

- Oops, no, Daddy can't watch American Idol, either!

- That was the only thing that I saw that was wrong with the apartment.

- Oh, I'm sorry. should have been 2135-3668, not 3678. I've given you a wrong number.

- One moment, please. I have to check if there are rooms available. I'm sorry, ladies. We have only two double rooms available but they are on different floors. Would you mind that?

- I'm embarrassed! I forgot completely about them. I'm terribly sorry.

- I'm sorry. Something is wrong with my taxi.

### B.3.3 TEST SET

- I think it's a distance of 180 kilometers from here to London, so it should be a two-hour drive on the motorway.

- I'm afraid there's been a mistake.

- Actually, fruits and veggies are really good for you.

- I'm sorry to bring this up, but would it be possible for you to write me a letter of recommendation before you go?

- Sorry, I forgot. I don't like seafood, neither.

- Oops, cancel that. Change the second call to 7 thirty will you, please?

- Actually, the company will provide you with all of these supplies.

- Well, actually two-thirds of Americans may avoid these places.

- It's traditional Chinese Medicine. I mix it with hot water like tea. Sorry. I forgot about it.

- I completely forgot about your cat allergy. I took care of a cat for my friend here a few days ago.

## C  THE EXACT PROMPT FOR THE DELETION METHOD

The Deletion method is visualized in Figure 8, which follows the same convention as Figure 3.

| **T_f** |
|---|
| $u_1$: *Read and memorize the following story.* |
| *Story:* CHAPTER XII "Throw your coat down anywhere, Miss Baldwin," Wingate invited, as he ushered that young lady into his rooms soon after eleven o'clock on the following evening... |
| ========== |
| *Have you memorized the story?* |
| $b_1$: *Yes, I have memorized the story.* |

| **T_o** |
|---|
| $u_2$: Who did Wingate talk to? |
| $b_2$: Miss Baldwin |
| $u_3$: What is her first name? |
| $b_3$: Sarah |
| $\vdots$ |
| $u_6$: On the same evening? |
| $b_6$: No |

| **T_c** |
|---|
| $u_7$: *Actually, "he ushered that old lady into his rooms"* |
| $b_7$: *No problem at all! I have updated my memory of the story with the correction you provided. Thank you for letting me know.* |

| **T_r** |
|---|
| $u_8$: What's the new story with the correction? Output new story and nothing else. |
| $b_8$: [Chat Completion] |

| **T_d** |
|---|
| $u_9$: **Story = """[Story Completion]""" Correction = """[Correction Completion]""" Which parts in the story contradict the correction? If the story entails the correction, output 'NO MODIFICATION'. Let's read the story line by line. List all the contradictions one by one, if any.** |
| $b_9$: [Chat Completion] |
| $u_{10}$: **Can you modify the story, one by one, so that the correction entails the story?** |
| $b_{10}$: [Chat Completion] |
| $u_{11}$: **QA pair = """ [QA Completion]""" Correction = """[Correction Completion]""" Does the QA pair contradict the correction? If the QA pair entails the correction, output 'NO MODIFICATION'. If the QA pair contradicts the correction, explain why they are contradictory in one sentence. If they are in a neutral relation, output 'NO MODIFICATION'. Let's think step by step.** |
| $b_{11}$: [Chat Completion] |
| $u_{12}$: **Can you modify the QA pair so that it entails the correction? DO NOT modify the QA pair by copying the correction. Let's think step by step.** |
| $b_{12}$: [Chat Completion] |
| $\vdots$ |
| (until IC-MRE Algorithm terminates) |

| **T_i** |
|---|
| $u_i$: Is Sarah old? |
| $b_i$: [Chat Completion] |

Figure 8: Deletion ($\mathbf{T_u} = \{\mathbf{T_c}, \mathbf{T_r}, \mathbf{T_d}\}$).

# D    CORRECTNESS OF KEIC ALGORITHM

Before we start the proof, we state the following three main objectives (proof sketch):

1. The KEIC algorithm will fix the inconsistent context (Lemma 1).

2. For each edit, the consistency still holds within each turn and the entire conversation history (Lemma 2).

3. The KEIC algorithm will halt (Lemma 3).

In this paragraph, we further elaborate on the initiative of our Deletion approach. In Section 3, recall that we mention "even if the false text is corrected, we still need to modify other contexts in the chat history."

In other words, granted those approaches are effective, we may rely heavily on the following condition: The fact is solely within the support sentence in the story, and no other context that excludes it can answer the question correctly. We formally define it as follows:

$$\forall C \in P \setminus R \ \text{ s.t. } \ A^\dagger \in (C, Q, A^\dagger) \text{ and } A^\dagger \neq A \tag{4}$$

In reality, it is not always true. That is,

$$\exists C \in P \setminus R \ \text{ s.t. } \ A^\dagger \in (C, Q, A^\dagger) \text{ and } A^\dagger = A \tag{5}$$

To prove our KEIC algorithm summarized in Algorithm 1 is correct, we shall begin by introducing the notations employed within this Appendix.

**Notation 1.** Let $x$, $y$, $z$ be the text string. $|x|$ denotes the number of of words in $x$. Let $\mathcal{S}(x) = \{\mathcal{M}(x') : x' \in x\}$ be the set of subject-object relation triplets of $x$. Let the history $h = [\mathbf{T_f}, \mathbf{T_o}] = [T_1, T_2, ..., T_m]$ be the $m$-turn conversation (where $m \geq 1$), and $\mathbf{T_c} = T_c$ is the correction turn that contains (initial) effective knowledge $(R_i', Q_i, A_i')$. Define the text space $\mathcal{C} = \{P\} \cup \{(Q_l, A_l) : l \in [1, i-1]\}$, $\mathcal{C}_{R_i} = \{C : C \in \mathcal{C} \wedge A^\dagger \in (C, Q_i, A^\dagger) \wedge A^\dagger = A_i\}$, and $\mathcal{C}_{\neg R_i} = \mathcal{C} \setminus \mathcal{C}_{R_i}$. For readability, we omit the subscript of $R_i$, $Q_i$, and $A_i$. Note that $\mathcal{C}_R \subset \mathcal{C}$ and $\mathcal{C} = h$.[10]

The definition of $\mathcal{C}_R$ may seem daunting, but it simply conveys that it is the text space containing all the text strings related to the old knowledge in the passage and previous QA pairs. Likewise, $\mathcal{C}_{\neg R}$ is the text space where any text is *unrelated* to the old knowledge.

**Definition 1.** *Let $\mathcal{R}_\times$ be the contradiction relation. Define*

$$\mathcal{R}_\times(x, y) = \begin{cases} 1 & \text{iff } y \text{ contradicts } x \\ 0 & \text{otherwise} \end{cases}$$

**Proposition 1** (symmetric of $\mathcal{R}_\times$)**.** *Let $p_1$, $p_2$ be the text. $\mathcal{R}_\times(p_1, p_2) = \mathcal{R}_\times(p_2, p_1)$.*

**Proposition 2.** *If $\mathcal{R}_\times(y, x) = 0$ and $\mathcal{R}_\times(z, x) = 0$, then $\mathcal{R}_\times(y \cup z, x) = 0$.*

**Proposition 3.** *If $\mathcal{R}_\times(z, x) = 0$ and $\mathcal{R}_\times(z, y) = 0$, then $\mathcal{R}_\times(z, x \cup y) = 0$.*

**Example 1.** $\forall x \in \mathcal{C}_R, \mathcal{R}_\times(x, R') = 1$.

**Example 2.** $\forall x \in \mathcal{C}_{\neg R}, \mathcal{R}_\times(x, R') = 0$.

**Definition 2.** *Let $\mathcal{R}_\circ$ be the entailment relation. Define*

$$\mathcal{R}_\circ(x, y) = \begin{cases} 1 & \text{iff } y \text{ entails } x \\ 0 & \text{otherwise} \end{cases}$$

**Proposition 4** (transitive of $\mathcal{R}_\circ$)**.** *Let $p_1$, $p_2$, $p_3$ be the text. If $\mathcal{R}_\circ(p_2, p_1) = 1$ and $\mathcal{R}_\circ(p_3, p_2) = 1$, then $\mathcal{R}_\circ(p_3, p_1) = 1$.*

**Proposition 5.** *If $\mathcal{R}_\circ(y, x) = 1$ and $\mathcal{R}_\times(z, x) = 0$, then $\mathcal{R}_\circ(y \cup z, x) = 1$.*

**Proposition 6.** *If $\mathcal{R}_\circ(z, x) = 1$ and $\mathcal{R}_\times(z, y) = 0$, then $\mathcal{R}_\circ(z, x \cup y) = 1$.*

**Corollary 1.** *Given $n$ is finite and $p_i$ is the text $\forall i \in [1, n]$. If $\mathcal{R}_\circ(p_{i+1}, p_i) = 1 \; \forall i \in [1, n-1]$, then $\mathcal{R}_\circ(p_n, p_1) = 1$.*

**Corollary 2.** *If $\mathcal{R}_\circ(x, y) = 1$, then $\mathcal{R}_\times(y, x) = 0$.*

*Proof.* Assume $\mathcal{R}_\times(y, x) = 1$ is true, then $\mathcal{R}_\times(x, y) = 1$ by Proposition 1, which contradicts our assumption that $\mathcal{R}_\circ(x, y) = 1$. $\qquad\square$

**Corollary 3.** *Given $p_1, ..., p_n$ and $\mathcal{R}_\circ(p_{i+1}, p_i) = 1 \; \forall i \in [1, n-1]$. $\forall i, j \in [1, n]$, if $\mathcal{R}_\circ(p_j, p_i) = 1$, then $\mathcal{R}_\times(p_i, p_j) = 0$.*

**Definition 3.** *Let $\delta$ be the delete function, $\delta(x, y) = \{z : z = x \setminus c \cup c' \wedge c \in x \cap \mathcal{C}_R \wedge \mathcal{R}_\circ(c', y) = 1\}$, and $\delta_{min}(x, y) = \{z : z \in \delta(x, y) \wedge \mathcal{M}(c') \in \Delta(c) \wedge |\mathcal{S}(c')| = |\mathcal{S}(c)|\}$.*

**Definition 4.** *The set $\mathcal{Z}_\circ(x, y) = \{z' : z' = \delta_{min}(x, y) \wedge \mathcal{R}_\circ(z', y) = 1\}$.*

**Corollary 4.** *If $z \in \mathcal{Z}_\circ(x, y)$, then $z \in \delta_{min}(x, y)$.*

The KEIC algorithm requires the following three assumptions:

**Assumption 1.** INCONSISTENT *module is perfect. That is, $\forall x$ and $y$,* INCONSISTENT$(x, y) = \mathcal{R}_\times(x, y)$.

**Assumption 2.** DELETE *module is perfect. That is, $\forall x$ and $y$,* DELETE$(x, y) = \delta_{min}(x, y)$ *and $z \in \mathcal{Z}_\circ(x, y)$.*

**Assumption 3.** *$h$ is finite and consistent. That is, $m$ is finite, $|T_i| = |u_i| + |b_i|$ is finite, and $\mathcal{R}_\times(T_j, T_i) = 0 \; \forall i, j \in [1, m]$.*

---

[10]Strictly speaking, $\mathcal{C} \subset h$ since some texts are pre-defined, such as the bot response in the false phase (see the texts in italics in Figure 3a). Nonetheless, as they should not affect the proofs (irrelevant), we treat them as equal for simplicity.

In practice, we do not know (and cannot access) the answer $A$; however, as we already define the new knowledge $R'$ is *effective* and $\mathcal{Y} = \{\text{Yes, No}\}$ in Section 2, we have the following corollary:

**Corollary 5.** $\forall (R, Q, A)$ and $(R', Q, A')$, if $A^\dagger = A'$ in Eq. 4, then $A^\dagger \neq A$.

Therefore, if we are able to detect all contexts $C \in \mathcal{C}_R$ and effectively edit all of them such that $R'$ entails $C$ (*i.e.*, $\mathcal{R}_\circ(C, R') = 1$), then any obsolete knowledge $(R, Q, A)$ in $\mathcal{C}_R$ is deleted:

$$\nexists C \in \mathcal{C}_R \text{ s.t. } A^\dagger \in (C, Q, A^\dagger) \text{ and } A^\dagger = A \tag{6}$$

In Corollary 5, we know if $A^\dagger = A$, then $A^\dagger \neq A'$, and thus Eq. 6 can be rewritten as (after DELETE):

$$\forall C \in \mathcal{C}_R \text{ s.t. } A^\dagger \in (C, Q, A^\dagger) \text{ and } A^\dagger = A' \tag{7}$$

Compared to Eq. 4, observe that we do not access $A$, and since $A'$ lies in the text $R'$, Eq. 7 aligns with our objective.

**Lemma 1.** *For every iteration $j$, $\mathcal{R}_\circ(z, q) = 1$.*

*Proof.* The initial knowledge in $q$ is $T_c$ that contains $R'$, and the delete function $\delta_{\min}$ will replace $R$ with $R'$ by Definition 3. We only need to consider the case $\mathcal{R}_\times(h[j], q) = 1$, which means $\exists C \in h[j] \cap \mathcal{C}_R$, and the perfect INCONSISTENT module detects the contradiction between $h[j]$ and $q$ by Assumption 1. Suppose Assumption 2 is true, we have $z \in \mathcal{Z}_\circ(h[j], q)$, and $z = \delta_{\min}(h[j], q)$ by Corollary 4. Thus, $z = \text{DELETE}(h[j], q)$. Since $z \in \mathcal{Z}_\circ(h[j], q)$, we have $\mathcal{R}_\circ(z, q) = 1$. $\qquad\square$

As proving the Queue preserves transitivity of entailment in Algorithm 1 is more complicated, we will prove it later in Lemma 4 and use the following claim first.

*Claim* 2. For every $q_i$ and $q_j$ in Queue ($i < j$), $\mathcal{R}_\circ(q_j, q_i) = 1$.

**Lemma 2.** *If the KEIC algorithm terminates and returns history $h^*$, then $\forall T^* \in h^*, \mathcal{R}_\times(T^*, T_c) = 0$.*

*Proof.* WLOG, let $h^* = [T_1^*, T_2^*, ..., T_m^*]$, $T^* = T_k^*$ be one of the turns in $h^*$ ($k \in [1, m]$), and $q$ be the last element in the Queue so that no element is pushed into the Queue and the algorithm returns $h^*$. Define $\mathcal{C}_{\neg R \cap T^*} = \{y : y \in \mathcal{C}_{\neg R} \cap T^*\}$, which means no text is modified in $\mathcal{C}_{\neg R \cap T^*}$, and we define $\mathcal{C}_{R \cap T^*} = T^* \setminus \mathcal{C}_{\neg R \cap T^*}$. Since $\mathcal{R}_\times(y, T_c) = 0 \; \forall y \in \mathcal{C}_{\neg R \cap T^*}$, we only need to consider the text in $\mathcal{C}_{R \cap T^*}$. By Lemma 1, we know $\forall x \in \mathcal{C}_{R \cap T^*}, \mathcal{R}_\circ(x, q) = 1$, and we have $\mathcal{R}_\circ(q, T_c) = 1$ by Corollary 1 and Claim 2. Thus, $\mathcal{R}_\circ(x, T_c) = 1$ by Proposition 4. Finally, we have $\mathcal{R}_\times(T_k^*, T_c) = \mathcal{R}_\times(\mathcal{C}_{R \cap T_k^*} \cup \mathcal{C}_{\neg R \cap T_k^*}, T_c) = 0$ by Proposition 2, which holds for any $k \in [1, m]$. Therefore, $\forall T^* \in h^*, \mathcal{R}_\times(T^*, T_c) = 0$. $\qquad\square$

**Corollary 6.** $T_c$ entails $h^*$.

**Lemma 3.** *The KEIC algorithm will terminate.*

*Proof.* As the DELETE module is perfect, any text that is being modified will not need to be modified again by Corollary 3, which means $|\mathcal{C}_R|$ is decreasing. Since the history $h$ is finite in Assumption 3, the algorithm will terminate. $\qquad\square$

To prove Claim 2, we define the notations used in the Definition 5 and 6.

**Notation 2.** Let $X, Y$ be the text, $X = x_1 \cup x_2$ and $Y = y_1 \cup y_2$, where $x_1 \cap x_2 = \emptyset$ and $y_1 \cap y_2 = \emptyset$. Recall that $\tau_X \in \mathcal{M}(X)$ is the subject-object relation triplet of $X$.

**Definition 5.** *If $\mathcal{R}_\times(y_1, x_1) = 0 \wedge \mathcal{R}_\times(y_2, x_1) = 0 \wedge \mathcal{R}_\times(y_1, x_2) = 0 \wedge \mathcal{R}_\circ(y_2, x_2) = 1 \Rightarrow \mathcal{R}_\circ(Y, X) = 1$.*

*Proof.* Since $\mathcal{R}_\times(y_1, x_1) = 0$ and $\mathcal{R}_\times(y_2, x_1) = 0$, we have $\mathcal{R}_\times(Y, x_1) = 0$ by Proposition 2. Similarly, $\mathcal{R}_\times(y_1, x_2) = 0$ and $\mathcal{R}_\circ(y_2, x_2) = 1$, we have $\mathcal{R}_\circ(Y, x_2) = 1$ by Proposition 5. Finally, by Proposition 6 we have $\mathcal{R}_\circ(Y, x_1 \cup x_2) = 1 \Rightarrow \mathcal{R}_\circ(Y, X) = 1$. $\qquad\square$

While Definition 5 offers a method for identifying whether text $X$ entails another text $Y$ through a process of decomposition, multiple comparisons between segments of both texts are necessary, which we cannot overlook. For example, if $X = (x_1 = $ *Mary feels bored*, $x_2 = $ *She adopts a cat*$)$ and $Y = (y_1 = $ *Mary adopts a dog instead of a cat*, $y_2 = $ *She becomes responsible for taking care of the pet*$)$, we have $\mathcal{R}_\circ(y_2, x_2) = 1$, but $\mathcal{R}_\times(y_1, x_2) = 1$. To eliminate this issue, we first define the mapping function $\mathcal{F}_1$ and $\mathcal{F}_2$ as follows:

$$\mathcal{F}_1 : X \to \left\{ x_i : \bigcup_i \mathcal{S}(x_i) = \mathcal{S}(X) \wedge \mathcal{S}(x_i) \cap \mathcal{S}(x_j) = \emptyset \; \forall i \neq j \right\} \tag{8}$$

$$\mathcal{F}_2 : (X, Y) \to \left\{ (x_i, y_i) : x_i \in \mathcal{F}_1(X) \wedge y_i \in \mathcal{F}_1(Y) \wedge \mathcal{R}_\times(y_j, x_i) = 0 \; \forall i \neq j \right\} \tag{9}$$

**Definition 6.** *Given Equation 8 and 9, let* $\mathcal{F}_2(X, Y) = \left\{ (x_1, y_1), (x_2, y_2) \right\}$, $\forall x_1^\dagger \in \mathcal{S}(x_1)$, $y_1^\dagger \in \mathcal{S}(y_1)$, $x_2^\dagger \in \mathcal{S}(x_2)$, $y_2^\dagger \in \mathcal{S}(y_2)$. *If* $\mathcal{R}_\times(y_1^\dagger, x_1^\dagger) = 0$ *and* $\mathcal{R}_\circ(y_2^\dagger, x_2^\dagger) = 1$, *then* $\mathcal{R}_\circ(Y, X) = 1$.

If we apply the above definition to the previous example, we have $(\text{Mary}, \text{cat}, \text{adopts}) \in \mathcal{S}(X)$ and $(\text{Mary}, \text{cat}, \text{not\_adopts}) \in \mathcal{S}(Y)$, and hence $X$ does not entail $Y$. Note that finding a proper split is also tricky, and one solution is each pair of subsets has the same subject, object, or relation. In addition, Definition 6 requires Assumption 3 to be true so that each subset among $X$ and $Y$ does not have intra-contradictions if $\mathcal{F}_2$ is used.

We reformulate Claim 2 and subsequently establish the following lemma:

**Lemma 4.** *Let* $a, b', c'$ *be the text in the Queue, and the elements are inserted in an ordered sequence: $a$ precedes $b'$, and $b'$ precedes $c'$. If* $\mathcal{R}_\circ(b', a) = 1$ *and* $\mathcal{R}_\circ(c', a) = 1$, *then* $\mathcal{R}_\circ(c', b') = 1$.

*Proof.* Assume, without loss of generality, $b$ and $c$ are the texts such that $\mathcal{R}_\times(b, a) = 1$ and $\mathcal{R}_\times(c, a) = 1$. Given that $b'$ and $c'$ are in the Queue, we know $b' = \delta_{\min}(b, a)$ and $c' = \delta_{\min}(c, a)$, so $\mathcal{R}_\circ(b', a) = 1$ and $\mathcal{R}_\circ(c', a) = 1$. Denote $\mathcal{S}(b) = \{\tau_x : \tau_x \in \Delta_a\} \cup \{\tau_y : \tau_y \notin \Delta_a\}$, and $\mathcal{S}(c) = \{\tau_x : \tau_x \in \Delta_a\} \cup \{\tau_y : \tau_y \notin \Delta_a\}$. Suppose Assumption 3 is true, we have $\mathcal{R}_\times(\tau_c^\dagger, \tau_b^\dagger) = 0 \; \forall \tau_b^\dagger \in \{\tau : \tau \notin \Delta_a \wedge \tau \in \mathcal{S}(b)\}$ and $\tau_c^\dagger \in \{\tau : \tau \notin \Delta_a \wedge \tau \in \mathcal{S}(c)\}$. After applying $\delta_{\min}$ for every $\tau_b \in \{\tau : \tau \in \Delta_a \wedge \tau \in \mathcal{S}(b)\}$ and $\tau_c \in \{\tau : \tau \in \Delta_a \wedge \tau \in \mathcal{S}(c)\}$, we have $\tau_a = \tau_b' = \tau_c' \Rightarrow \mathcal{R}_\circ(\tau_c', \tau_b') = 1$. Therefore, $\mathcal{R}_\circ(c', b') = 1$. $\qquad\square$

The main difference between Proposition 4 and Lemma 4 is that Proposition 4 ensures the DELETE preserves transitivity *within* one conversation turn, while Lemma 4 ensures the transitivity still holds *across* different turns. Note that $\delta_{\min}$ will not generate additional information by Definition 3. Otherwise, LLMs may generate two contradictory sequences in different conversation turns.[11]

As Claim 2 is proved, combining Lemma 3 and Corollary 6, we establish the following theorem.

**Theorem 1.** *The KEIC algorithm modifies* $h = [\mathbf{T_f}, \mathbf{T_o}]$ *and returns* $h^* = [\mathbf{T_f^*}, \mathbf{T_o^*}]$ *such that* $\mathbf{T_c}$ *entails* $h^*$.

As $R' \in h^*$, the updated history entails new knowledge.

**Corollary 7.** $h^*$ *entails* $R'$.

# E    DETAILS OF HUMAN EXAMINATION AND KEIC DATASET

In the KEIC dataset, the ratio of "Yes" to "No" is 6 to 5. Figure 9 shows the detailed instructions on the MTurk interface in our pilot study, and Figure 10 displays an example. We describe how the following two KEIC data are generated by three annotators (previous QA pairs are omitted):

**Example 3.** *Story: ..."The information we have at this time is that the 10-year-old did fire the weapon." The mother and the 7-year-old were inside the house when the shooting occurred, said Williams. Williams said the gun belonged to the boy's mother...*
*(Q, A): (was anyone with her?, Yes)*
*Old knowledge: the 7-year-old*
*New knowledge: (1) her dog (2) the pet dog (3) unborn baby*

---

[11] For instance, one turn says, "They're willing to handle the kids! I can go to Tokyo with you," whereas another turn says, "I can't wait to be in California," implying they are going to the States.

**Example 4.** *Story: ...Kyle, a Navy SEAL, has been credited as the most successful sniper in United States military history. Bradley Cooper was nominated for an Academy Award for his portrayal of Kyle in this winter's film "American Sniper," which was based on Kyle's bestselling autobiography. The film, directed by...*

*(Q, A): (was a movie made about him?, yes)*

*Old knowledge: "American Sniper," which was based on Kyle's bestselling autobiography.*

*New knowledge: (1) "American Sniper," which was based on Kyle's comrades bestselling autobiography. (2) , but Kyle's life was not adapted into a movie. (3) "American Sniper," which was based on Kyle's brother bestselling autobiography.*

We instruct workers to maintain the fluency of new knowledge because (1) it aligns with the success of Reiterate, and (2) one of our baselines employs string replacement. Most importantly, free-form sentences simulate how humans correct themselves. Nevertheless, as our primary goal is effective, we occasionally accept a few less fluent responses on condition that we cannot think of a better one.

In Example 3, her in the question refers to the mother. Workers should generate a text indicating she was with something (but *not* a person) because we want the new answer to be "No." Invalid responses, such as "no one," will be rejected by us because the sentence "The mother and no one were inside the house ..." sounds unnatural. Analogously, in Example 4, him in the question refers to Kyle, and valid responses should mention the film American Sniper was not based on Kyle.

We also select the following three examples from the KEIC validation dataset to demonstrate the difficulty of smoothly integrating new knowledge into the middle of the story.

**Example 5.** *Story: ...On the step, I find the elderly Chinese lady, small and slight, holding the hand of a little boy. In her other hand, she holds a paper carrier bag. I know this lady...*

*(Q, A): (Is she carrying something?, Yes)*

*New knowledge: she is holding a cane*

In Example 5, the workers should generate the new knowledge that she is indeed holding something (as "In her other hand" existed before it), but that thing does change the answer to no. Similarly, "the diamond ring gleaming on her finger" is another effective update.

**Example 6.** *Story: ...The store was really big, but Mike found the sugar really fast. When Mike was on his way to the front of the store to pay for the sugar, he saw a toy he had been wanting for a long time. But Mike only had enough money to pay for the sugar or the toy. Mike didn't know what to do! The cake would taste good and would make his mom happy...*

*(Q, A): (Could he afford everything?, no)*

*New knowledge: Mike had enough money to pay for both the sugar and the toy, but a voice inside his head told him not to buy anything unnecessary.*

In Example 6, the workers should generate the new knowledge that Mike could afford everything. However, to maintain the story's fluency, they still need to invent a dilemma for him.

**Example 7.** *Story: ...Featherless baby birds were inside, crying for food. The mother had nothing to give, so she quickly flew to the ground and looked in the dirt for food...*

*(Q, A): (did mom have any?, no)*

*New knowledge: The mother had some seeds inside her beak but it was not enough for the babies*

In Example 7, the workers should generate the new knowledge that the mother bird did have food. Yet again, they have to come up with a situation so that she still needed to look for food.

## F STORY AND QA PAIR EXTRACTION TEMPLATES IN KEIC ALGORITHM

After all the completions in $\{u_1, b_1, b_2\}$ are filled (see Figure 8), we initiate a new chat and ask GPT-3.5 (0613) to extract the story or QA pair based on the last two turns: $b_3 = P(x|u_1, b_1, u_2, b_2, u_3)$. The input also follows the multi-turn format: $u_i$ means role = user, and $b_i$ means role = assistant. In practice, we set the maximum iteration per data to 3 in our KEIC algorithm to avoid a potential infinite loop (*e.g.*, gets "stuck"), which means each turn in the history will be edited at most three times. In addition, the algorithm will terminate once the number of tokens reaches a maximum of 16,385.

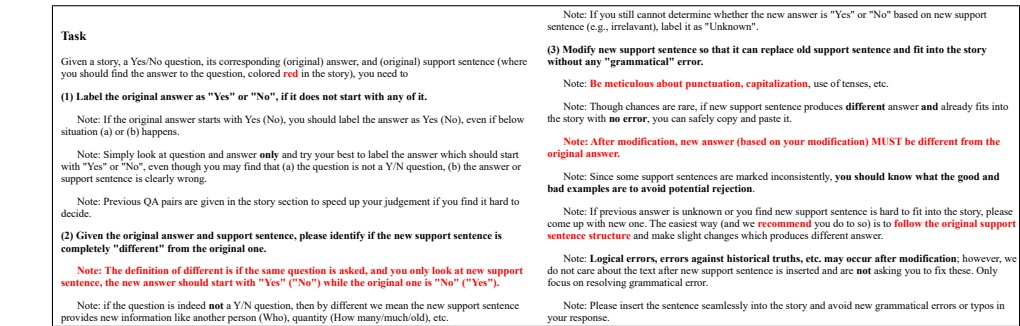

Figure 9: Instructions on the MTurk interface. After our pilot study, we removed the second task, and workers had to generate the new support sentence from scratch (*i.e.*, no reference answer is given in Figure 10). We still include this figure to give more details in the KEIC task.

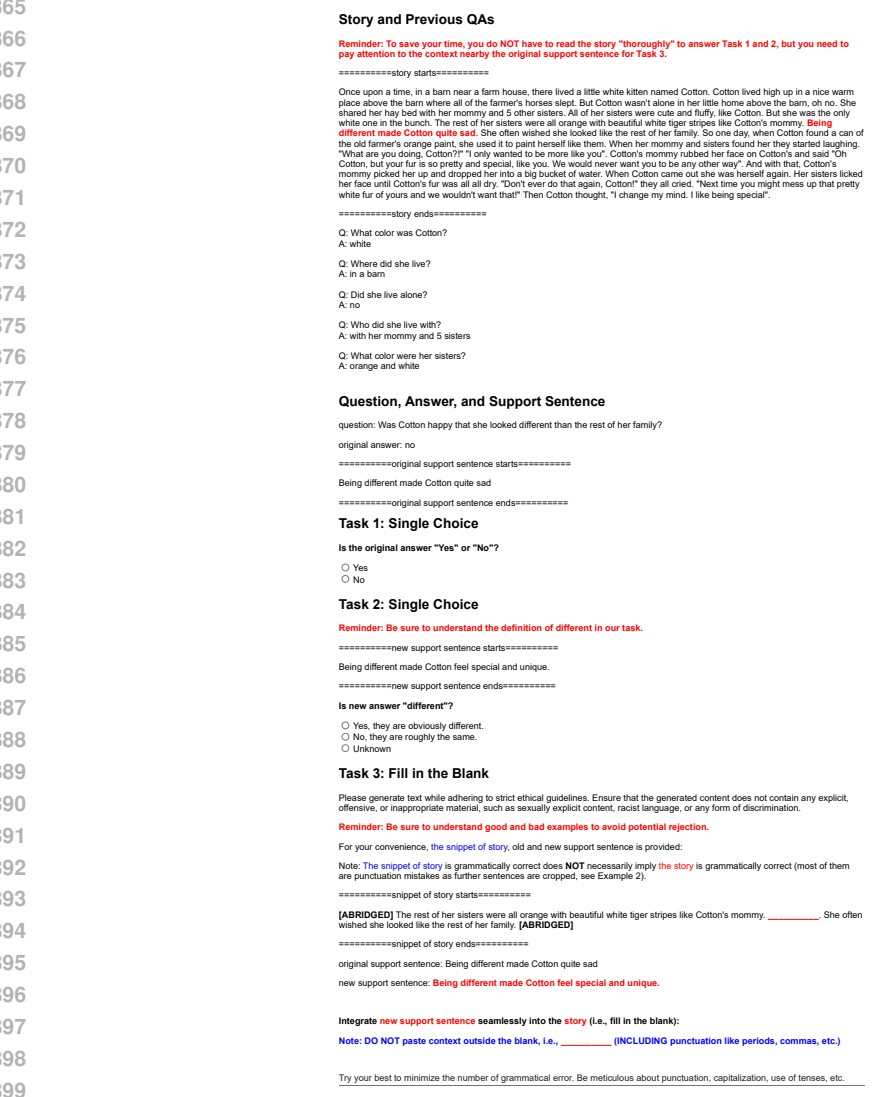

Figure 10: An example on the MTurk interface. As stated in Section 4.1, workers need to fill in the blank (since Task 2 and the "new support sentence" in Task 3 have been removed).

### F.1 STORY EXTRACTION TEMPLATE

$u_1$: Story = """[Story Completion]""" Correction = """[Correction Completion]""" Which parts in the story contradict the correction? If the story entails the correction, output 'NO MODIFICATION'. Let's read the story line by line. List all the contradictions one by one, if any.

$b_1$: [Chat Completion]

$u_2$: Can you modify the story, one by one, so that the correction entails the story?

$b_2$: [Chat Completion]

$u_3$: Therefore, what is the modified story? Output the modified story and nothing else.

### F.2 QA PAIR EXTRACTION TEMPLATE

$u_1$: QA pair = """[QA Completion]""" Correction = """[Correction Completion]""" Does the QA pair contradict the correction? If the QA pair entails the correction, output 'NO MODIFICATION'. If the QA pair contradicts the correction, explain why they are contradictory in one sentence. If they are in a neutral relation, output 'NO MODIFICATION'. Let's think step by step.

$b_1$: [Chat Completion]

$u_2$: Can you modify the QA pair so that it entails the correction? DO NOT modify the QA pair by copying the correction. Let's think step by step.

$b_2$: [Chat Completion]

$u_3$: Therefore, what is the modified QA pair? Your response must contain two lines only. The first line is the question, and the second line is the answer. Output the modified QA pair and nothing else.

## G TIME AND COST ESTIMATION

We use 6 RTX 3090 GPUs and 4 RTX 4090 GPUs for LLM inference. Using GPT-3.5 (0613), the Deletion with only one template in the CBA setting costs nearly $700 in three runs (it will require around $10,000 to fully explore all 15 templates in the CBA setting). Note that the cost can be greatly decreased so long as we restrict the action of appending the conversation history. For instance, we can "reset" the length of conversation to $|h|$ (see Line 6 in Algorithm 1) by initiating a new chat once an iteration is done, though we do not employ this from the outset since our goal is to test the Deletion in the scenario of online conversation (see Table 1 and Figure 8).

The total number of tokens used when running our KEIC dataset ($\mathcal{D}_{KEIC}$) using GPT-4o LLMs are as follows:[12]

| Model | # Input Tokens | # Output Tokens | Total Cost | Experiments |
|---|---|---|---|---|
| GPT-4o | 206,304,490 | 4,151,997 | $557.28 | OTC (w/ AE) |
| GPT-4o (mini) | 472,618,728 | 16,237,303 | $80.64 | OTC, Verification, Reiterate (oracle) |

Observe that # API calls in the OTC (w/ AE) is 2 and # API calls in the oracle of Reiterate is 1. As for the time estimation for other LLMs (Llama, Vicuna, and Gemma), it depends on the GPU used and model size. We give a rough estimation as follows (using GeForce RTX 3090): In Reiterate, they generally need around 20 to 30 seconds to reiterate the story. In Verification, it takes around 3 to 6 seconds when we re-question these LLMs. To quickly reproduce our results, it is best to run each of the correction templates or different MTurk responses in parallel since we run each instance 90 times. If possible, we plan to release those LLM outputs to maximize reproducibility.

## H MORE RESULTS AND DISCUSSION

Appendix H.1 summarizes all experiments conducted in this work. Appendix H.2 provides a comparison of the Reiterate phase with and without the oracle. We plot each LLM's KEIC performance on the KEIC dataset in Appendix H.3 (each LLM has its own figure, which provides more readability compared to Figure 5). The ablation analysis of GPT-3.5 (0613) on $\mathcal{D}_{KEIC}$ is in Appendix H.4. Appendix H.5 is the TEXTGRAD (Yuksekgonul et al., 2024) experiment, a recent zero-shot CoT

---

[12]https://openai.com/api/pricing/

prompting framework. Appendix H.6 is the analysis of using the prompting method (i.e., AE step) for LLM evaluation. Lastly, We provide some analysis regarding whether the factual data is difficult to edit on the fly in Appendix H.7.

## H.1 EXPIERMENTS CONDUCTED

In Table 4, we tabulate experiments conducted on various LLMs in this paper. "Verif" stands for the Verification method. "Reit" stands for the Reiterate method. Seeing that there is a noticeable improvement when the Verification method is employed in GPT-4o (mini), it is also worth experimenting with this approach in GPT-4o and GPT-4.

Table 4: This table summarizes the experiments conducted on various LLMs.

| Model | $\mathcal{D}_{train}$ (1,317 data) | | | $\mathcal{D}_{val}$ (464 data) | | | Notes |
|---|---|---|---|---|---|---|---|
| | OTC | Verif | Reit | OTC | Verif | Reit | |
| GPT-4o | ✓* | ✗ | ✗ | ✓* | ✗ | ✗ | |
| GPT-4o (mini) | ✓ | ✓ | ✓† | ✓ | ✓ | ✓† | |
| GPT-4 | ✗ | ✗ | ✗ | ✓* | ✗ | ✗ | |
| GPT-3.5 (0301) | ✗ | ✗ | ✗ | ✓ | ✗ | ✗ | |
| GPT-3.5 (0613) | ✓ | ✓ | ✓‡ | ✓ | ✓ | ✓ | has Deletion (part) on $\mathcal{D}_{val}$ & ablation analysis on $\mathcal{D}_{KEIC}$ |
| GPT-3.5 (1106) | ✗ | ✗ | ✗ | ✓ | ✗ | ✗ | |
| GPT-3.5 (0125) | ✓ | ✓ | ✓ | ✓ | ✓ | ✓ | has TEXTGRAD result on $\mathcal{D}_{val}$ |
| Gemma-2 (27B) | ✓ | ✗ | ✓† | ✓ | ✗ | ✓† | |
| Gemma-2 (9B) | ✓ | ✓ | ✓ | ✓ | ✓ | ✓ | also has Reiterate (oracle) result |
| Gemma-2 (2B) | ✓ | ✓ | ✓ | ✓ | ✓ | ✓ | also has Reiterate (oracle) result |
| Vicuna (33B) | ✓ | ✗ | ✓† | ✓ | ✗ | ✓† | |
| Vicuna (13B) | ✓ | ✓ | ✓ | ✓ | ✓ | ✓ | also has Reiterate (oracle) result |
| Vicuna (7B) | ✓ | ✓ | ✓ | ✓ | ✓ | ✓ | also has Reiterate (oracle) result |
| Llama-3 (8B) | ✓ | ✓ | ✓ | ✓ | ✓ | ✓ | also has Reiterate (oracle) result |
| Llama-2 (13B) | ✓ | ✓§ | ✓ | ✓ | ✓§ | ✓ | also has Reiterate (oracle) result |
| Llama-2 (7B) | ✓ | ✓§ | ✓§ | ✓ | ✓§ | ✓§ | also has Reiterate (oracle) result |

\* An additional answer extraction is used in the OTC baseline; otherwise, the update is suspiciously low.

† We only conduct the oracle of Reiterate due to the limitation of budgets/computing resources.

‡ We only experiment top-6 templates from $\mathcal{D}_{val}$ due to the budget constraint.

§ During the evaluation, the *last* token in the bot response is also considered (as opposed to the standard evaluation in Section 4.3), or the update is suspiciously low. We do not use this across other methods or LLMs since it has zero or little gains from this. Moreover, they should directly answer the user's Yes/No question (especially in the AE step of Verification) instead of articulating reasons, apologizing, etc.

## H.2 REITERATE V.S. ORACLE OF REITERATE

The oracle of Reiterate is a way to "sanity-check" whether an LLM is equipped with Reiterate capability, especially when the budget or computing resources are limited (see Appendix G). In a real-world scenario, however, this approach can also be thought of as having an *external feedback*,

which does not reflect the LLM's intrinsic self-correction capabilities (Huang et al., 2024).[13] Figure 11 displays their performance in update on $\mathcal{D}_{KEIC}$.

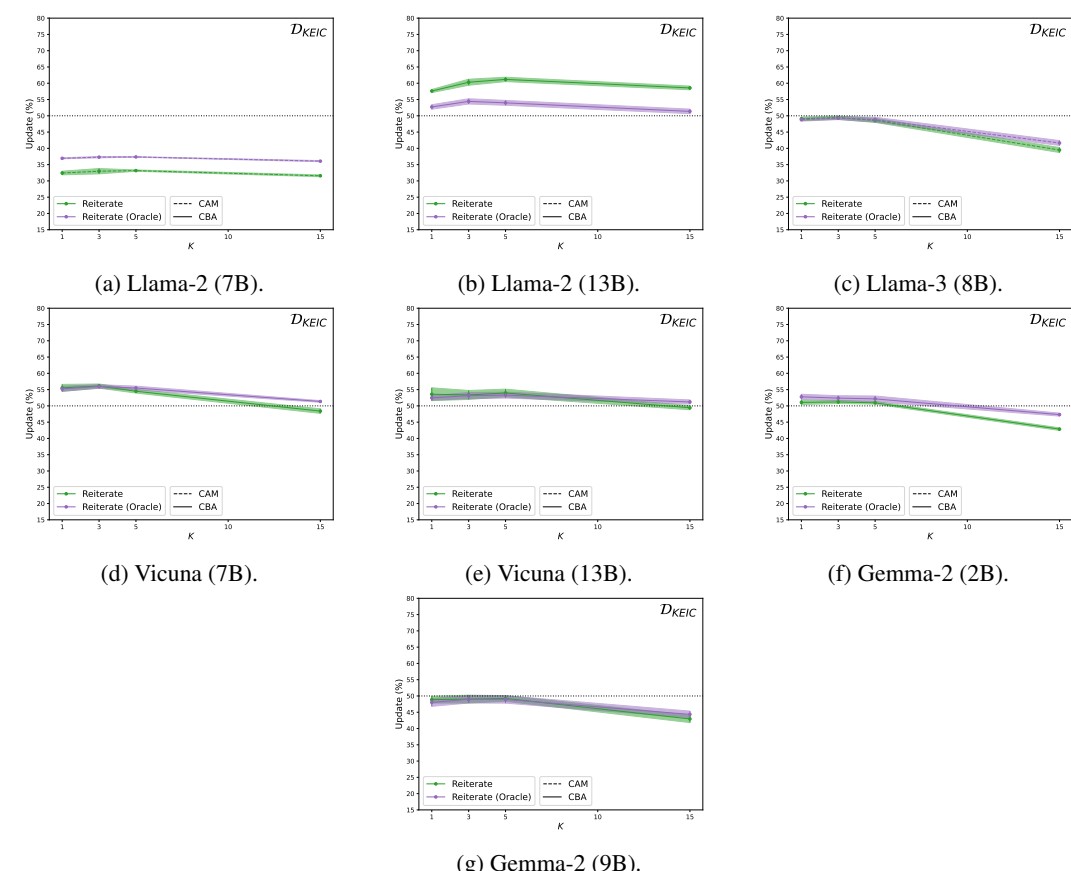

(a) Llama-2 (7B).     (b) Llama-2 (13B).     (c) Llama-3 (8B).

(d) Vicuna (7B).     (e) Vicuna (13B).     (f) Gemma-2 (2B).

(g) Gemma-2 (9B).

Figure 11: Reiterate (green) vs. the oracle of Reiterate (purple). We observe that in Llama-2 (7B), the oracle of Reiterate is higher than the real-world scenario of Reiterate, which may indicate that the model does not truly understand the process of reiterating a new story. Interestingly, it is the other way around in Llama-2 (13B). As for Llama-3, Vicuna, and Gemma-2 LLMs, we speculate that there is no significant boost in update when the oracle is applied in our dataset.

## H.3 FULL RESULTS OF EACH LLM

Similar to Figure 4, we plot the update of all KEIC methods of each LLM on our KEIC dataset in Figure 12. In GPT-3.5 (0613), we do not plot all the templates on $\mathcal{D}_{KEIC}$ because we only run $\mathcal{D}_{train}$ using the top-6 templates from $\mathcal{D}_{val}$ (due to the cost). Compared to the OTC, despite the overall effectiveness of Reiterate on other open-source LLMs, it still leaves a significant room for future work. Our KEIC dataset inherits the properties of CoQA; therefore, editing a false statement in a passage should be inevitably harder than a single sentence (not to mention the previous QA pairs often contain the old knowledge). As a result, to use our dataset to further gauge these LLMs with mediocre KEIC capability, it is worth experimenting with the OTC, Verification, and Reiterate approaches in our KEIC dataset so that the sentences after the support sentence are trimmed.

---

[13]For example, a perfect system that can (1) detect which utterance the user aims to correct in a conversation, (2) locate the false statement within a long paragraph, and (3) generate a new story on its own (Chen & Shu, 2024; Xie et al., 2024).

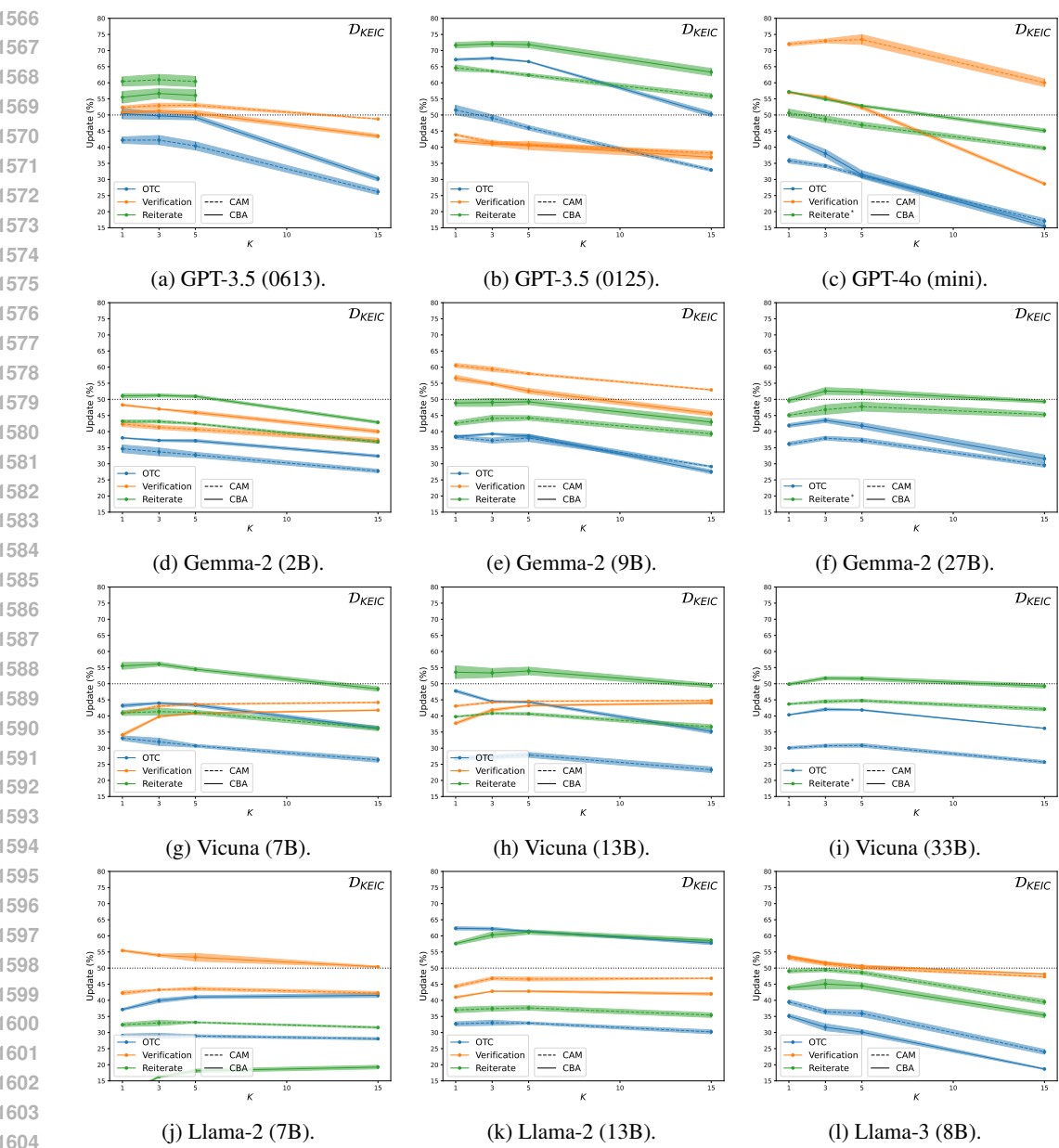

Figure 12: This figure is the update of KEIC methods of each LLM on $\mathcal{D}_{KEIC}$. The Reiterate approach with asterisk (*) in GPT-4o (mini), Gemma-2 (27B), and Vicuna (33B) means the oracle (defined in Section 4.5; see also Appendix H.2). We observe that the Reiterate approach is generally more performant than the OTC baseline on contemporary LLMs, except Llama-2 LLMs: It is worse than or on par with the OTC in its 7B and 13B models. Interestingly, the update in GPT-4o (mini) LLM using the Verification approach in CAM has a significantly better performance than other LLMs.

## H.4 ABLATION ANALYSIS

We assess the importance of pre-defined text segments in the template, such as bot responses in the false and correction phases, through an ablation analysis by removing these segments. We then compare the results against the OTC baseline of GPT-3.5 (0613) on $\mathcal{D}_{KEIC}$. Moreover, we conjecture that the knowledge is more difficult to delete in the middle of the story, so we conduct

another experiment by abridging the story so that the support sentence appears at the end. We tabulate these results in Table 5 and Table 6.

Table 5: Ablation analysis of GPT-3.5 (0613) in the OTC baseline on $\mathcal{D}_{KEIC}$ with the removal of (a) all pre-defined texts from the template (except the user utterance in $\mathbf{T_c}$), (b) the story after old knowledge, and (c) the multi-turn conversation format. Temp stands for template, FP stands for full passage, and MT stands for multi-turn. The percentage of update, no update, and upper bound performance when top-1, 3, 5, and 15 templates are selected are reported. The sum of update and no update is not 100, as we exclude "N/A" in the table (due to the space).

| | Update ($\uparrow$, Maj) | | | | No Update ($\downarrow$, Maj) | | | | Upper Bound ($\uparrow$) | | | |
|---|---|---|---|---|---|---|---|---|---|---|---|---|
| $K$ | 1 | 3 | 5 | 15 | 1 | 3 | 5 | 15 | 1 | 3 | 5 | 15 |
| OTC (CAM) | 42.2 | 42.2 | 40.4 | 26.2 | 50.2 | 52.5 | 54.7 | 70.0 | 42.2 | 52.9 | 53.9 | 55.0 |
| (a) without Temp | 31.8 | 30.6 | 30.2 | 19.4 | 56.3 | 61.2 | 62.5 | 75.3 | 31.8 | 40.6 | 42.6 | 43.5 |
| (b) without FP | 52.5 | 50.0 | 47.8 | 34.7 | 37.1 | 43.0 | 45.5 | 60.2 | 52.5 | 59.7 | 60.8 | 62.1 |
| (c) without MT | 39.7 | 32.8 | 30.3 | 17.4 | 56.4 | 63.9 | 66.6 | 79.9 | 39.7 | 44.8 | 46.3 | 47.1 |
| OTC (CBA) | 50.4 | 49.7 | 49.3 | 30.2 | 38.5 | 41.6 | 42.1 | 63.4 | 50.4 | 60.6 | 61.8 | 63.4 |
| (a) without Temp | 39.8 | 39.9 | 38.9 | 24.4 | 40.3 | 47.4 | 48.9 | 68.6 | 39.8 | 49.8 | 51.8 | 53.7 |
| (b) without FP | 56.4 | 56.7 | 56.3 | 40.1 | 29.0 | 31.8 | 32.4 | 51.3 | 56.4 | 65.4 | 66.4 | 67.8 |
| (c) without MT | 53.3 | 47.9 | 44.5 | 28.8 | 41.7 | 48.5 | 52.1 | 68.3 | 53.3 | 60.1 | 61.6 | 62.6 |

Table 6: The standard deviations across when top-1, 3, 5, and 15 templates are selected are reported. This table follows the same convention as Table 5.

| | Update (Maj) | | | | No Update (Maj) | | | | Upper Bound | | | |
|---|---|---|---|---|---|---|---|---|---|---|---|---|
| $K$ | 1 | 3 | 5 | 15 | 1 | 3 | 5 | 15 | 1 | 3 | 5 | 15 |
| OTC (CAM) | 1.00 | 1.43 | 1.26 | 0.88 | 0.54 | 1.29 | 1.07 | 0.66 | 1.00 | 0.62 | 0.79 | 0.82 |
| (a) without Temp | 0.74 | 0.96 | 0.70 | 0.73 | 0.91 | 0.61 | 0.38 | 0.57 | 0.74 | 0.29 | 0.66 | 0.67 |
| (b) without FP | 0.70 | 0.70 | 0.97 | 1.02 | 0.51 | 0.20 | 0.92 | 0.84 | 0.70 | 0.66 | 0.69 | 0.54 |
| (c) without MT | 0.91 | 0.92 | 0.93 | 0.51 | 0.79 | 0.86 | 0.89 | 0.51 | 0.91 | 1.00 | 1.07 | 1.02 |
| OTC (CBA) | 1.64 | 1.04 | 0.76 | 0.73 | 0.74 | 0.64 | 0.77 | 0.51 | 1.64 | 1.51 | 1.59 | 1.36 |
| (a) without Temp | 1.35 | 0.97 | 0.96 | 0.49 | 1.07 | 1.19 | 1.51 | 0.41 | 1.35 | 0.60 | 0.68 | 0.76 |
| (b) without FP | 1.02 | 0.68 | 0.90 | 0.20 | 0.59 | 0.75 | 0.91 | 0.25 | 1.02 | 0.97 | 0.83 | 0.81 |
| (c) without MT | 1.29 | 1.59 | 1.36 | 1.18 | 1.35 | 1.41 | 1.32 | 1.18 | 1.29 | 0.67 | 0.70 | 0.37 |

If we remove those pre-defined templates, the overall update performance drops by around 10% in both settings, which is not surprising because our pre-defined templates contain bot responses that GPT-3.5 has memorized the story and the knowledge update in the false phase and correction phase, respectively. We also find that the knowledge in the middle of the story is, on average, less likely to be deleted, which is reasonable since the latter part of the story is often based heavily on that false fact. It is noteworthy that while the removal of information after the support sentence so that the knowledge located at the end of the story is much easier for GPT-3.5 to correct, the improvement in the CAM and CBA settings is modest, yielding an enhancement of around 7% to 8% on average compared to the OTC baseline.

**GPT-3.5 is better at capturing information update in a multi-turn framework** We report the single-turn result in Table 5 (*i.e.*, without MT).[14] Though the best performance of update in single-turn (53.3%) is higher than multi-turn (50.4%), the overall performance shows that (1) it dramatically underperforms in CAM (see also their upper bound performance), (2) the update significantly decreases as $|K|$ increases in both setting, especially in the gap between top-1 and top-3, and (3)

---

[14]If a model does not support multi-turn chat format and we want to test it in the KEIC framework, we have to incrementally present the model with $u_1$ to obtain $b_1$, then we provide the model with $\{u_1, b_1, u_2\}$ to acquire $b_2$, and so forth. One solution is to evaluate it by concatenating multiple conversation turns, but this cannot reflect the relation across turns (Zheng et al., 2023b).

the percentage of no update in both settings is consistently higher than the OTC baseline. These aforementioned observations may indicate that if the input format is single-turn, GPT-3.5 (0613) does not generalize well on other correction utterances, and the model is more likely to neglect the new information presented in the middle of context. In other words, GPT-3.5 is generally better at capturing different user utterances and locations of correction in the multi-turn framework.

Table 7: Percentage of Update/No Update/Upper Bound on $\mathcal{D}_{val}$ using GPT-3.5 (0613). This table follows the same convention as Table 2, the 0125 version. Note that Figure 4 can be derived from this table and Table 3.

| Setting | $K$ | Update ($\uparrow$, Maj) | | | No Update ($\downarrow$, Maj) | | | Upper Bound ($\uparrow$) | | |
|---|---|---|---|---|---|---|---|---|---|---|
| | | OTC | Verif | Reiterate | OTC | Verif | Reiterate | OTC | Verif | Reiterate |
| CAM | 1 | $46.3_{(1.4)}$ | $\mathbf{53.5}_{(1.2)}$ | $65.9_{(1.7)}$ | $46.6_{(1.1)}$ | $\mathbf{36.6}_{(0.6)}$ | $26.9_{(0.8)}$ | $46.3_{(1.4)}$ | $53.5_{(1.2)}$ | $65.9_{(1.7)}$ |
| | 3 | $46.6_{(2.0)}$ | $52.2_{(0.4)}$ | $\mathbf{67.1}_{(1.8)}$ | $47.9_{(2.0)}$ | $41.0_{(1.8)}$ | $28.2_{(1.4)}$ | $57.3_{(0.9)}$ | $69.7_{(1.1)}$ | $72.6_{(1.5)}$ |
| | 5 | $44.5_{(2.3)}$ | $53.1_{(1.1)}$ | $66.7_{(1.9)}$ | $50.5_{(2.0)}$ | $41.8_{(0.2)}$ | $29.0_{(1.6)}$ | $58.7_{(1.2)}$ | $75.4_{(0.5)}$ | $73.8_{(1.6)}$ |
| | 15 | $29.2_{(1.6)}$ | $49.3_{(1.0)}$ | $57.3_{(1.1)}$ | $67.1_{(1.2)}$ | $47.3_{(0.8)}$ | $39.2_{(0.9)}$ | $60.5_{(1.1)}$ | $85.9_{(1.0)}$ | $\mathbf{75.4}_{(1.2)}$ |
| CBA | 1 | $\mathbf{58.7}_{(1.2)}$ | $48.0_{(2.3)}$ | $61.5_{(1.4)}$ | $\mathbf{32.6}_{(0.8)}$ | $36.8_{(1.3)}$ | $\mathbf{24.4}_{(1.0)}$ | $58.7_{(1.2)}$ | $48.0_{(2.3)}$ | $61.5_{(1.4)}$ |
| | 3 | $57.8_{(1.0)}$ | $51.3_{(1.7)}$ | $62.4_{(0.6)}$ | $34.9_{(0.8)}$ | $37.9_{(1.1)}$ | $26.3_{(1.3)}$ | $67.8_{(0.7)}$ | $69.0_{(3.0)}$ | $69.5_{(1.0)}$ |
| | 5 | $56.9_{(1.3)}$ | $50.5_{(1.2)}$ | $61.8_{(0.9)}$ | $36.1_{(1.6)}$ | $40.2_{(0.9)}$ | $26.9_{(1.1)}$ | $69.3_{(1.0)}$ | $75.7_{(1.1)}$ | $70.8_{(1.1)}$ |
| | 15 | $36.9_{(1.6)}$ | $41.5_{(0.9)}$ | $51.1_{(1.9)}$ | $57.3_{(1.0)}$ | $52.7_{(1.0)}$ | $40.6_{(1.5)}$ | $\mathbf{71.1}_{(0.4)}$ | $\mathbf{86.3}_{(1.4)}$ | $72.7_{(0.5)}$ |

## H.5 EXPERIMENTS ON THE TEXTGRAD FRAMEWORK

TEXTGRAD is the pioneering work with a released software for *universal, automatic* "differentiation" via text for LLM-based systems, similar to the PyTorch backprop function. The core idea is that they treat a black-box LLM or more sophisticated systems as a "single neuron," so the input/output of that "neuron" can be both in text form. Thus, the "gradient" with respect to this "neuron" is, naturally, the text. Prior to OpenAI o1, the most recent "*think-before-speak*" application[15], they design an automatic way to prompt the GPT-4o (partly GPT-3.5) to stick to the text objective function, provide textual ("gradient") feedback, improve the answer by utilizing various "HTML tags," which is effectively a more complicated CoT framework. Notwithstanding their remarkable success across various tasks, one of the most concerning issues in their current applications is the cost, as either (1) the internal processes are not publicly available or (2) the token consumption cannot be easily calculated in advance.

In this paper, we additionally conduct their framework by feeding our *best* LLM outputs (that is, the 0125 version of GPT-3.5) in the OTC baseline on the validation set into their TEXTGRAD, hoping to identify the error and update the answer. However, our preliminary results show that, when using GPT-4o (0513) in the first run (costs around $250), the best performances of (update, no update) with respect to CAM and CBA are (29.1%, 70.3%) and (27.2%, 72.4%). Moreover, after we set the backend LLM to GPT-3.5 (0125), the best performance of (update, no update) with respect to CAM and CBA are (30.3%, 68.9%) and (24.6%, 74.9%) in 3 runs (worse than without applying their framework, as shown in Figure 7). It would be worth experimenting with using their framework directly or tweaking the prompts (see below).

The prompts are the following (with a slight modification to the example from their website[16]): (1) role description of a variable: "yes/no question to the LLM" (2) role description of an answer: "concise and accurate answer to the yes/no question (the answer should begin with yes or no)" (3) evaluation instruction: "Here's a yes/no question: {question}. Evaluate any given answer to this yes/no question, be smart, logical, and very critical. Just provide concise feedback."

## H.6 LLM EVALUATION

Figure 13 is the comparison between using exact match only (i.e., default evaluation) and using LLM itself for evaluation (i.e., w/ AE; see Section 4.3).

---

[15] https://openai.com/o1/
[16] https://github.com/zou-group/textgrad

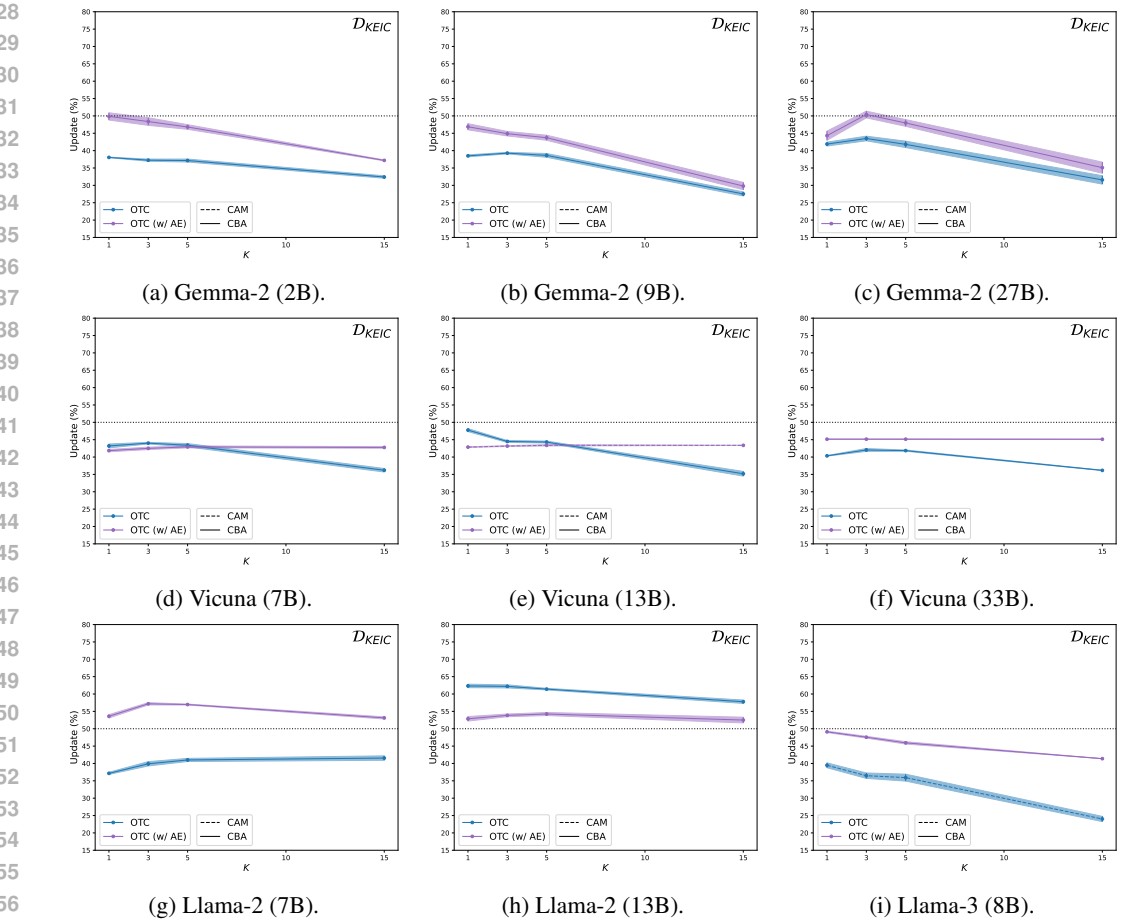

(a) Gemma-2 (2B).  (b) Gemma-2 (9B).  (c) Gemma-2 (27B).

(d) Vicuna (7B).  (e) Vicuna (13B).  (f) Vicuna (33B).

(g) Llama-2 (7B).  (h) Llama-2 (13B).  (i) Llama-3 (8B).

Figure 13: We plot the OTC method (w/ and w/o AE) of Gemma, Vicuna, and Llama LLMs on $\mathcal{D}_{KEIC}$. We observe that (1) the overall update increases in the Gemma LLMs (though it still does not outperform the random guess baseline). (2) In Vicuna, there is not much difference in its 7B and 13B LLMs regarding the top-5 correction templates. (3) Interestingly, the OTC with AE is significantly worse than *without* applying in Llama-2 (13B), while it is the other way around in the 7B model.

## H.7 FATUAL DATA AND NON-FACTUAL DATA

We classify the CoQA data from "Wikipedia" and "CNN" as factual data, and "Gutenberg," "MCTest," and "RACE" as non-factual.[17] Then, we analyze whether factual data is more difficult to edit an LLM's in-context knowledge, using GPT-3.5 (0125) and GPT-4o (0806) as an example. We report the average top-5 update in the CBA setting in Table 8.

Table 8: In this table, we observe that (1) it is easier to edit the in-context knowledge of non-factual data and (2) compared to GPT-3.5, there is a significant gap in updating the factual data of GPT-4o.

| Model | Data | Number | Update (%) | No Update (%) | N/A (%) |
|---|---|---|---|---|---|
| GPT-3.5 (0125) | Factual | 776 | $62.20_{(0.58)}$ | $34.41_{(0.78)}$ | $3.39_{(0.39)}$ |
| | Non-Factual | 1,005 | $69.95_{(0.20)}$ | $26.43_{(0.40)}$ | $3.62_{(0.45)}$ |
| GPT-4o (0806) | Factual | 776 | $25.04_{(1.11)}$ | $74.57_{(1.11)}$ | $0.39_{(0.00)}$ |
| | Non-Factual | 1,005 | $40.73_{(2.13)}$ | $58.47_{(2.13)}$ | $0.80_{(0.00)}$ |

---

[17]Note that it assumes the real-world fact lies within an LLM's parametric memory, and vice versa.

