# OpenReview forum: "On Re-Encoding Short-Term Memory of Large Language Models in Conversations"
_ICLR.cc/2025/Conference — Submitted to ICLR 2025_

### Official Review · Reviewer_yjK4 · 2024-10-27

**Soundness:** 3
**Presentation:** 2
**Contribution:** 3
**Rating:** 5
**Confidence:** 2

**Summary:**

This work studies the Memory Re-Encoding  (MRE) task, which tries to correct the misinformation in the existing dialogue histories on the fly. Consequently, this work proposes a Knowledge Editing In Conversation (KEIC) framework to measure the adaptability of LLMs.

This work is beyond my research scope,  so my opinions may not be accurate.

**Strengths:**

1. The proposed framework KEIC can be used in correcting many general misinformation types (hallucination, notorious, etc.).

2. The proposed framework KEIC does not need to tune the model parameters.

3.  A very detailed and comprehensive problem analysis.

4. This work has built a high-quality dataset.

5. Four model-agnostic MRE methods are proposed and strong experiments are conducted.

**Weaknesses:**

1. The organization and notations should be improved. The current readability is somewhat lacking. For example:

- Line 147-148:  the mixture of $r$ and r.
- The arrangement of Figures is not very good.  The texts on the $n$ page always require checking a figure that appears on the $n-2/3$ page.

2. The proposed methods involve many additional processes. Three advanced methods cost much more tokens (Table 1. #Input Tokens) , which may subsequently worsen the latency.

3. Experiments may lack performance evaluation on the general metrics. For example, using BLEU ROUGE to evaluate the quality of the generated dialogues.

**Questions:**

How does the proposed method affect the general performance of related tasks?

---

> ### Author Response · Authors · 2024-11-13
> **Response to Reviewer yjK4**
>
> > 1. The organization and notations should be improved. The current readability is somewhat lacking. For example:
> > Line 147-148: the mixture of $r$ and r.
>
> Please refer to the **same** Lines 147-148, as we explicitly state: Note that the relation "r" is in the normal text font, which should not be confused with the notation of fact "$r$" (in italics).
>
> > The arrangement of Figures is not very good. The texts on the $n$ page always require checking a figure that appears on the $n - 2/3$ page.
>
> Can the reviewer report which Figures exactly? We have checked that if a Figure is **FIRST** mentioned in text on page x, then that Figure appears on page y (where $y \geq x$).
>
> * Please see an example of Figure 1: It is first mentioned on page 1, which is placed on page 2. Note that we also refer to Figure 1 on pages 2, 4, and 10.
>
> > 2. The proposed methods involve many additional processes. Three advanced methods cost much more tokens (Table 1. #Input Tokens) , which may subsequently worsen the latency.
>
> This is the major drawback of most, if not all, CoT papers.
>
> * Please also refer to more time and token consumption in Appendix G.
>
> > 3. Experiments may lack performance evaluation on the general metrics. For example, using BLEU ROUGE to evaluate the quality of the generated dialogues.
>
> As the question type is Yes/No, using exact match is sufficient in this paper. If open-domain questions are asked, then using these metrics would be suitable.

---

> > ### Author Response · Authors · 2024-11-23
> > **Response to Reviewer yjK4 (cont.)**
> >
> > > > 1. The organization and notations should be improved. The current readability is somewhat lacking. For example:
> > > Line 147-148: the mixture of $r$ and r.
> >
> > > Please refer to the **same** Lines 147-148, as we explicitly state: Note that the relation "r" is in the normal text font, which should not be confused with the notation of fact "$r$" (in italics).
> >
> > [Update] We will re-organize the format. For one thing, we use $\texttt{s}$, $\texttt{o}$, and $\texttt{r}$ (\texttt) instead of the normal text font (s, o, r). For another, we revise the previous context in Lines 162-165:
> >
> > ### Previous Version
> >
> > For instance, given a fact ($r$=*Michael Jordan played fifteen seasons in the $\textcolor{red}{NBA}$*, $q$=*Did Jordan play basketball*, $a$=*Yes*) and its triplet $\mathcal{M}(r) =$ (s=Michael Jordan, $\textcolor{red}{\mbox{o=basketball}}$, r=played sport), one effective fact is $r'$ = ``*Michael Jordan played fifteen seasons in the $\textcolor{blue}{MLB}$*'' because $\mathcal{M}(r') =$ (s=Michael Jordan, $\textcolor{blue}{\mbox{o$'$=baseball}}$, r=played sport) $\in \Delta(r)$ and $a'$ $\in$ {No}.
> >
> > ### Current Version
> >
> > For instance, given a fact $(r, q, a) =$ (*Michael Jordan played fifteen seasons in the $\textcolor{red}{NBA}$*, *Did Jordan play basketball*, *Yes*) and its triplet $\mathcal{M}(r) =$ (Michael Jordan, $\textcolor{red}{\mbox{basketball}}$, played sport), one effective fact is $r' = $ ``*Michael Jordan played fifteen seasons in the $\textcolor{blue}{MLB}$*'' because $\mathcal{M}(r') =$ (Michael Jordan, $\textcolor{blue}{\mbox{baseball}}$, played sport) $\in \Delta(r)$ and $a'$ $\in$ {No}.
> >
> > Specifically, we do not make the relation $\texttt{r}$ and fact $r$ appears in the same equation.

---

### Official Review · Reviewer_q6Uz · 2024-11-02

**Soundness:** 1
**Presentation:** 2
**Contribution:** 2
**Rating:** 3
**Confidence:** 3

**Summary:**

This paper investigates large language models' (LLMs) ability to integrate corrections made during a conversation, a process termed as Memory Re-Encoding (MRE). The authors introduce the Knowledge Editing in Conversation (KEIC) dataset, derived from CoQA, by introducing corrections mid-conversation to test LLMs' adaptation to updated information. Four correction methods—One-Turn Correction (OTC), Verification, Recall, and Deletion—are proposed, with Deletion emerging as the most effective in experimental evaluations on various LLMs, followed by Recall and OTC.

**Strengths:**

- The paper addresses the important challenge of enabling LLMs to incorporate corrections seamlessly, an essential feature for reliable conversational AI.
- The authors’ structured approach to enhancing MRE is interesting.
- Up to section 2, the paper is well-written and easy to follow.

**Weaknesses:**

- The KEIC dataset may lack diversity in conversational flow.
    - Stories are consistently positioned at the start of each dialogue, which simplifies real-world conversational dynamics.
    - Both stories and conversations are brief, with only yes/no questions, potentially narrowing the scope of the model's evaluation. Expanding question types could offer richer insights with minimal adjustments in answer evaluation.
    - Correction utterances appear only adjacent to the story or directly preceding the question.
- The experiments could benefit from a deeper analysis of how model performance varies with factors such as:
    - The distance between the corrected story and the question.
    - The position of correction utterances relative to the question.
- Sections 3–5 could be clearer. Please refer to the questions in the Questions section. I will raise my score if misunderstandings due to unclear explanations are addressed.

**Questions:**

- Are CAM and CBA the only configurations for correction placement? Testing additional positions for corrections (i.e., various places between the correction and the question) could reveal valuable insights into positional effects on model performance.
- Line 314: Is the “update” baseline the original CoQA? The phrase "we directly replace the old fact in the story with a new one" suggests the original story is used without an explicit correction within the conversation.
- Table 1: Why does Deletion require more computational resources in the CAM setting compared to the CBA setting?
- Lines 354–357: This passage is difficult to follow. It would be helpful to reference Figure 9. What does "The goal of evaluating the former approach aligns with that of our baseline with no update phase" mean? Perhaps, this unclarity is related to my question about line 314.
- Line 374: The concept of "top-K upper bound performance" requires clarification and further explanation. Please provide a clearer definition.
- Lines 377–378: The statement regarding the “best five out of 15 correction utterances” is confusing. The definition suggests that performance should increase with higher K values since any of the top-K templates should trigger a correct response. However, the performance in the plots fluctuates or even declines as K increases.
- Figure 5: Does the statement “the baseline with no update phase has 56.5% of update” mean that GPT’s performance on the original CoQA is only 56.5%?
- Table 2: The terms "Update/No Update/Upper Bound" in the caption are not clearly defined, particularly “Upper Bound.”
- Line 452: The poor performance of GPT-4 and GPT-4o is quite unexpected. Please share some specific examples of failure cases.
- Line 464: The paper suggests that the Recall method outperforms OTC. Could this simply be due to Recall positioning the updated story closer to the question than in OTC?

---

> ### Author Response · Authors · 2024-11-18
> **Response to Reviewer q6Uz**
>
> ### Weaknesses
>
> > 1. The KEIC dataset may lack diversity in conversational flow. Stories are consistently positioned at the start of each dialogue, which simplifies real-world conversational dynamics.
>
> The story is placed at the start because that is what the CoQA framework is designed for. Moreover, our framework does not depend on this scenario in Section 2.
>
> * Please also refer to Figure 1, where the false turn does not appear at the start. This example is further exemplified in Section 2.3 in KEIC framework.
>
> > Both stories and conversations are brief, with only yes/no questions, potentially narrowing the scope of the model's evaluation.
>
> Please refer to why we choose the Yes/No questions in Lines 157-158: Because answers are free-form in CoQA, we focus on Yes/No (YN) questions to simplify the analysis.
>
> We also explicitly state the rationale behind why we choose the Y/N answer in the Limitation section (see the "KEIC Dataset" paragraph on page 17).
>
> > Expanding question types could offer richer insights with minimal adjustments in answer evaluation.
>
> In reality, it is not because the answers labeled by MTurk Workers are not often consistent. For example, some people label only the name when asking "who-questions," whereas others don't. This makes exact match evaluation unsuitable and would require further post-processing (or, using LLM to evaluate the answer).
>
> > Correction utterances appear only adjacent to the story or directly preceding the question.
>
> Note that we are interested in whether the correction utterances are nearest or farthest to the test question. Hence, we do not conduct in the middle because we speculated the performance would be somehow in the middle (or, even worse).
>
> > 2. The experiments could benefit from a deeper analysis of how model performance varies with factors such as:
> The distance between the corrected story and the question.
>
> Please see the above response.
>
> > The position of correction utterances relative to the question.
>
> It is done in the ablation analysis in Table 6 on page 31 (Appendix H.4). Specifically, we trimmed the story so that the false statement appears at the end of the story ("without FP" row).
>
> ### Questions
>
> > Are CAM and CBA the only configurations for correction placement? Testing additional positions for corrections (i.e., various places between the correction and the question) could reveal valuable insights into positional effects on model performance.
>
> The answer is Yes for "correction placement." However, we also test different "correction types" and "new facts" in this paper. Please refer to Lines 211-214, Appendix B, and Appendix E.
>
> We already test the two settings (CAM and CBA) and would want to emphasize in Lines 322-323 that **we run each instance 90 times** (15 correction utterances, three MTurk responses, and two settings). Moreover, there are 13 LLMs (different sizes) conducted in this paper. Hence, testing more positions in the middle is another possible further work but is not our first priority, unless there are not budget constraints.
>
> > Line 314: Is the “update” baseline the original CoQA? The phrase "we directly replace the old fact in the story with a new one" suggests the original story is used without an explicit correction within the conversation.
>
> No, it is not the original CoQA data. It is *literally* what it means: "the old fact in the story is replaced by a new one." The story in the original CoQA is changed.
>
> * Regarding a concrete example in the main content, please refer to the footnote 5 on page 5. The red sentence (old fact) in the story is replaced by the blue sentence (new fact).
>
> > Table 1: Why does Deletion require more computational resources in the CAM setting compared to the CBA setting?
>
> It does not mean Deletion in the CAM setting would require more computing resources compared to the CBA. It means utilizing Deletion is costly.
>
> * Please refer to Lines 1438-1439 in Appendix G: Using GPT-3.5 (0613), the Deletion with only one template in the CBA setting costs nearly $700 in three runs.
>
> Hence, if we used the top-1 template in CAM, it would require $700.
>
> > Lines 354–357: This passage is difficult to follow. It would be helpful to reference Figure 9. What does "The goal of evaluating the former approach aligns with that of our baseline with no update phase" mean? Perhaps, this unclarity is related to my question about line 314.
>
> Yes, they are related. Please compare the conversation flow of $\mathbf{T_f}\mathbf{T_p}\mathbf{T_c}\mathbf{T_r}\mathbf{T_d}\mathbf{T_i}$ and $\mathbf{T_f^*}\mathbf{T_p^*}\mathbf{T_i}$. Concretely, the correction phase, recall phase, and deletion phase are excluded in the latter case ($\mathbf{T_f^*}\mathbf{T_p^*}\mathbf{T_i}$).
>
> As you may have noticed, if you compare the two baselines in Line 314, it is (1) "$\mathbf{T_f}\mathbf{T_c}\mathbf{T_p}\mathbf{T_i}$ (CAM) or  $\mathbf{T_f}\mathbf{T_p}\mathbf{T_c}\mathbf{T_i}$ (CBA)" and (2) "$\mathbf{T_f'}\mathbf{T_p}\mathbf{T_i}$".

---

> > ### Author Response · Authors · 2024-11-18
> > **Response to Reviewer q6Uz (cont.)**
> >
> > > Line 374: The concept of "top-K upper bound performance" requires clarification and further explanation. Please provide a clearer definition.
> >
> > We already give an example and explain this in the following Lines 377-380 (For example, to measure the top-5 upper bound in update, we first select the best five out of the 15 correction utterances. If *any* of these triggers an LLM to respond correctly based on the new fact, we consider that the LLM has MRE capability in this KEIC instance).
> >
> > * For example, suppose there are five data and 5 templates, where the templates predict the update sorted by update accuracy is 100%, 80%, 60%, 40%, and 20% ("o" means update, "x" means no update, and "-" means N/A).
> > * 1st : [o, o, o, o, o] $\rightarrow$ 100% update
> > * 2nd: [o, x, o, o, o] $\rightarrow$ 80% update
> > * 3rd : [x, o, x, o, o] $\rightarrow$ 60% update
> > * 4th : [x, -, o, x, o] $\rightarrow$ 40% update
> > * 5th : [-, x, -, o, x] $\rightarrow$ 20% update
> > * The top-3 upper bound would be selecting 1st to 3rd templates and then see any of the three templates updates the answer in each data instance. Hence, the top-3 upper bound would be [o, o, o, o, o] $\rightarrow$ 100% update.
> > * We give another example of the upper bound of "3rd to 5th" templates (since the 1st template has 100% update). The upper bound performance of the "3rd, 4th, and 5th" templates is [x, o, o, o, o] $\rightarrow$ 80% update. Because the first data is (x, x, -), the second data is (o, -, x), the third data is (x, o, -), the fourth data is (o, x, o), and the fifth data is (o, o, x).
> >
> >
> > > Lines 377–378: The statement regarding the “best five out of 15 correction utterances” is confusing. The definition suggests that performance should increase with higher K values since any of the top-K templates should trigger a correct response. However, the performance in the plots fluctuates or even declines as K increases.
> >
> > No, please refer to **OTC paragraph in Section 3** (two types of correction templates) and **Transferability of correction phase across MRE approaches in Section 6**.
> >
> > For instance, suppose there are five data and 5 templates as follows:
> > * 1st : [o, o, o, o, o] $\rightarrow$ 100% update
> > * 2nd: [o, x, o, o, o] $\rightarrow$ 80% update
> > * 3rd : [x, o, x, o, o] $\rightarrow$ 60% update
> > * 4th : [x, -, o, x, o] $\rightarrow$ 40% update
> > * 5th : [-, x, -, o, x] $\rightarrow$ 20% update
> >
> > If we do majority voting of all the templates ("o": +1, "x": -1, "-": 0), it is [-, -, o, o, o] $\rightarrow$ 60% update. However, the top-3 majority voting update performance is [o, o, o, o, o] $\rightarrow$ 100% update. Hence, there is a decline when all templates are used in some LLMs because they significantly underform in other types of correction utterances.
> >
> > > Figure 5: Does the statement “the baseline with no update phase has 56.5% of update” mean that GPT’s performance on the original CoQA is only 56.5%?
> >
> > No, it's not. Please refer to your question regarding Line 314. The 56.5% update is the baseline of $\mathbf{T_f'}\mathbf{T_p}\mathbf{T_i}$.
> >
> > > Table 2: The terms "Update/No Update/Upper Bound" in the caption are not clearly defined, particularly “Upper Bound.”
> >
> > This is the duplicate question regarding the upper bound. Please refer to the above response.
> >
> > > Line 452: The poor performance of GPT-4 and GPT-4o is quite unexpected. Please share some specific examples of failure cases.
> >
> > We provide one of the examples that GPT-4 (1106) fail (CAM setting, 8th correction template, 1st MTurk response) on the validation set. Note that CAM is the best performing setting, as displayed in Figure 7. **Note also that GPT-4 (1106) fail on this CoQA instance in all 30 cases (15 correction utterances and 2 (CAM and CBA) settings).**
> >
> > **Please refer to the next response for full conversation flow.**
> >
> > correction templates: I'm sorry to bring this up, but I mistakenly gave you "[O]." In fact, "[N]"
> >
> > old fact: "Being different made Cotton quite sad"
> >
> > new fact: "Being different made Cotton feel special and unique"
> >
> > first OTC response: "No, being different made Cotton quite sad. She wished she looked like the rest of her family."
> >
> > second answer extraction (AE) step: 'No'
> >
> >
> > > Line 464: The paper suggests that the Recall method outperforms OTC. Could this simply be due to Recall positioning the updated story closer to the question than in OTC?
> >
> > We speculate once we somehow "bypass" the LLM and let it state the "updated context," it could overwrite the in-context knowledge or even its underlying parameters. However, further analysis would be necessary for future work. This could also be seen in the work of [1] (see Line 1565).
> >
> > [1] Xie et al., Adaptive chameleon or stubborn sloth: Revealing the behavior of large language models in knowledge conflicts, ICLR 2024

---

> > > ### Author Response · Authors · 2024-11-18
> > > **An Example that GPT-4 fail**
> > >
> > > * $u_1$ 'Read and memorize the following story.\n\nStory:\n\nOnce upon a time, in a barn near a farm house, there lived a little white kitten named Cotton. Cotton lived high up in a nice warm place above the barn where all of the farmer\'s horses slept. But Cotton wasn\'t alone in her little home above the barn, oh no. She shared her hay bed with her mommy and 5 other sisters. All of her sisters were cute and fluffy, like Cotton. But she was the only white one in the bunch. The rest of her sisters were all orange with beautiful white tiger stripes like Cotton\'s mommy. $\textcolor{red}{\text{Being different made Cotton quite sad}}$. She often wished she looked like the rest of her family. So one day, when Cotton found a can of the old farmer\'s orange paint, she used it to paint herself like them. When her mommy and sisters found her they started laughing. \n\n"What are you doing, Cotton?!" \n\n"I only wanted to be more like you". \n\nCotton\'s mommy rubbed her face on Cotton\'s and said "Oh Cotton, but your fur is so pretty and special, like you. We would never want you to be any other way". And with that, Cotton\'s mommy picked her up and dropped her into a big bucket of water. When Cotton came out she was herself again. Her sisters licked her face until Cotton\'s fur was all all dry. \n\n"Don\'t ever do that again, Cotton!" they all cried. "Next time you might mess up that pretty white fur of yours and we wouldn\'t want that!" \n\nThen Cotton thought, "I change my mind. I like being special".\n\n==========\n\nHave you memorized the story?'
> > > * $b_1$: Yes, I have memorized the story.
> > > * $u_2$: I'm sorry to bring this up, but I mistakenly gave you "$\textcolor{red}{\text{Being different made Cotton quite sad.}}$" In fact, "$\textcolor{blue}{\text{Being different made Cotton feel special and unique}}$"
> > > * $b_2$: No problem at all! I have updated my memory of the story with the correction you provided. Thank you for letting me know.
> > > * $u_3$: What color was Cotton?
> > > * $b_3$: white
> > > * $u_4$: Where did she live?
> > > * $b_4$: in a barn
> > > * $u_5$: Did she live alone?
> > > * $b_5$: no
> > > * $u_6$: Who did she live with?
> > > * $b_6$: with her mommy and 5 sisters
> > > * $u_7$: What color were her sisters?
> > > * $b_7$: orange and white
> > > * $u_8$: Was Cotton happy that she looked different than the rest of her family?
> > > * $b_8$: No, being different made Cotton quite sad. She wished she looked like the rest of her family.
> > > * $u_9$: Therefore, based on your previous response, your answer to the last question is more likely to be "Yes", "No"? You must output "Yes" or "No" first.
> > > * $b_9$: No

---

> > > ### Comment · Reviewer_q6Uz · 2024-11-20
> > >
> > > >>Line 374: The concept of "top-K upper bound performance" requires clarification and further explanation. Please provide a clearer definition.
> > > >
> > > >We already give an example and explain this in the following Lines 377-380 (For example, to measure the top-5 upper bound in update, we first select the best five out of the 15 correction utterances. If any of these triggers an LLM to respond correctly based on the new fact, we consider that the LLM has MRE capability in this KEIC instance).
> > > >
> > > >>Lines 377–378: The statement regarding the “best five out of 15 correction utterances” is confusing. The definition suggests that performance should increase with higher K values since any of the top-K templates should trigger a correct response. However, the performance in the plots fluctuates or even declines as K increases.
> > > >
> > > >No, please refer to OTC paragraph in Section 3 (two types of correction templates) and Transferability of correction phase across MRE approaches in Section 6.
> > > >
> > > >For instance, suppose there are five data and 5 templates as follows:
> > >
> > > So, do the y-axes in Figures 5 and 6 represent the majority voting performance? If so, this should be clarified.
> > >
> > > >>Table 2: The terms "Update/No Update/Upper Bound" in the caption are not clearly defined, particularly “Upper Bound.”
> > > >
> > > >This is the duplicate question regarding the upper bound. Please refer to the above response.
> > >
> > > Understood now.
> > >
> > > I recommend the definitions of upper bound performance and majority voting performance be presented more clearly at the same place in the paper.
> > >
> > > >>Line 452: The poor performance of GPT-4 and GPT-4o is quite unexpected. Please share some specific examples of failure cases.
> > > >
> > > >We provide one of the examples that GPT-4 (1106) fail (CAM setting, 8th correction template, 1st MTurk response) on the validation set. Note that CAM is the best performing setting, as displayed in Figure 7. Note also that GPT-4 (1106) fail on this CoQA instance in all 30 cases (15 correction utterances and 2 (CAM and CBA) settings).
> > >
> > > Thank you for the example.
> > >
> > > >>Line 464: The paper suggests that the Recall method outperforms OTC. Could this simply be due to Recall positioning the updated story closer to the question than in OTC?
> > > >
> > > >We speculate once we somehow "bypass" the LLM and let it state the "updated context," it could overwrite the in-context knowledge or even its underlying parameters. However, further analysis would be necessary for future work. This could also be seen in the work of [1] (see Line 1565).
> > >
> > > This remains speculative partly because there is no analysis of positional effects of correction utterances.

---

> > > > ### Author Response · Authors · 2024-11-21
> > > >
> > > > > So, do the y-axes in Figures 5 and 6 represent the majority voting performance? If so, this should be clarified.
> > > >
> > > > Yes.
> > > >
> > > > > I recommend the definitions of upper bound performance and majority voting performance be presented more clearly at the same place in the paper.
> > > >
> > > > Sure.
> > > >
> > > > >> We speculate once we somehow "bypass" the LLM and let it state the "updated context," it could overwrite the in-context knowledge or even its underlying parameters. However, further analysis would be necessary for future work. This could also be seen in the work of [1] (see Line 1565).
> > > >
> > > > > This remains speculative partly because there is no analysis of positional effects of correction utterances.
> > > >
> > > > As [1] experimented with this (swapping context) in their paper, it would be somewhat incremental. Despite this, your insight can be added to the potential future work. Thank you.

---

> > ### Comment · Reviewer_q6Uz · 2024-11-20
> >
> > >> The KEIC dataset may lack diversity in conversational flow. Stories are consistently positioned at the start of each dialogue, which simplifies real-world conversational dynamics.
> >
> > > The story is placed at the start because that is what the CoQA framework is designed for. Moreover, our framework does not depend on this scenario in Section 2.
> > >
> > >Please also refer to Figure 1, where the false turn does not appear at the start. This example is further exemplified in Section 2.3 in KEIC framework.
> >
> > As far as I understand, your KEIC dataset is based on CoQA and always has stories at the beginning of each dialogue, and all your experiments are based on this assumption, no?
> >
> > >>Both stories and conversations are brief, with only yes/no questions, potentially narrowing the scope of the model's evaluation.
> > >
> > >Please refer to why we choose the Yes/No questions in Lines 157-158: Because answers are free-form in CoQA, we focus on Yes/No (YN) questions to simplify the analysis.
> > >
> > >We also explicitly state the rationale behind why we choose the Y/N answer in the Limitation section (see the "KEIC Dataset" paragraph on page 17).
> >
> > I understand your rationale, and it seems like both of us agree that this is a limitation.
> >
> > >>Expanding question types could offer richer insights with minimal adjustments in answer evaluation.
> > >
> > >In reality, it is not because the answers labeled by MTurk Workers are not often consistent. For example, some people label only the name when asking "who-questions," whereas others don't. This makes exact match evaluation unsuitable and would require further post-processing (or, using LLM to evaluate the answer).
> >
> > What are some common evaluation methods used for short answer questions in CoQA? The evaluation of short and long answer questions has been a long-standing challenge, with many approaches proposed. I am not convinced that it is justified to ignore these questions simply because each one is annotated with multiple answers. This also makes the work appear incomplete, as a “memory” test is conducted only in a yes/no format.
> >
> > >>Correction utterances appear only adjacent to the story or directly preceding the question.
> > >
> > >Note that we are interested in whether the correction utterances are nearest or farthest to the test question. Hence, we do not conduct in the middle because we speculated the performance would be somehow in the middle (or, even worse).
> >
> > It would be nice to verify your speculation, especially since adding this setting is very straightforward and easy — you can simply place a correction utterance at difference positions in each dialogue without making any other changes.

---

> > > ### Author Response · Authors · 2024-11-21
> > >
> > > It appears that there are some misunderstandings during the discussion so we would like to further clarify these.
> > >
> > > > > The story is placed at the start because that is what the CoQA framework is designed for. Moreover, our framework does not depend on this scenario in Section 2. Please also refer to Figure 1, where the false turn does not appear at the start. This example is further exemplified in Section 2.3 in KEIC framework.
> > >
> > > > As far as I understand, your KEIC dataset is based on CoQA and always has stories at the beginning of each dialogue, and all your experiments are based on this assumption, no?
> > >
> > > It is. Please see our previous response: We mentioned the **dataset** has the story at the beginning (because it is what it is in their CoQA framework).
> > >
> > > On the other hand, we emphasize that our proposed KEIC **framework** do **not** rely on this (i.e., the false statement is always in the first turn) and Section 2.3 describes this scenario (using Figure 1 as an example). Moreover, our framework do not ask LLMs to refer to the first turn of the dialogue.
> > >
> > > **Since an LLM need to respond the test question with a corrected answer (to align with the "user update"), it needs to catch the "user update" in the middle of a conversation.** As such, from an LLM's point of view, if the test question is the $n$-th turn, the **corrected fact (user update)** could potentially be in any turn of a conversation (i.e., 2nd, 3rd, ... (n-1)-th turn; though our primarily experiment it is the 2nd and (n-1)-th turn). Hence, the KEIC framework can be applied to any conversational dataset in this scenario.
> > >
> > > Note that we have to bold the above context because the reviewer really missed the key point in this paper in the later comments: (1) As far as I understand, a new story refers to the original story in CoQA... and (2) ... but to me "T_f' T_p T_i" looks like an original CoQA dialogue without any correction.
> > >
> > > They will be addressed later. In brief, we would like to emphasize that the update turn (containing "user update") is in the *middle* of the conversation. To answer the test question "based on the 'user update'," an LLM need to catch the "user update" in the middle of the conversation. Even if the story containing a false fact is in the first turn, the new fact ("user update") are not.
> > >
> > >
> > > That said, we can conduct some experiments by randomly inserting unrelated text (e.g., utterances in DailyDialog) before the story using GPT-4o (mini) in the OTC baseline in the Appendix and assess whether the performance drops.
> > >
> > >
> > > > What are some common evaluation methods used for short answer questions in CoQA? The evaluation of short and long answer questions has been a long-standing challenge, with many approaches proposed. I am not convinced that it is justified to ignore these questions simply because each one is annotated with multiple answers. This also makes the work appear incomplete, as a “memory” test is conducted only in a yes/no format.
> > >
> > > (Typically) For short answer: exact match (accuracy). For long answer: F1.
> > >
> > > **Since Yes/No question is considered short** (that is, if a person asks another, the response typically starts with a Yes/No), **it does not make the evaluation incomplete in this paper.**
> > >
> > > As for "a 'memory' test is conducted only in a yes/no format," we would like to emphasize that most, if not all, wh-questions can be converted to Yes/No (i.e., binary), and there is a study on this such as [1]. For example,
> > > * "Q: Who is the president? A: X" $\rightarrow$ "Q: Is X the president? A: Yes"
> > > * "Q: Where are you going? A: Y" $\rightarrow$ "Q: Are you going to Z? A: No"
> > >
> > > We are aware of potential random guessing behavior. And, if an LLM only randomly guesses the answer, then the accuracy is 50%. Notably, we mentioned "The average 'random guess' update is 50%" in Figure 5's caption and drawn this random guess baseline in Figures 5, 6, 7, 8, 12, and 13.
> > >
> > > > It would be nice to verify your speculation, especially since adding this setting is very straightforward and easy — you can simply place a correction utterance at difference positions in each dialogue without making any other changes.
> > >
> > > We could insert this in the *middle* using GPT-4o (mini) in the OTC baseline. But for the budget issue, it would be impractical to conduct *all* the locations, as it would require extraordinary time and budget to run all the experiments. Specifically, if the previous phase has $n$ QA pairs, it would require $n+1$ insertion for LLM inference. To visualize, the conversation flow is as follows (from left to right):
> > >
> > > False (story) $-$[insert (done in CAM)]$-$ QA1 $-$[insert]$-$ QA2 $-$[insert]$-$ QA3 $-$[insert (done in CBA)]$-$ test question (QA4)
> > >
> > > [1] Clark et al., BoolQ: Exploring the Surprising Difficulty of Natural Yes/No Questions, NAACL 2019

---

> > > ### Author Response · Authors · 2024-11-28
> > > **Correction utterances in the middle experiment**
> > >
> > > >>> Correction utterances appear only adjacent to the story or directly preceding the question.
> > >
> > > >> Note that we are interested in whether the correction utterances are nearest or farthest to the test question. Hence, we do not conduct in the middle because we speculated the performance would be somehow in the middle (or, even worse).
> > >
> > > > It would be nice to verify your speculation, especially since adding this setting is very straightforward and easy — you can simply place a correction utterance at difference positions in each dialogue without making any other changes.
> > >
> > > We experiment when the correction utterances is (exactly) in the middle using GPT-4o (mini), and the Update (accuracy) results are as follows (on the validation set, # = 464):
> > >
> > > This table is the original performance when the correction utterances is exactly after the false story or before the test question:
> > >
> > > | Setting \ Top-K | 1     | 3     | 5     | 15    |
> > > |-----------------|-------|-------|-------|-------|
> > > | CAM             | $42.67_{(0.77)}$ | $41.16_{(0.78)}$ | $37.79_{(0.66)}$ | $23.49_{(0.57)}$ |
> > > | CBA             | $51.80_{(1.30)}$ | $46.91_{(0.87)}$ | $41.66_{(0.97)}$ | $21.62_{(0.69)}$ |
> > >
> > > This table is when the correction utterances is (exactly) in the middle of the previous QA pairs (It is not surprising that CAM and CBA are nearly identical because the location is fixed):
> > >
> > > | Setting \ Top-K | 1     | 3     | 5     | 15    |
> > > |-----------------|-------|-------|-------|-------|
> > > | CAM (middle)           | $36.13_{(0.25)}$ | $32.33_{(0.99)}$ | $27.16_{(0.57)}$ | $13.22_{(0.76)}$ |
> > > | CBA (middle)            | $36.42_{(0.77)}$ | $31.61_{(0.87)}$ | $27.01_{(0.54)}$ | $13.58_{(1.35)}$ |
> > >
> > > From the above results we found that the performance drops (i.e., the middle setting is worse than CAM or CBA setting in GPT-4o (mini)).

---

> > > > ### Comment · Reviewer_q6Uz · 2024-12-03
> > > >
> > > > Thank you for the clarification and analysis.
> > > >
> > > > I mistakenly conflated old facts with new facts. Part of the reason for this confusion is that, in this case, the original answer should be flipped (correct?), yet the paper does not mention this explicitly (I couldn't find it). I naturally assumed that the answer remained the same, leading me to believe that the original story should be included in the correction utterance. Additionally, I did not find any statements strongly contradicting this misunderstanding.
> > > >
> > > > I appreciate the authors' efforts to address the reviewers' questions and feedback.
> > > > While the authors emphasize that the KEIC framework is general enough, all their claims are based on a limited experimental setting, where a story is placed at the beginning of a dialogue, and this setting feels somewhat unnatural for daily conversations. I am not convinced that the claims in the paper would straightforwardly generalize to other dialogue settings.
> > > > Furthermore, as other reviewers have also pointed out, the clarity of the paper needs improvement. Therefore, I will maintain my original score.

---

> > > > > ### Author Response · Authors · 2024-12-03
> > > > >
> > > > > > Part of the reason for this confusion is that, in this case, the original answer should be flipped (correct?), yet the paper does not mention this explicitly (I couldn't find it). I naturally assumed that the answer remained the same, leading me to believe that the original story should be included in the correction utterance. Additionally, I did not find any statements strongly contradicting this misunderstanding.
> > > > >
> > > > > This is an interesting statement.
> > > > >
> > > > > 1. If you look at Figure 2 on page 2, then you will find that it is explicitly mentioned in the caption $-$ illustrated in the figure as well $-$ that
> > > > >
> > > > > ```
> > > > > We consider an LLM updates its knowledge if its answer to the same question is changed (e.g., "No" → "Yes"), then we evaluate this "update" behavior on four LLMs (see §4.3).
> > > > > ```
> > > > >
> > > > > 2. In Section 2 (Lines 117-118), we mention in the first sentence that "The KEIC task aims to test if an LLM can dynamically update its knowledge when the user corrects the previous false fact" $\rightarrow$ How could this behavior not change the answer?
> > > > >
> > > > > 3. Our effective is defined as $a' \in $ { $x \in \mathcal{Y} : x \neq a$ }, and $ \mathcal{Y} \in$ {Yes, No} in Section 2.2. How can $a'$ and $a$ be the same, if you look at the mathematical definition? This is really a simple set notation in discrete mathematics. Furthermore, *even if you found that difficult*, an explicit example using Michael Jordan is given below.
> > > > >
> > > > >
> > > > > We don't quite understand why there is a serious misunderstanding in your review.

---

> > > > > > ### Comment · Reviewer_q6Uz · 2024-12-03
> > > > > >
> > > > > > >1. If you look at Figure 2 on page 2, then you will find that it is explicitly mentioned in the caption $-$ illustrated in the figure as well $-$ that
> > > > > >
> > > > > > I apologize for missing the detail in the caption of Figure 2.
> > > > > > Considering the importance of flipping the original answers in the CoQA data, it would be helpful to clearly describe this process in the main text, such as Section 2.4.
> > > > > >
> > > > > > >2. In Section 2 (Lines 117-118), we mention in the first sentence that "The KEIC task aims to test if an LLM can dynamically update its knowledge when the user corrects the previous false fact" $\rightarrow$ How could this behavior not change the answer?
> > > > > >
> > > > > > Because I thought the modified stories are placed at the beginning of the dialogues and the original stories in CoQA are placed in correction utterances.
> > > > > >
> > > > > > > 3. Our effective is defined as $a' \in $ { $x \in \mathcal{Y} : x \neq a$ }, and $ \mathcal{Y} \in$ {Yes, No} in Section 2.2. How can $a'$ and $a$ be the same, if you look at the mathematical definition? This is really a simple set notation in discrete mathematics. Furthermore, even if you found that difficult, an explicit example using Michael Jordan is given below.
> > > > > >
> > > > > > I failed to infer that this Michael Jordan example suggests the original answers in CoQA should be reversed. The overwhelming use of notations is likely another factor contributing to the distraction.

---

> > > > > > > ### Author Response · Authors · 2024-12-03
> > > > > > >
> > > > > > > > I apologize for missing the detail in the caption of Figure 2. Considering the importance of flipping the original answers in the CoQA data, it would be helpful to clearly describe this process in the main text, such as Section 2.4.
> > > > > > >
> > > > > > > Do you know that these three are not an exhaustive search...?
> > > > > > >
> > > > > > > In Section 2.4, we write that "In CoQA data, $T_1$ is always in the false phase because we render a piece of text in the passage $P$ obsolete for the user to correct afterward (and $P \in u_1$)" This explicitly state that the original passage is obsolete (i.e., false).
> > > > > > >
> > > > > > > Moreover, in Section 2.3, we mention that "In our KEIC dataset, we extend each instance from CoQA by labeling misinformation in the passage and adding a correction."
> > > > > > >
> > > > > > > ...Do we have to list one by one, from page 1 to page 10, to infer that you miss all of them?

---

> > ### Comment · Reviewer_q6Uz · 2024-11-20
> >
> > >>Are CAM and CBA the only configurations for correction placement? Testing additional positions for corrections (i.e., various places between the correction and the question) could reveal valuable insights into positional effects on model performance.
> > >
> > >The answer is Yes for "correction placement." However, we also test different "correction types" and "new facts" in this paper. Please refer to Lines 211-214, Appendix B, and Appendix E.
> > >
> > >We already test the two settings (CAM and CBA) and would want to emphasize in Lines 322-323 that we run each instance 90 times (15 correction utterances, three MTurk responses, and two settings). Moreover, there are 13 LLMs (different sizes) conducted in this paper. Hence, testing more positions in the middle is another possible further work but is not our first priority, unless there are not budget constraints.
> >
> > Understood.
> >
> > >>Line 314: Is the “update” baseline the original CoQA? The phrase "we directly replace the old fact in the story with a new one" suggests the original story is used without an explicit correction within the conversation.
> > >
> > >No, it is not the original CoQA data. It is literally what it means: "the old fact in the story is replaced by a new one." The story in the original CoQA is changed.
> > >
> > >Regarding a concrete example in the main content, please refer to the footnote 5 on page 5. The red sentence (old fact) in the story is replaced by the blue sentence (new fact).
> >
> > Sorry I meant the "no update" baseline. As far as I understand, a new story refers to the original story in CoQA and an old story refers to a modified story that will be corrected later in the dialogue. Hence, "the old fact in the story is replaced by a new one" sounds like the original story in CoQA is presented at the beginning of the dialogue. If my understanding is incorrect, I assume many readers will get confused as well.
> >
> > >>Table 1: Why does Deletion require more computational resources in the CAM setting compared to the CBA setting?
> > >
> > >It does not mean Deletion in the CAM setting would require more computing resources compared to the CBA. It means utilizing Deletion is costly.
> > >
> > >Please refer to Lines 1438-1439 in Appendix G: Using GPT-3.5 (0613), the Deletion with only one template in the CBA setting costs nearly $700 in three runs.
> > >Hence, if we used the top-1 template in CAM, it would require $700.
> >
> > Understood.
> >
> > >>Lines 354–357: This passage is difficult to follow. It would be helpful to reference Figure 9. What does "The goal of evaluating the former approach aligns with that of our baseline with no update phase" mean? Perhaps, this unclarity is related to my question about line 314.
> > >
> > >Yes, they are related. Please compare the conversation flow of ...
> >
> > Unfortunately I cannot follow this because I still don't understand the second baseline. I don't know exactly what you mean by T_f', but to me "T_f' T_p T_i" looks like an original CoQA dialogue without any correction (which you seem to be pointing out as a misunderstanding above).

---

> > > ### Author Response · Authors · 2024-11-21
> > >
> > > > Sorry I meant the "no update" baseline.
> > >
> > > No, there are no "no update" baseline.
> > >
> > > > As far as I understand, a new story refers to the original story in CoQA and an old story refers to a modified story that will be corrected later in the dialogue. Hence, "the old fact in the story is replaced by a new one" sounds like the original story in CoQA is presented at the beginning of the dialogue.
> > >
> > > A new story refers to "the new story in CoQA" and an old story refers to "an old story" (that will be corrected later in the dialogue).
> > >
> > > "The old fact in the story is replaced by a new one" means "in the original (old) story, the old fact is replaced by a new fact" (see also footnote 5 for concrete example). Specifically,
> > > * old (original CoQA) story: CHAPTER XII “..., Miss Baldwin,” Wingate invited, as $\textcolor{red}{\text{he ushered that young lady into his rooms}}$ soon after ....
> > > * old fact: $\textcolor{red}{\text{he ushered that young lady into his rooms}}$
> > > * new fact: $\textcolor{blue}{\text{he ushered that old lady into his rooms}}$
> > > * new (modified) story: CHAPTER XII “..., Miss Baldwin,” Wingate invited, as $\textcolor{blue}{\text{he ushered that old lady into his rooms}}$ soon after ....
> > >
> > > > If my understanding is incorrect, I assume many readers will get confused as well.
> > >
> > > If possible, could the reviewer kindly explain which context makes you think of the "new story" refers to "original (old)" and "old" refers to "modified (new)"?
> > >
> > > This is not really common interpretation and we are sure that the context do not even make the readers interpret like this.
> > >
> > > > Unfortunately I cannot follow this because I still don't understand the second baseline. I don't know exactly what you mean by T_f', but to me "T_f' T_p T_i" looks like an original CoQA dialogue without any correction (which you seem to be pointing out as a misunderstanding above).
> > >
> > > This is similar to the above misunderstanding. And we hope that this can be clarified for now.
> > >
> > > As for the reviewer does not know what $\mathbf{T_f'}$ is, we explained this in the footnote 7 (which is in the same line): $\mathbf{T_f'} = \mathbf{T_f}  \setminus R_{old} \cup R_{new}'$ (similar to the success in Recall).
> > >
> > > This notation states that (re-using the previous examples):
> > >
> > > * $\mathbf{T_f}$ = old (original CoQA) story: CHAPTER XII “..., Miss Baldwin,” Wingate invited, as $\textcolor{red}{\text{he ushered that young lady into his rooms}}$ soon after ....
> > > * $R_{old}$ = old fact: $\textcolor{red}{\text{he ushered that young lady into his rooms}}$
> > > * $R_{new}'$ = new fact: $\textcolor{blue}{\text{he ushered that old lady into his rooms}}$
> > > * $\mathbf{T_f'}$ = new (modified) story: CHAPTER XII “..., Miss Baldwin,” Wingate invited, as $\textcolor{blue}{\text{he ushered that old lady into his rooms}}$ soon after ....

---

### Official Review · Reviewer_MxUL · 2024-11-05

**Soundness:** 2
**Presentation:** 2
**Contribution:** 2
**Rating:** 3
**Confidence:** 4

**Summary:**

This paper introduces an in-context memory re-encoding process for large language models (LLMs), allowing memory updates and probing through conversational in-context prompts.
To achieve this, the authors created a dataset comprising {background context (a story), pairs of probing Q&A in a human-AI conversational format, and a final probing question}.

The paper details four in-context prompt memory updating types:

1. One-Turn Correction (OTC): The human corrects the LLM immediately after an error is made.
2. Verification: The LLM first answers a probing question, followed by a human verification question to confirm the accuracy.
3. Recall: The LLM rewrites the entire story after human correction, followed by probing.
4. Deletion: The LLM removes any sentence in the story that contradicts newly updated information.

The paper evaluates these memory-updating prompt schemes across different LLMs.

**Strengths:**

- Introduces a comprehensive dataset for studying in-context memory updating.
- Proposes four distinct prompting methods for updating memory in LLMs.

**Weaknesses:**

The paper has several issues, despite arguing for the importance of in-context memory editing over parametric knowledge editing:

1. While the task setting is intriguing, it seems more like adversarial question answering in a long-term conversational context (i.e., questions that are designed to trick the model into providing wrong answer) than true memory editing. Context-level memory editing would ideally involve managing an external database that saves and updates facts or events within the conversation [1][2].

2. There is a discrepancy between the dataset and the examples in Figures 1 and 3. The figures suggest memory editing in a human-to-human conversational context, while the dataset primarily involves single-turn stories and human-AI QA-style probing. To reduce potential reader confusion, I recommend that the authors use real data examples as motivating illustrations, better aligning the figures with the dataset's actual structure.

3. The authors used 15 different memory-updating prompts but only one prompt for memory probing. Exploring prompts that focus on recent information or limit historical context could yield more convincing results. -- "Based on the recent correction from the user, how would you answer to this question: {question}?" can be one example.

[1] Personalized Large Language Model Assistant with Evolving Conditional Memory., Yuan et al., 2024

[2] Evaluating Very Long-Term Conversational Memory of LLM Agents., Maharana et al., 2024

**Questions:**

None

---

> ### Author Response · Authors · 2024-11-13
> **Response to Reviewer MxUL**
>
> > 1. While the task setting is intriguing, it seems more like adversarial question answering in a long-term conversational context (i.e., questions that are designed to trick the model into providing wrong answer) than true memory editing.
>
> It is more related to "preference alignment" because we want the LLM to update based on user correction.
>
> It is adversarial if the correction utterance is somewhat "misleading" (see [1] and [2]), such as we ask "the president of country A (suppose the answer is X)," but later mention that "the president of country B is changed to Y," then we ask "Who is the president of country A?" at the end of the dialogue (the answer is still X).
>
> [1] Jia and Liang, Adversarial Examples for Evaluating Reading Comprehension Systems, EMNLP 2017
>
> [2] Shi et al., Large Language Models Can Be Easily Distracted by Irrelevant Context, ICML 2023
>
> > Context-level memory editing would ideally involve managing an external database that saves and updates facts or events within the conversation [1][2].
>
> This line of work is mentioned in Lines 93-96 and the Related Work in Lines 944-948 (see Appendix A).
>
> Our (1) "OTC baseline without update phase" and (2) "oracle of Recall" (see Lines 1512-1513 in Appendix H) can be viewed as having a (perfect) external system to update facts in the story (**because we directly replace the old fact in the story with a new one**, as stated in Line 315).
>
> As for the Deletion, it can also be viewed as utilizing external systems to update facts because each turn is fed into the two modules (see Lines 260-261 and the corresponding footnote), which are initiated separately (see Appendix F for implementation).
>
>
> > 2. There is a discrepancy between the dataset and the examples in Figures 1 and 3. The figures suggest memory editing in a human-to-human conversational context, while the dataset primarily involves single-turn stories and human-AI QA-style probing. To reduce potential reader confusion, I recommend that the authors use real data examples as motivating illustrations, better aligning the figures with the dataset's actual structure.
>
> **The KEIC task and framework do not differentiate human-human or human-AI conversations and apply to both scenarios. Please refer to the definition in Section 2.**
> * Notably, we illustrate how to map Figure 1's conversation (human-human) into our framework in Section 2.3.
> * As for another type of conversation (human-AI), see Section 2.4 (also see Figure 4 for one of our real KEIC data).
>
> > 3. The authors used 15 different memory-updating prompts but only one prompt for memory probing. Exploring prompts that focus on recent information or limit historical context could yield more convincing results. -- "Based on the recent correction from the user, how would you answer to this question: {question}?" can be one example.
>
> This could be used for future work, yet we directly use the same question in the CoQA dataset.
>
> From another perspective, given that the original question does not provide strings like "Based on the recent correction from the user," this demonstrates the difficulty of the OTC baseline (because the questions sound more natural for humans), which further showcases that there is a significant performance boost using the Recall approach in GPT-3.5 and other LLMs in Figure 13 (on page 30).

---

> ### Author Response · Authors · 2024-11-20
> **Response to Reviewer MxUL (cont.)**
>
> We would like to further elaborate on why using the original question. Apart from directly re-using the question from the CoQA task, in the KE task, the same question is typically asked before/after the LM is edited. Some paraphrase questions are asked but often do not make too much change.
>
> As we understand the reviewer is asking for better prompting in test questions, we experiment and report the 3rd Weakness on the validation set ($|\mathcal{D}_{val}|$ = 464) using GPT-4o (mini), and the Update (accuracy) results are as follows:
>
> * **GPT-4o (mini)** using the **original** question in this paper (i.e., question)
>
> | Setting \ Top-K | 1     | 3     | 5     | 15    |
> |-----------------|-------|-------|-------|-------|
> | CAM             | $42.67_{(0.77)}$ | $41.16_{(0.78)}$ | $37.79_{(0.66)}$ | $23.49_{(0.57)}$ |
> | CBA             | $51.80_{(1.30)}$ | $46.91_{(0.87)}$ | $41.66_{(0.97)}$ | $21.62_{(0.69)}$ |
>
> * **GPT-4o (mini)** using the **suggested** question (i.e., Based on the recent correction from the user, how would you answer to this question: {question}?):
>
> | Setting \ Top-K | 1     | 3     | 5     | 15    |
> |-----------------|-------|-------|-------|-------|
> | CAM             | $65.16_{(1.56)}$ | $67.14_{(1.22)}$ | $66.17_{(1.31)}$ | $57.40_{(1.25)}$ |
> | CBA             | $65.95_{(0.78)}$ | $67.38_{(1.19)}$ | $66.74_{(0.66)}$ | $58.84_{(1.08)}$ |
>
>
> On the other hand,
>
> * **GPT-3.5 (0125)**, using the **original** question in this paper (i.e., question)
>
> | Setting \ Top-K | 1     | 3     | 5     | 15    |
> |-----------------|-------|-------|-------|-------|
> | CAM             | $57.47_{(2.05)}$ | $56.25_{(1.08)}$ | $53.23_{(0.86)}$ | $39.30_{(1.08)}$ |
> | CBA             | $73.56_{(0.76)}$ | $73.78_{(1.08)}$ | $73.06_{(0.57)}$ | $60.34_{(0.99)}$ |
>
>
> Despite an improvement in GPT-4o (mini), we find that (1) the suggested question does not solve this problem perfectly (below 70%) and (2) "GPT-4o (mini) w/ suggested questions" still does not outperform "GPT-3.5 (0125) w/ original questions" in the CBA setting.
>
> It further demonstrates that the GPT LLM does not attend to the most recent context in this KEIC task even when the question is directly specified for it to do so (reviewer T49M).

---

> ### Comment · Reviewer_MxUL · 2024-11-20
> **Response to Author**
>
> Dear author,
>
> Thank you for your reply.
>
> 1. I agree that the task is not strictly adversarial QA. While the setup does not involve explicit adversarial elements, such as deliberately misleading questions, it exhibits adversarial-like characteristics. Specifically $T_c$ and $T_i$ create scenarios where the model’s ability to update memory and reason through conflicting inputs is tested, which aligns with some goals of adversarial QA. However, I want to emphasize that **I do not believe this task is directly related to memory re-encoding**.
> The correction introduced in $T_c$ appears to impact only the immediate dialogue flow, with no clear evidence that the updated information persists beyond the current session.
> Once the conversation ends, it is uncertain whether the system retains this correction for future interactions.
> This raises questions about the purpose of such short-term memory updates if they do not lead to lasting re-encoding.
>
> 2. Thank you for the clarification. It would be helpful if these points were explicitly clarified in your paper.
> The use of varying dialogue formats across different figures creates confusion and makes it difficult to consistently interpret the task setting.
> Aligning the dialogue examples with a consistent format or providing additional context for the variations would greatly improve clarity.
>
> 3. Thank you for sharing the additional experiments. While GPT-3.5 with the original question outperforms GPT-4o-mini with the suggested question, it is evident that different prompt formats can lead to substantial variance in the final results. To address this, it would be helpful to report the standard deviation across different prompt formats to quantify the variability. Additionally, identifying or optimizing the best prompt format for the task would further strengthen the conclusions and enhance the reliability of the findings.

---

> > ### Author Response · Authors · 2024-11-20
> >
> > > 1.
> >
> > As another reviewer (t3Go) raises this concern, we can remove this term and consistently use KEIC in this paper.
> >
> > > 2.
> >
> > This will be addressed.
> >
> > > 3.
> >
> > Since we report the average performance in 3 runs, we add the standard deviations in the previous tables (in parentheses). Regarding the best templates for this task or different KEIC approaches, reviewer T49M asked a similar question, and we are willing to provide the analysis in the Appendix.

---

### Official Review · Reviewer_T49M · 2024-11-07

**Soundness:** 3
**Presentation:** 3
**Contribution:** 3
**Rating:** 6
**Confidence:** 4

**Summary:**

This paper examines the scenario where context that appears previously in an ongoing conversation with a LLM is corrected by the user and evaluates whether the LLMs can reflect the corrected context in their responses. The authors formalize this scenario as the Knowledge-editing in-conversation (KEIC) framework and refer to the phenomena of knowledge update as memory reencoding (MRE). Specifically, KEIC and MRE refer to non-parametric and in-context knowledge editing i.e., the model parameters are not updated and the model’s behavior is influenced through prompting only. The authors of this paper collect a dataset for evaluating KEIC in contemporary LLMs through four different update methods. The dataset is collected by asking human annotators to edit a fact in the CoQA dataset, which is the updated knowledge. Accordingly, the ground truth answer to the corresponding Yes/No question in the CoQA dataset is updated. With this dataset, the authors propose four update methods: One-Time Correction (OTC), Recall, Verification and Deletion under two different settings i.e., Correction After Mistake (CAM) where the update is applied right after the incorrect fact is presented in the conversation and Correction Before Asking (CBA) where the update is applied right before asking the corresponding Yes/No question. Results are presented for various GPT, Gemma, LLaMA and Vicuna models. Updates are applied through 15 correction templates mined from Daily Dialog. Results show that the Deletion-based update method is most effective, but the Recall-based update method has the best trade-off between effectiveness and token-efficiency. Moreover, the CBA works better than CAM, which is expected because of the proximity of the update tokens to the QA tokens in the auto-regressive model. Surprisingly, gpt4o models are less reactive to these update methods than GPT3.5 models. Overall, there is room for improvement in KEIC in contemporary LLMs.

**Strengths:**

**Interesting topic in LLM conversational memory and well-written paper**: The scenario of KEIC is an interesting, if not one of the most important, topics in the domain of conversational memory of LLMs. The authors do a great job of formalizing the KEIC process, which serves as a framework for effectively thinking about in-context knowledge editing.


**Dataset contribution**: The authors contribute an important dataset that contains conversations with edited knowledge and corresponding questions. It is a sizable dataset and can be further extended to questions beyond the Yes/No format in a semi-synthetic manner by leveraging the KEIC framework for future work.


**Extensive results**: The authors present several plausible settings, methods and conduct extensive experiments under these settings, showing crucial gaps in the performance of current LLMs. Importantly, these experiments provide further evidence that LLMs are not 100% proficient at attending to context, which can prohibit their deployment in error-sensitive situations. Some of these results are surprising and worth investigating further i.e., GPT4o performs worse than GPT3.5 in the OTC setting.

**Weaknesses:**

**Limited Generalizability of Results**: The correction templates adopted by the authors (and listed in Appendix B) indicate that the knowledge updates that are present in the collected dataset are of a single kind i.e., fixing errors in previously reported events. This leaves out some other highly-plausible knowledge editing scenarios in conversations such as (1) temporal change in knowledge i.e., previous knowledge was true at that time but it has now evolved to a different state (such as change in Presidents) (2) evolving of user preferences or opinions (such as a change in music taste or other user preferences like name) etc. This also stems partly from the scope of the CoQA dataset which has a format of two participants discussing a passage from news or fiction. Further, the authors do not make the distinction between real-world facts and narrative fiction in this dataset. It is unclear from the results if it is hard to edit knowledge in an in-context manner if that knowledge is present in the parametric memory of the model (it is also entirely possible that GPT models are trained on the CoQA models and so this data exists in their parametric memory).

**Doubts about the persistency of the KEIC scenario**: Since this problem of models underperforming in the OTC scenario can be traced back to the issue of LLMs not attending to their context correctly, I am unsure if this will continue being a problem if it can simply be fixed by better positional encodings, training data, attention architectures in oncoming versions of LLMs.

**Additional Results**: There are three aspects missing from the Results section:

(1) Since the KEIC problem is a reflection of how good LLMs are at attending to their in-context memory, it warrants analysis in terms of the length of the input. For example, does the update % change with longer distance between the $T_{u}$ (update turns) and $T_{i}$ test turns. Similarly, does the update % vary with longer distance between $T_{e}$ (error turns) and $T_u$?

(2) Some of the correction templates improve coreference resolution at the update step while other don't. How would the results look for one template vs. the other? On that note, is there one or more templates that consistently stay in the top-1,3,5 results?

(3) It would be great to see results from some of the latest open-source models such as LLaMA 3.1, OlMo, and perform analysis on the attention weights is possible.


**Confusing Jargon**: The authors have introduced some terms that may be prohibitive in understanding the results correctly. For instance, the reported metric is 'Update %' in Figure 5. It is unnecessary since it has already been made clear that the metric is simply the accuracy of the Yes/No question. I would suggest that the authors stick to 'Accuracy' to reduce confusion. Another example is the use of 'Recall' for a method, which is a frequently used term for evaluation metrics. Using this term as a method makes for confusing reading. Another term is 'Previous Phase' which really refers to the 'Irrelevant turns'.

**Questions:**

- Who do you think that MRE is a standalone problem that should be studied and not merely a different sub-perspective of the conversational memory problem that could be potentially solved, for example, by using RAG + external memory module or better long-context modeling approaches?

- Suggestion to separate CBA and CAM results in figures to allow the reader to consider both settings individual in terms of the four proposed methods.

See Weaknesses for other suggestions.

---

> ### Author Response · Authors · 2024-11-15
> **Response to Reviewer T49M**
>
> ## Limited Generalizability of Results
>
> > The correction templates adopted by the authors (and listed in Appendix B) indicate that the knowledge updates that are present in the collected dataset are of a single kind i.e., fixing errors in previously reported events. This leaves out some other highly-plausible knowledge editing scenarios in conversations such as (1) temporal change in knowledge i.e., previous knowledge was true at that time but it has now evolved to a different state (such as change in Presidents) (2) evolving of user preferences or opinions (such as a change in music taste or other user preferences like name) etc. This also stems partly from the scope of the CoQA dataset which has a format of two participants discussing a passage from news or fiction.
>
> The CoQA dataset indeed does not contain temporal change in knowledge and user preferences because the discussion is often fiction or news. However, existing datasets encompass this like MultiWoZ datasets are quite noisy (lots of annotation errors) in their papers (please refer to the evolvement of MultiWOZ 2 to 2.4 and other conference/workshop papers addressing this issue). As for the others, they generally are not long-term dialogue $-$ rather easy in this sense $-$ so we exclude them to test our framework comprehensively.
>
> As a result, we choose to use the CoQA dataset in this paper (long-term dialogue + the false fact lies within a long (user) utterance).
>
> Moreover, our pilot study in the MultiWoZ dataset when only using "exact match" (i.e., strict string matching) shows that the result still underperforms. Applying this to other datasets, or proposing an effective way to evaluate the open domain question (without losing explainability) is also a promising avenue for future work.
>
> Note that even if the CoQA dataset does not contain temporal change in knowledge and user preferences, our correction templates can also be applied to these two facets:
>
> * Given (1) old fact = "I like jazz" (``jazz``) and (2) new fact = "I like classical music" (``classical music``), we can directly use these templates and say (a) "Actually, I like classical music" and (b) "Oh, I'm sorry. Should have been "I like classical music," not "I like jazz."
>
> * Incidentally, if those datasets do not label the entire sentence (i.e., ``jazz`` and ``classical music``), then it is more natural for us to say (a) "Actually, classical music" and (b) "Oh, I'm sorry. Should have been classical music, not jazz."
>
> * For another example of changing the current president, it would be something like (a) "Actually, Donald Trump is the U.S. president" (label the entire sentence) and (b) "Oh, I'm sorry. Should have been Donald Trump, not Joe Biden." (label the name only)
>
> > Further, the authors do not make the distinction between real-world facts and narrative fiction in this dataset. It is unclear from the results if it is hard to edit knowledge in an in-context manner if that knowledge is present in the parametric memory of the model (it is also entirely possible that GPT models are trained on the CoQA models and so this data exists in their parametric memory).
>
> As the dataset is 1,781 and further partitioning the KEIC dataset would make fact and non-fact data < 1,000, we do not include this error analysis in this paper. However, if the reviewer is indeed interested in this question, we will provide some analyses later.
>
> [update] we classify "wikipedia" and "cnn" as factual data, whereas "gutenberg," "mctest," and "race" as non-factual data. The exact number of factual data is 776, so the number of non-factual data is 1,005. Please refer to the below "Analysis of factual/non-factual data."
>
> ## Doubts about the persistency of the KEIC scenario
>
> > Since this problem of models underperforming in the OTC scenario can be traced back to the issue of LLMs not attending to their context correctly, I am unsure if this will continue being a problem if it can simply be fixed by better positional encodings, training data, attention architectures in oncoming versions of LLMs.
>
> This task is an unsolved, emerging issue of LLMs and several training methodologies are proposed (e.g., DPO and KTO), it is still unknown whether applying these would solve the problem (though we believe using DPO could probably solve it). However, as we mention in the Related Work (see Lines 932-936), applying this should be cautious because this requires more advanced contextual understanding. It may be exploited by malicious users to produce harmful or counter-fact responses, as stated in the Ethics Statement.
>
> As for the oncoming LLMs, we test a series of state-of-the-art GPT LLMs and demonstrate that the problem persists even in GPT-4 and GPT-4o LLMs (see Figure 6-a; also see Figure 13 on page 30).

---

> ### Author Response · Authors · 2024-11-15
> **Response to T49M (cont.)**
>
> ## Additional Results
> > (1) Since the KEIC problem is a reflection of how good LLMs are at attending to their in-context memory, it warrants analysis in terms of the length of the input. For example, does the update % change with longer distance between the $T_u$ (update turns) and $T_i$ test turns. Similarly, does the update % vary with longer distance between $T_e$ (error turns) and $T_u$?
>
> We'll analyze this later.
>
> [update] Please refer to the below "Analysis of the length (i.e., # turns) of the previous phase"
>
> > (2) Some of the correction templates improve coreference resolution at the update step while other don't. How would the results look for one template vs. the other? On that note, is there one or more templates that consistently stay in the top-1,3,5 results?
>
> It really depends on the settings. In this paper, we have {OTC, Verification, Recall} x {CAM, CBA} x {3 MTurk Responses}.
> Here we list the roughly scanned trend (**not sorted by performance**) of GPT-3.5 (0125):
>
> OTC: 2, 3, 4, 6, 8, 9.
>
> Verification: 5, 7, 10, 11, 12.
>
> Recall: 2, 3, 4, 6, 8, 9.
>
> > (3) It would be great to see results from some of the latest open-source models such as LLaMA 3.1, OlMo, and perform analysis on the attention weights is possible.
>
> **We report the open-source Gemma-2 LLM (2B, 9B, and 27B) in this paper (the release date is Jun 2024). Since we conducted LLaMA 3 before LLaMA 3.1 was released, we do not test LLaMA 3.1 (highly similar).** However, we could test LLaMA 3.1 in some experiments and report the results in the Appendix after the anonymous period.
>
> ## Confusing Jargon
>
> > The authors have introduced some terms that may be prohibitive in understanding the results correctly. For instance, the reported metric is 'Update %' in Figure 5. It is unnecessary since it has already been made clear that the metric is simply the accuracy of the Yes/No question. I would suggest that the authors stick to 'Accuracy' to reduce confusion.
>
> **We use "Update" instead of Accuracy (Acc) in Lines 308-311 because we also report how many percent of data that an LLM stick to its original knowledge ("No Update")**, given that the user corrects the false statement. Hence, we use "Update" (there is no opposite term of "No Acc" from our knowledge).
>
> > Another example is the use of 'Recall' for a method, which is a frequently used term for evaluation metrics. Using this term as a method makes for confusing reading.
>
> [update] We will rename the "recall" method as "reiterate."
>
> > Another term is 'Previous Phase' which really refers to the 'Irrelevant turns'.
>
> **These turns are not always "irrelevant" in CoQA**, as stated in Lines 185-186, 255-257, and Appendix D (the initiative of the IC-MRE algorithm). Hence, renaming this could be more misleading from this perspective.
> Moreover, the CoQA paper also states that their dataset needs to answer the question based on the previous question/answer (coreference, or, incremental aspect), not the passage only. Please refer to Figure 2 in their paper.
>
> On the other hand, since we use $\mathbf{T_i}$ to denote the test phase (we name it from the $i$-th question's perspective; $\mathbf{T_t}$ looks weird), it would also contradict the "irrelevant phase" (if we use the first letter of irrelevant, which is "i").
>
> [update] We can also rename the "previous phase" to "other phase" so there is no notation conflict.
>
> ## Questions
>
> > Who do you think that MRE is a standalone problem that should be studied and not merely a different sub-perspective of the conversational memory problem that could be potentially solved, for example, by using RAG + external memory module or better long-context modeling approaches?
>
> See the 4th paragraph in the Introduction. Some cases are easy enough and using RAG would overkill this task (see Figure 1).
>
> Though using RAG + external memory could potentially solve this problem, it still persists if an LLM is not trained with such adaptability. There are other drawbacks (e.g., computational overhead) using these methods when the changes become more frequent or subtle (see Lines 93-96).
>
> Please also refer to Lines 930-936 in Related Work (for future dataset construction):
> * ... to align with the user's updated knowledge, we highlight that the chatbot sometimes even needs to contradict its previous in-context response to ensure the conversation remains accurate and coherent ... these conversational datasets, although aiming to improve an LLM’s consistency and reduce self-contradiction is of paramount importance, may hamper its adaptability ... In light of this, balancing between the two seemingly paradoxical yet highly correlated tasks during training would be one of the key challenges and opportunities for future work.
>
> > Suggestion to separate CBA and CAM results in figures to allow the reader to consider both settings individual in terms of the four proposed methods.
>
> We can add both CBA and CAM results in Figure 13 on page 30 (the current version only reports the best for now).

---

> ### Author Response · Authors · 2024-11-23
> **Analysis of factual/non-factual data**
>
> >> Further, the authors do not make the distinction between real-world facts and narrative fiction in this dataset. It is unclear from the results if it is hard to edit knowledge in an in-context manner if that knowledge is present in the parametric memory of the model (it is also entirely possible that GPT models are trained on the CoQA models and so this data exists in their parametric memory).
>
> > As the dataset is 1,781 and further partitioning the KEIC dataset would make fact and non-fact data < 1,000, we do not include this error analysis in this paper. However, if the reviewer is indeed interested in this question, we will provide some analyses later.
>
> > [update] we classify "wikipedia" and "cnn" as factual data, whereas "gutenberg," "mctest," and "race" as non-factual data. The exact number of factual data is 776, so the number of non-factual data is 1,005.
>
>
> We analyze GPT-3.5 (0125)'s top-5 templates of the OTC baseline on our dataset (# = 1,781).
>
> We define factual data as wikipedia and cnn, so the rest are non-factual data (gutenberg, mctest, and race).
>
> | GPT-3.5 (0125) | Update       | No Update    | N/A        |
> |----------------|--------------|--------------|------------|
> | Factual (# = 776)       | 478 (61.60%) | 271 (34.92%) | 27 (3.48%) |
> | Non-Factual (# = 1,005)    | 705 (70.15%) | 268 (26.67%) | 32 (3.18%) |
>
>
> From the above table, we could deduce that non-factual data is rather "easier" to edit on the fly (though it is unknown whether the GPT LLM or other LLMs are trained on factual/non-factual data unless they explicitly mentioned in their technical reports or papers).

---

> ### Author Response · Authors · 2024-11-24
> **Analysis of the length (i.e., # turns) of the previous phase**
>
> >> (1) Since the KEIC problem is a reflection of how good LLMs are at attending to their in-context memory, it warrants analysis in terms of the length of the input. For example, does the update % change with longer distance between the $T_u$ (update turns) and $T_i$ test turns. Similarly, does the update % vary with longer distance between $T_e$ (error turns) and $T_u$?
>
> > As another reviewer asks a similar question, we will analyze this later.
>
>
> We report the number of turns in the previous phase. A turn has two (user and chatbot) utterances.
> This analysis and the previous factual/non-factual one use one MTurk response, as the reviewer can see the update of GPT-3.5 (0125) when the top-5 templates are selected in the OTC baseline (CBA setting) is 1,183/1,781 = 66.42% (our numerical values in three runs is 66.57% with standard deviation 0.14).
>
> From the below table, we found that there is no specific trend that GPT-3.5 (0125) is better at shorter lengths in the previous (or, irrelevant) phase. Specifically, regardless of factual/non-factual data, when we look at the length of $\mathbf{T_p} = 0$ (i.e., $\mathbf{T_p} = \emptyset$), we observed that GPT-3.5 does *not* have better in-context knowledge editing ability (46.94%). Note also that as # data in each turn are small, more analysis (data) would be necessary.
>
> | # turns in $\mathbf{T_p}$ | Update | # data |
> |---------|--------|--------|
> | 0       |   23 (46.94%)     |    49    |
> | 1       |   44 (57.14%)    |    77    |
> | 2       |   63 (50.81%)     |    124  |
> | 3       |   65 (61.90%)    |    105  |
> | 4       |   89 (66.92%)    |    133  |
> | 5       |   89 (64.49%)    |    138  |
> | 6       |   79 (69.30%)    |    114  |
> | 7       |   78 (70.91%)    |    110  |
> | 8       |   95 (73.64%)    |    129  |
> | 9       |   84 (66.14%)    |    127  |
> | 10       |   78 (70.91%)    |   110  |
> | 11       |   51 (65.38%)    |    78  |
> | 12       |   56 (68.29%)    |    82  |
> | 13       |   53 (76.81%)    |    69  |
> | 14       |   45 (70.31%)    |    64  |
> | 15       |   45 (76.27%)    |    59  |
> | 16       |   44 (75.86%)    |    58  |
> | 17       |   34 (59.65%)    |    57  |
> | 18       |   28 (63.64%)    |    44  |
> | 19       |   36 (72.00%)    |    50  |
> | 20       |   3 (100%)    |      3    |
> | 21       |   0 (n/a)    |      0    |
> | 22       |   1  (100%)     |       1   |
> | Total   |   1,183        |    1,781  |

---

### Official Review · Reviewer_t3Go · 2024-11-10

**Soundness:** 3
**Presentation:** 1
**Contribution:** 2
**Rating:** 3
**Confidence:** 4

**Summary:**

This paper explores whether LLMs can gracefully recover from false information presented in context when a correction is subsequently issued by the user. The authors connect this setting to the phenomena of memory re-encoding (MRE) from psychology. To study this task, a new benchmark KEIC is proposed that builds on top of CoQA by introducing conflicting statements into conversations through an annotation process. The authors propose various prompting-based methods for LLM-based MRE and also a more expensive deletion method that utilizes an LLM to identify and delete portions of the chat history that contradict the corrected information. The paper also includes an extensive evaluation on KEIC with a wide range of open and closed-source models. The authors claim that their benchmark will aid in the development of chatbots that are adaptable and conducive to long-term single-user usage.

**Strengths:**

1. The paper studies a relevant topic of faithfulness over contexts that are growing increasingly large as new models are deployed.
2.  The authors draw some interesting and unexpected conclusions from their results including that GPT-4o tends to be "stubborn", often not adapting as readily to corrected information in the context compared to GPT-3.5.
3. The cost evaluation and analysis is relevant given the need for LLMs to analyze long contexts and is thorough.

**Weaknesses:**

1. **MRE:** The connection to MRE is quite tenuous. Since MRE seems to involve updating memory, this phenomenon seems most similar to the knowledge editing literature, where model parameters are updated.
2. **Benchmark Limitations:** KEIC only consists of dialogues in CoQA with a Yes/No answer. From the example in Figure 4, it also seems like the questions are quite simple and closely related to the corrected information e.g. "young / old lady" and "Is Sarah old?". These choices make KEIC seem quite limited. For instance, the benchmark does not seem to consider how corrected information alters deductive reasoning and inferences e.g. if "Mary had blue and yellow paint" is corrected to "Mary had blue and purple paint", then the answer to "Can Mary make green paint?" changes from yes to no.
Additionally, conversations in CoQA tend to be quite short. If the goal of this task is to understand how chatbots deal with corrected information over truly long contexts, a dataset that contains long conversations with topic switches should be studied such as [TopiOCQA](https://arxiv.org/pdf/2110.00768).
3. **Concerns with Evaluation:** The evaluations in section 5 do not address whether performance on questions unrelated to the modified information changes as corrected information is introduced.
4. **Notational Problems:** There is an excessive amount of notation in this paper. I would recommend that the authors rewrite sections in natural language and use notation sparingly. Some specific problems are outlined below:
   * Often there are two notations used for the same object. For instance, in 2.3 the authors refer to a turn with the $T_i$ notation, when they previously used the $(u_i, b_i)$ convention.
   * Alternatives in notation are presented for readability (e.g., dropping the indices of $R$, $Q$, and $A$), but often hamper understanding and flow given how frequently they are proposed.
5. **Clarity / Writing:** The writing is quite confusing, especially in the introduction. For instance, in the introduction, the authors claim that "Recall" is an effective technique for MRE, but Recall had not yet been mentioned nor explained and could easily be confused with the recall metric. Section headings could be much more descriptive. For instance, Section 2.2 is titled "Fact" when it contains information about the representation of facts in relation form. Finally, the KEIC benchmark seems to be one of the main contributions, but most details about its construction and annotation are relegated to the Appendix.

**Questions:**

1. What is the purpose of the proof section in Appendix D? Is the main takeaway that if the proposed methods in Section 3 worked perfectly, then questions would be perfectly answered?
2. For the deletion method, did the authors try having LLMs rewrite the inconsistent turns in the conversation instead of deleting them? This might be a better strategy since a turn may contain some relevant information to an answer even if some other information is incorrect.
3. For the deletion method, how was the inconsistency of a turn determined by an LLM?
4. How do models adapt answers when multiple facts are corrected in a single conversation?

---

> ### Author Response · Authors · 2024-11-13
> **Response to Reviewer t3Go**
>
> ## Weakness 1 (MRE)
>
> > The connection to MRE is quite tenuous. Since MRE seems to involve updating memory, this phenomenon seems most similar to the knowledge editing literature, where model parameters are updated.
>
> We clearly state the definition of MRE on page 1 of this paper.
>
> * Please refer to **in-context MRE** in Line 40 with footnote 1 (Here, "in-context" is "in a conversation." To save space, we often abbreviate "in-context MRE" as MRE).
>
> * Moreover, our title has "short-term" and can also be found in Line 98 (Practically, if we can re-encode an LLM's short-term memory on the fly, there would be...).
>
>
> ## Weakness 2 (Benchmark Limitations)
> > KEIC only consists of dialogues in CoQA with a Yes/No answer.
>
> We use exact match and do not want LLM to evaluate the other types of questions (as it often lacks explainability, even though there is a huge amount of papers doing so).
> * Please refer to Lines 157-158: Because answers are free-form in CoQA, we focus on Yes/No (YN) questions to simplify the analysis, ...
>
> * We also explicitly state the rationale behind why we choose the Y/N answer in the Limitation section (see the "KEIC Dataset" paragraph on page 17).
>
> > From the example in Figure 4, it also seems like the questions are quite simple and closely related to the corrected information e.g. "young / old lady" and "Is Sarah old?". These choices make KEIC seem quite limited. For instance, the benchmark does not seem to consider how corrected information alters deductive reasoning and inferences e.g. if "Mary had blue and yellow paint" is corrected to "Mary had blue and purple paint", then the answer to "Can Mary make green paint?" changes from yes to no.
>
> The questions are directly re-used from the CoQA dataset.
> * See Figure 2: We consider that an LLM updates its knowledge if its answer to the **same** question is changed.
> * Also see the definition of effective in Section 2.2: The question $q$ is not changed in Line 152.
>
> As for the updated utterance, the real data in the KEIC dataset is not as simple as shown in Figure 4; it is just a simple but informative example.
> * See Examples 3-7 for the complicated ones in Appendix E (mentioned in Lines 285-286 and Reproducibility Statement).
> * Please also refer to the caption of Figure 3 (We extend the CoQA data by adding a non-trivial (i.e., multi-hop reasoning), effective fact $R'_4$ that contradicts $R_4$ in the story).
>
> > Additionally, conversations in CoQA tend to be quite short. If the goal of this task is to understand how chatbots deal with corrected information over truly long contexts, a dataset that contains long conversations with topic switches should be studied such as TopiOCQA.
>
> Please refer to Lines 286-287: The average number of turns in the previous phase is 8.27 and 8.48, respectively.
>
> Note that the TopiOCQA dataset is longer than CoQA **only if** we (1) filter the data with turn $<$ 3 and (2) **always** use the last question for testing. However, It breaks the randomness. Not to mention the rationale is often not properly labeled (compared to CoQA).
>
> ```
> >>> from datasets import load_dataset
> >>> ds = load_dataset("McGill-NLP/TopiOCQA")
> >>> ds['train']['Turn_no'][2] # has test QA turn
> 3
> >>> ds['train']['Context'][2]
> ["what was australia's contribution to the battle of normandy?", 'The army personnel and thousands of Australian airmen took part in the battle.', 'was the battle fought in australia?', 'UNANSWERABLE']
> >>> num_turn = [_ for _ in ds['train']['Turn_no']]
> >>> sum(num_turn) / len(num_turn)
> 7.41
> >>> tmp = [_ for _ in num_turn if _ >= 3]
> >>> sum(tmp) / len(tmp)
> 8.49
> ```
>
> ## Weakness 3 (Concerns with Evaluation)
>
> > The evaluations in section 5 do not address whether performance on questions unrelated to the modified information changes as corrected information is introduced.
>
> As this paper focuses on studying whether LLMs can change the corresponding answer, another direction for future work would be asking questions related to the update ("ripple effect" in [1]).
> Asking irrelevant questions can be viewed as an (easy) out-of-scope question (see [2]), but they are *slightly* less interested in evaluating the KE task, except they are designed to trick the LLM. See the example below:
>
> ```
> old fact: A is the president of country C.
> new fact: B is the president of country C.
>
> fact (fixed): A's sister is X.
> fact (fixed): B's sister is Y.
> fact (fixed): D is the president of country Z.
>
> in-scope question: Who is the president in country C? (A -> B)
> ("hard") out-of-scope question: Who is the president in country Z? (D)
> "ripple effect" question: What is the president's sister's name in country C? (X -> Y)
>
> irrelevant (i.e., "easy" out-of-scope) question: What is the color of the sky? (blue)
> ```
>
> [1] Cohen et al., Evaluating the ripple effects of knowledge editing in language models, TACL 2024
>
> [2] Mitchell et al., Memory-based model editing at scale, ICML 2022

---

> > ### Author Response · Authors · 2024-11-13
> > **Response to Reviewer t3Go (cont.)**
> >
> > ## Weakness 4 (Notational Problems)
> >
> > > There is an excessive amount of notation in this paper. I would recommend that the authors rewrite sections in natural language and use notation sparingly.
> >
> > **This paper standardizes the dataset construction and the proposed KEIC framework, and the notation is consistently used for proving our IC-MRE algorithm**, so the math notation is necessary in Section 2 (also see Lines 954-959 in Related Work).
> >
> > If the notations are omitted in the main content, then the dataset construction is not standardized, and we have to re-state the definition of fact and KEIC framework in Appendix D.
> >
> > > * Often there are two notations used for the same object. For instance, in 2.3 the authors refer to a turn with the $T_i$ notation, when they previously used the $(u_i, b_i)$ convention.
> >
> > There are other papers that use "turn" to denote a pair of user and chatbot utterances (e.g., **the CoQA paper** and [1]), and, perhaps, after LLM-as-a-Judge [2], the term "turn" gradually shifts to this definition from our knowledge.
> >
> > We formalize this task and use this as a standard. Consequently, defining each turn $T_i$ containing a pair of user-chatbot utterances $(u_i, b_i)$ is essential.
> >
> > > * Alternatives in notation are presented for readability (e.g., dropping the indices of $R$, $Q$, and $A$, but often hamper understanding and flow given how frequently they are proposed.
> >
> > The subscript is dropped because *we are interested in asking the $i$-th question*, so including this $i$ in the main content is redundant. Also, see Lines 1149-1150 regarding the necessity of dropping indices (subscript of subscript).
> >
> > **This convention is also used in many papers.** For example, in [3], they write the following sentence in Section 4:
> > Throughout this paper, our focus will be on states representing the last subject token $S$ of prompt $p_i$, so we shall abbreviate  $h_i^l = h_{[S]}^l(p_i)$, ...
> >
> > If this is still a concern, we can move this abbreviation sentence to Appendix D.
> >
> > [1] Sun et al., Parrot: Enhancing Multi-Turn Instruction Following for Large Language Models, ACL 2024
> >
> > [2] Zheng et al., Judging LLM-as-a-judge with MT-bench and chatbot arena, NIPS 2023
> >
> > [3] Meng et al., Mass-editing memory in a transformer, ICLR 2023
> >
> > ## Weakness 5 (Clarity / Writing)
> >
> > > The writing is quite confusing, especially in the introduction. For instance, in the introduction, the authors claim that "Recall" is an effective technique for MRE, but Recall had not yet been mentioned nor explained and could easily be confused with the recall metric.
> >
> > This can be fixed by adding "method" or "§3".
> >
> > > Section headings could be much more descriptive. For instance, Section 2.2 is titled "Fact" when it contains information about the representation of facts in relation form.
> >
> > We already mention this in the *first* paragraph of Section 2.
> > * See Lines 119-120: Next, we define how to elicit knowledge stored in LLMs and formalize its form in a conversation.
> >
> > > Finally, the KEIC benchmark seems to be one of the main contributions, but most details about its construction and annotation are relegated to the Appendix.
> >
> > The construction of our dataset is in Section 4.1 (which does *not mostly* appear in the Appendix) because we already define the "effective" in Section 2.2.
> > * Despite this, how exactly a fact is "effective **and fluent**" depends on each individual. For transparency, we include Appendix E to give examples of how we accept responses that are "effective, fluent, and ethically sound" (as stated in Lines 277 and 281).
> > * Note also that the dataset construction is also mentioned in the previous sections (e.g., Section 2.4, as stated in Line 276).
> >
> > ## Question
> >
> > > 1. What is the purpose of the proof section in Appendix D? Is the main takeaway that if the proposed methods in Section 3 worked perfectly, then questions would be perfectly answered?
> >
> > Yes, you got it right. It serves as the purpose that the context (re-written by LLMs) does not have old knowledge, given that the Inconsistent and Delete modules are perfect.
> >
> > > 2. For the deletion method, did the authors try having LLMs rewrite the inconsistent turns in the conversation instead of deleting them? This might be a better strategy since a turn may contain some relevant information to an answer even if some other information is incorrect.
> >
> > It **is** "re-write," the "delete" means "deleting the old fact."
> > * See the prompt in Appendix F for details, as stated in the Reproducibility Statement.
> >
> > > 3. For the deletion method, how was the inconsistency of a turn determined by an LLM?
> >
> > Please see Appendix F and Figure 9 (on page 21). In this paper, GPT-3.5 (0613) judges on its own.
> >
> > > 4. How do models adapt answers when multiple facts are corrected in a single conversation?
> >
> > This is an interesting work for future research as stated in the Limitations section in Lines 913-914 (on page 17).

---

> > > ### Comment · Reviewer_t3Go · 2024-11-19
> > > **Reply to author response**
> > >
> > > Thank you for a comprehensive response.
> > >
> > > >Please refer to in-context MRE in Line 40 with footnote 1 (Here, "in-context" is "in a conversation." To save space, we often abbreviate "in-context MRE" as MRE).
> > >
> > > My question is more about why you are making a connection to MRE at all. Specifically, it seems like there is little specific discussion that draws connections with cognitive science literature aside from the first paragraph of the introduction. Through the introduction of the term in-context MRE, I understand the attempt to make a connection between MRE and LLM memory / understanding.
> > >
> > > However, in order to realistically claim such a connection, the authors should more deliberately show a connection by (1) providing a more comprehensive review of MRE for the vast majority of ML researchers who are unfamiliar with this term (potentially this could go in a background section) (2) illustrate concrete connections with your task and MRE (3) explain why the community should be interested in *in-context* MRE given the large amount of model editing research which seems to explore similar ideas from a more analogous standpoint i.e. brain encoding new information and models encoding new information in parameters.
> > >
> > > > We use exact match and do not want LLM to evaluate the other types of questions (as it often lacks explainability, even though there is a huge amount of papers doing so).
> > >
> > > Even if LLM evaluation is not perfect, I think it would allow you to greatly expand the types of questions as I mentioned in my review. The authors might consider a subset of their dataset with reliable yes/no evaluation and another subset that is evaluated with an LLM.
> > >
> > > > It is "re-write," the "delete" means "deleting the old fact."
> > >
> > > > Please see Appendix F and Figure 9 (on page 21). In this paper, GPT-3.5 (0613) judges on its own.
> > >
> > > Thank you for the clarification.
> > >
> > > > Yes, you got it right. It serves as the purpose that the context (re-written by LLMs) does not have old knowledge, given that the Inconsistent and Delete modules are perfect.
> > >
> > > In this case, I do not think this section is needed since it is quite trivial and unnecessarily complicated. Most papers that use some sort of prompting approach could also include such a proofs section proving that their method works perfectly *if* the prompted LLM performs its role perfectly, but they almost always do not. I think if removing this section allows you to simplify the notation in the main text, it would be a helpful change.

---

> > > > ### Author Response · Authors · 2024-11-20
> > > >
> > > > > My question is more about why you are making a connection to MRE at all. Specifically, it seems like there is little specific discussion that draws connections with cognitive science literature aside from the first paragraph of the introduction. ...
> > > >
> > > > We could remove the MRE term and stick to KEIC if that improves the clarity of this paper.
> > > >
> > > > > Even if LLM evaluation is not perfect, I think it would allow you to greatly expand the types of questions as I mentioned in my review. The authors might consider a subset of their dataset with reliable yes/no evaluation and another subset that is evaluated with an LLM.
> > > >
> > > > Since the dataset is constructed and there is not enough time to run the experiments on other types of question, we can add LLM evaluation of Yes/No question.
> > > > Specifically, though not included in the current paper, we previously have conducted the answer extraction (AE) step of the OTC baseline on our KEIC dataset (# = 1,781) using Llama-2 (7B and 13B), Llama-3 (8B), Vicuna (7B, 13B, and 33B), and Gemma-2 (2B, 9B, and 27B).
> > > >
> > > > The AE step is the same as the paper (*Therefore, based on your previous response, your answer to the last question is more likely to be "Yes", "No"? You must output "Yes" or "No" first.*).
> > > >
> > > > The results show that some LLMs improve the Update (accuracy metric) to some extent, while others do not (even deteriorate). We will include these Figures in Appendix H for visualization (just like the Recall method v.s. the oracle of Recall method in Figure 12).
> > > >
> > > > > In this case, I do not think this section is needed since it is quite trivial and unnecessarily complicated. Most papers that use some sort of prompting approach could also include such a proofs section proving that their method works perfectly if the prompted LLM performs its role perfectly, but they almost always do not. I think if removing this section allows you to simplify the notation in the main text, it would be a helpful change.
> > > >
> > > > Could the reviewer be more specific about which notation/equation/line is confusing? As this paper standardizes the dataset construction and the KEIC framework, some notations are not merely used to prove the algorithm. We are unsure about which part is complicated for the reviewer or potential readers. Thank you.

---

> ### Comment · Reviewer_t3Go · 2024-11-28
> **Reply to author response**
>
> > We could remove the MRE term and stick to KEIC if that improves the clarity of this paper.
> Yes, I think that this change would help clarify and avoid confusion.
>
>
> >Since the dataset is constructed and there is not enough time to run the experiments on other types of question, we can add LLM evaluation of Yes/No question. Specifically, though not included in the current paper, we previously have conducted the answer extraction (AE) step of the OTC baseline on our KEIC dataset (# = 1,781) using Llama-2 (7B and 13B), Llama-3 (8B), Vicuna (7B, 13B, and 33B), and Gemma-2 (2B, 9B, and 27B).
>
> I agree that there may not be enough time to change this now, but I think it would be very useful in a camera-ready version or future submission.
>
> > Could the reviewer be more specific about which notation/equation/line is confusing? As this paper standardizes the dataset construction and the KEIC framework, some notations are not merely used to prove the algorithm. We are unsure about which part is complicated for the reviewer or potential readers. Thank you.
>
> I mentioned some cases in my initial review. One other example: the authors use notation involving subscripting $T$ such as $T_f , T_u, T_i, T_p, T_c, T_v, T_r, T_d$. This notation is overbearing and quite difficult for the reader to keep track of. I would encourage the authors to instead denote phases and methods explicitly with natural language.
>
>
> As I believe the paper would benefit from another round of revision, I will keep my current score.

---

> > ### Author Response · Authors · 2024-12-04
> >
> > > This notation is overbearing and quite difficult for the reader to keep track of. I would encourage the authors to instead denote phases and methods explicitly with natural language.
> >
> > In our KEIC framework, there are only the first four notations. Please refer to Lines 175-176 in Section 2.3 ($\mathbf{T_u}$ is a general notation for KEIC (see Section 3))
> >
> > Since we propose four methods in Section 3, we then dissect the update phase ($\mathbf{T_u}$) in Line 202. Moreover, the latter four notations are mainly used for visualization the conversation flow in Figure 3 and Table 1 (to maximize reproducibility).

---

### Comment · Area_Chair_emLW · 2024-11-19
**Reviewer-author discussion**

@Reviewers:
* Thank you for your reviews. Since the authors have responded to each of your reviews, it would be helpful if you could follow up and let them know whether you found their responses satisfactory. @Reviewer t3Go, thank you for already following up.

@Authors:

* It seems you don’t agree with most of the stated weaknesses, aside e.g. from acknowledging in a few cases that the reviewers’ points could inspire interesting future work. Do you have anything further to add regarding the weaknesses? My reason for asking is that the discussion phase often aims for both parties to agree on how the paper can be improved during this period or (if applicable) in the camera-ready version. In some cases, I noticed you responded to broad concerns (e.g., about writing/clarity) with quite specific comments (e.g., about notation). Are you planning to address these broader concerns? If so, it would be helpful to state your plans or ask the reviewers to elaborate on their feedback.
* I found your approach of indirectly addressing concerns with 'please refer to Lines xxx-yyy / Figure / Table y' a bit unclear at times. Please keep in mind that ACs/SACs/PCs sometimes read the comments but may not have enough time to dive into each paper individually. Additionally, this approach can sometimes come across as dismissive or assuming that the reviewers misunderstood the paper or missed key points, though this may not always be the case (e.g., see Reviewer t3Go's latest comment on Weakness 1). If there’s any uncertainty, it might be helpful to assume the reviewers understood and provide additional clarifications to ensure alignment. Here are a few other examples:
  * Reviewers t3Go, T49M, q6Uz, and yjK4 have independently raised concerns about the writing, clarity, organization, readability, notation, and jargon. However, your responses generally focus on pointing out the rationales behind your choices in the paper. It’s worth considering the possibility that the reviewers read these rationales yet still find the paper difficult to follow. Do you have any plans or suggestions for improving the paper’s readability? If you are not sure what else to change, feel free to ask reviewers for further clarifications.
  * Your rebuttal to the limitation "KEIC only consists of dialogues in CoQA with a Yes/No answer" refers to the limitation section (page 17). So, it seems both parties agree this is a limitation...
  * You responded: "Please refer to Lines 286–287: The average number of turns in the previous phase is 8.27 and 8.48, respectively." Does this imply that you disagree with the claim that "CoQA tends to be quite short"? If so, please state this explicitly. Averages around 8 suggest about 4 Q&A pairs per conversation, which, to me, still seems quite short.

Thank you,

AC

---

### Author Response · Authors · 2024-11-28

Dear ACs and Reviewers,

We sincerely appreciate your thoughtful and constructive feedback. We thank reviewers for recognizing our work (1) studies an interesting topic in LLM's conversational memory or faithfulness/reliability of LLM (t3Go, T49M, and q6Uz), (2) constructs a high-quality/sizable dataset (T49M, MxUL, and yjK4), (3) has extensive experiments/thorough analyses (t3Go, T49M, and yjK4), and (4) "draws some surprising/unexpected conclusions that worth investigating further (e.g., GPT-4o worse than GPT-3.5 in this task)" (t3Go and T49M).

In the revised paper, we have carefully addressed your comments and made necessary updates. We summarize and address the common/major concerns as follows:

1. To improve clarity, we remove the term "MRE" (t3Go and MxUL) and consistently use "KEIC" or "in-context KE." We also renamed the "Recall" method as "Reiterate" (t3Go and T49M). We changed the title accordingly.
2. Regarding the evaluation, Reviewers (t3Go, q6Uz, and yjK4) ask why not (a) include questions other than Yes/No or (b) use LLM to evaluate the answer (or other metrics).

* We acknowledge that the "dataset" focusing on Yes/No is a limitation (as it is explicitly stated in the Limitations section). On the other hand, we would like to emphasize that our proposed "framework" as well as the conducted experiments do not depend on this scenario, which can be applied to other open-domain questions or chat datasets. In this paper, *our primary goal is to propose a "unified and formalized framework" for the KEIC task*, as we standardize the KEIC task to avoid misinterpretation, which frequently happens in the current work of the self-correction task (please refer to the most recent (2024/06) survey paper in [1]).

* Because the current (#=1,781) dataset involved tremendous human effort and budget, there is not enough time to follow the same criteria during the discussion period to generate high-quality data for other questions. Instead, **we have included the result of using LLM for evaluation (see Figure 13 on page 33)**. This figure also demonstrates that some LLMs have no significant improvement in top-5 performance (or, even worse) when using LLM to evaluate the result (which lacks explainability).

* As for other evaluation metrics, we address that the answer to a Yes/No question is short since a person would normally reply with a Yes/No to it. Hence, using the exact match (accuracy) is suitable in this paper, which does not make the evaluation of this paper incomplete.

3. Additional experiments/analysis:

> Reviewer MxUL suggests a (better) test question for probing: "Based on the recent correction from the user, how would you answer to this question: {question}?"

We conduct experiments using GPT-4o (mini) and find that the result, though having some improvements in update, does not solve this task ($<$ 70%), and, interestingly, still underperforms compared to "GPT-3.5 (0125) with the original (same) question."

> Reviewer T49M asks whether the authors differentiate the fact/non-fact data.

In the updated version (see Table 8 on page 33), we analyze the GPT-3.5 (0125) and GPT-4o (0806) results and deduce that the real-world fact is relatively hard to edit. Moreover, the gap between factual and non-factual in update is significant in GPT-4o (fact: 25.04; non-fact: 40.73) compared to GPT-3.5 (fact: 62.20; non-fact: 69.95).

This figure could also explain why those GPT-4 LLMs (4o, 4o-mini, 4) underperform GPT-3.5 (0125) in this task.

> Reviewers T49M and q6Uz suggest the analysis of different locations of user correction

We insert the correction utterances (exactly) in the middle of the dataset and test them on GPT-4o (mini).

The result supports our hypothesis that GPT-4o (mini) performs worse in the middle than in the CAM (i.e., closer to the false fact) and CBA (i.e., closer to the test question) setting.

> Reviewer T49M and MxUL are interested in whether some (e.g., top-5) templates share across different KEIC methods.

We provide some analysis and found that OTC and Recall often share the same templates in Reviewer T49M's response.

> Reviewer T49M is interested in whether it is easier/difficult when the # turns grows in the CAM or CBA setting.

We provide the analysis of GPT-3.5 (0125) in the CBA setting and conjecture that there is no significant "trend" in GPT-3.5 (0125) LLM.

> Reviewer MxUL asks whether "this paper involves managing an external database that saves and updates facts within the conversation." Similarly, Reviewer T49M asks whether "this task is not merely a different sub-perspective of the conversational memory problem that could be potentially solved by using RAG + external memory module."

We clarify that both the oracle of Reiterate and Deletion encompass this experimental setting (a (perfect) external system to manage context).

[1] Kamoi et al., When Can LLMs Actually Correct Their Own Mistakes? A Critical Survey of Self-Correction of LLMs, TACL 2024

Regards,

Authors

---

### Meta-Review · Area_Chair_emLW · 2024-12-15

**Metareview:**

This paper examines the ability of large language models (LLMs) to adapt to corrections provided mid-conversation, a skill akin to what cognitive psychology terms Memory Re-Encoding (MRE). The authors formalize this scenario within the Knowledge Editing in Conversation (KEIC) framework, introducing a benchmark dataset derived from CoQA with human-annotated conflicting statements to evaluate LLMs’ adaptability in context. The study includes extensive experiments to analyze the effectiveness of four proposed correction methods across various LLMs.

The reviewers identified several strengths in the work. The paper addresses the challenge of enabling LLMs to incorporate corrections in large and dynamic contexts by proposing the KEIC framework for in-context knowledge editing without requiring model fine-tuning. It also contributes a dataset designed for studying conversational memory updates and conducts experiments that reveal interesting differences in how LLMs handle corrections.

However, the reviewers raised significant concerns, particularly regarding:

* Motivation: Two reviewers were unconvinced by the framing around MRE, with one noting that the task closely resembles knowledge editing. While the paper differs methodologically from prior work by focusing on an in-context setting, some aspects of the new MRE framework and its connection to psychology felt gratuitous. Beyond the first paragraph of the introduction, there was little follow-up on the connection to human MRE abilities or their psychological basis. Although the updated version of the paper attempts to address this issue by largely replacing references to MRE with references to KEIC, this raises doubts about the level of contribution, as KEIC is closely related to knowledge editing (KE), which has been extensively studied in prior work. The primary difference appears to be whether KE is performed in-context or not. However, the distinction between in-context and fine-tuning typically pertains more to the *method* than to the *task* (e.g., in-context and fine-tuning can be alternative approaches to the same task), which introduces uncertainties about the task's significance and usefulness.

* Clarity: Reviewers found the paper’s writing, organization, readability, and notation lacking, which made it difficult to follow at times.

As none of the reviewers were willing to raise their ratings during the discussion period, with significant motivation problems remaining, I recommend rejection based on the negative reviews and ratings.

**Additional Comments On Reviewer Discussion:**

During discussions with the authors, reviewers’ initial concerns were compounded. Many of the authors’ responses emphasized that the characteristics highlighted by the reviewers were intentional, and the authors appeared to suggest that reviewers misunderstood key parts of the paper. Even if misunderstandings were present, this only further underscored the clarity issues. Additionally, the authors’ reluctance to acknowledge significant weaknesses (until late in the discussion period) raised doubts about their willingness to address the reviewers' feedback, which was particularly concerning given the three reject recommendations from the reviewers.

In the reviewers-AC discussion, the reviewers acknowledged reading the authors’ latest responses and the updated version of the paper. While we appreciated that some key concerns, such as the references to MRE, had finally been addressed, the changes involved substantial reframing of the paper and its contributions. The revised version focused more on KEIC as a dataset contribution, creating uncertainties about the level of contribution and giving the sense that the new framing (including a significantly different title) would require a new round of reviewing.

---

### Decision · Program_Chairs · 2025-01-22

Reject